# RANDOMNESS HELPS RIGOR:
# A PROBABILISTIC LEARNING RATE SCHEDULER BRIDGING THEORY AND DEEP LEARNING PRACTICE

## ABSTRACT

Learning rate schedulers have shown great success in speeding up the convergence of learning algorithms in practice. However, their convergence to a minimum has not been theoretically proven. This difficulty mainly arises from the fact that, while traditional convergence analysis prescribes to monotonically decreasing (or constant) learning rates, schedulers opt for rates that often increase and decrease through the training epochs. We aim to bridge this gap by proposing a probabilistic learning rate scheduler (PLRS) that does not conform to the monotonically decreasing condition, while achieving provable convergence guarantees. To demonstrate the practical effectiveness of our approach, we evaluate it on deep neural networks across both vision and language tasks, showing competitive or superior performance compared to state-of-the-art learning rate schedulers. Specifically, our experiments include (a) image classification on CIFAR-10, CIFAR-100, Tiny ImageNet, and ImageNet-1K using ResNet, WRN, VGG, and DenseNet architectures, and (b) language model fine-tuning on the SQuAD v1.1 dataset with pretrained BERT. Notably, on ImageNet-1K with ResNet-50, our method surpasses the leading knee scheduler by 2.79% in classification accuracy.

## 1 INTRODUCTION

Over the last two decades, there has been an increased interest in analyzing the convergence of gradient descent-based algorithms. This can be majorly attributed to their extensive use in the training of neural networks and their numerous derivatives. Stochastic Gradient Descent (SGD) and their adaptive variants such as Adagrad (Duchi et al., 2011), Adadelta (Zeiler, 2012), and Adam (Kingma & Ba, 2014) have been the choice of optimization algorithms for most machine learning practitioners, primarily due to their ability to process enormous amounts of data in batches. Even with the introduction of adaptive optimization techniques that use a default learning rate, the use of stochastic gradient descent with a tuned learning rate was quite prevalent, mainly due to its generalization properties (Zhou et al., 2020). However, tuning the learning rate of the network can be computationally intensive and time consuming.

Various methods to efficiently choose the learning rate without excessive tuning have been explored. One of the initial successes in this domain is the random search method (Bergstra & Bengio, 2012); here, a learning rate is randomly selected from a specified interval across multiple trials, and the best performing learning rate is ultimately chosen. Following this, more advanced methods such as Sequential Model-Based Optimization (SMBO) (Bergstra et al., 2013) for the choice of learning rate became prevalent in practice. SMBO represents a significant advancement over random search by tracking the effectiveness of learning rates from previous trials and using this information to build a model that suggests the next optimal learning rate. A tuning method for shallow neural networks based on theoretical computation of the Hessian Lipschitz constant was proposed by Tholeti et al. (Tholeti & Kalyani, 2020).

Several works on training deep neural networks prescribed the use of a decaying Learning Rate (LR)[1] scheduler (He et al., 2016; Zhang et al., 2019; Szegedy et al., 2015). Recently, much attention

---

[1]We abbreviate learning rate only in the context of learning rate scheduler as LR scheduler.

has been paid to cyclically varying learning rates (Smith, 2017). By varying learning rates in a triangular schedule within a predetermined range of values, the authors hypothesize that the optimal learning rate lies within the chosen range, and the periodic high learning rate helps escape saddle points. Although no theoretical backing has been provided, it was shown to be a valid hypothesis owing to the presence of many saddle points in a typical high dimensional learning task (Dauphin et al., 2014). Many variants of the cyclic LR scheduler have henceforth been used in various machine learning tasks (Howard & Ruder, 2018; Dhillon et al., 2020; Andriushchenko & Flammarion, 2020). A cosine-based cyclic LR scheduler proposed by Loshchilov et al. (Loshchilov & Hutter, 2017b) has also found several applications, including Transformers (Zamir et al., 2022; Caron et al., 2021). Following the success of the cyclic LR schedulers, a one-cycle LR scheduler proposed by Smith et al. (Smith & Topin, 2019) has been observed to provide faster convergence empirically; this was attributed to the injection of 'good noise' by higher learning rates which helps in convergence. Although empirical validation and intuitions were provided to support the working of these LR schedulers, a theoretical convergence guarantee has not been provided to the best of our knowledge.

There is extensive research on the convergence behavior of perturbed SGD methods, where noise is added to the gradient during updates. In Jin et al. (Jin et al., 2017), the vanilla gradient descent is perturbed by samples from a ball whose radius is fixed using the optimization function-specific constants. They show escape from a saddle point by characterizing the distribution around a perturbed iterate as uniformly distributed over a perturbation ball along which the region corresponding to being stuck at a saddle point is shown to be very small. In Ge at al. (Ge et al., 2015), the saddle point escape for a perturbed stochastic gradient descent is proved using the second-order Taylor approximation of the optimization function, where the perturbation is applied from a unit ball to the stochastic gradient descent update. Following Ge at al. (Ge et al., 2015), several works prove the convergence of noisy stochastic gradient descent in the additive noise setting (Zhang et al., 2017; Jin et al., 2021; Arjevani et al., 2023; Yiming Cao et al., 2025). In contrast to the above works which operate in the *additive* noise setting, our proposed LR scheduler results in *multiplicative* noise. Analyzing the convergence behavior under the new multiplicative noise setting is fairly challenging and results in a non-trivial addition to the literature.

## 1.1 MOTIVATION

Traditional convergence analysis of gradient descent algorithms and its variants requires the use of a constant or a decaying learning rate (Nesterov, 2014). However, with the introduction of LR schedulers, the learning rates are no longer monotonically decreasing. Rather, their values heavily fluctuate, with the occasional use of very large learning rates. Although there are ample justifications provided for the success of such methods, there are no theoretical results which prove that stochastic gradient descent algorithms with fluctuating learning rates converge to a local minimum in a non-convex setting. With the increase of emphasis on trustworthy artificial intelligence, we believe that it is important to no longer treat optimization algorithms as black-box models, and instead provide provable convergence guarantees while deviating from the proven classical implementation of the descent algorithms. In this work, we aim to bridge the gap by providing rigorous mathematical proof for the convergence of our proposed probabilistic LR scheduler with SGD.

## 1.2 OUR CONTRIBUTIONS

1. We propose a new Probabilistic Learning Rate Scheduler (PLRS) where we model the learning rate as an instance of a random noise distribution.

2. We provide convergence proofs to show that SGD with our proposed PLRS converges to a local minimum in Section 4. To the best of our knowledge, we are the first to theoretically prove convergence of SGD with a LR scheduler that does not conform to constant or monotonically decreasing rates. We show how our LR scheduler, in combination with inherent SGD noise, speeds up convergence by escaping saddle points.

3. Our proposed probabilistic LR scheduler, while provably convergent, can be seamlessly ported into practice without the knowledge of theoretical constants (like gradient and Hessian-Lipschitz constants). We illustrate the efficacy of the PLRS through extensive experimental validation, where we compare the accuracies with state-of-the-art schedulers in Section 5. We show that the proposed method outperforms popular schedulers such as

cosine annealing (Loshchilov & Hutter, 2017b), one-cycle (Smith & Topin, 2019), knee (Iyer et al., 2023) and the multi-step scheduler when used with ResNet-110 on CIFAR-100, DenseNet-40-12 on CIFAR-100, VGG-16 on CIFAR-10, WRN-28-10 on CIFAR-10 datasets and ResNet-50 on Tiny ImageNet datasets respectively, while performing competitively with baselines when used on NLP datasets like SQuAD v1.1 and IWSLT'14 with BERT and Transformer respectively. Furthermore, we outperform the baseline results on the CommonVoice 11.0 Hindi dataset with Whisper model on the Automatic Speech Recognition application (ASR) application. We also observe lesser spikes in the training loss across epochs which leads to a faster and more stable convergence. We provide our base code with all the hyperparameters for reproducibility in the supplemental material.

## 2 PROBABILISTIC LEARNING RATE SCHEDULER

Let $f : \mathbb{R}^d \to \mathbb{R}$ be the function to be minimized. The unconstrained optimization, $\min_{\mathbf{x} \in \mathbb{R}^d} f(\mathbf{x})$, can be solved iteratively using stochastic gradient descent whose update equation at time step $t$ is given by

$$\mathbf{x}_{t+1} = \mathbf{x}_t - \eta_{t+1} g(\mathbf{x}_t). \tag{1}$$

Here, $\eta_{t+1} \in \mathbb{R}$ is the learning rate and $g(\mathbf{x}_t)$ is the stochastic gradient of $f(\mathbf{x})$ at time $t$. In this work, we propose a new LR scheduler, in which the learning rate $\eta_{t+1}$ is sampled from a uniform random variable,

$$\eta_{t+1} \sim \mathcal{U}[L_{min}, L_{max}], \quad 0 < L_{min} < L_{max} < 1. \tag{2}$$

Note that contrary to existing LR schedulers, which are deterministic functions, we propose that the learning rate at each time instant be a realization of a uniformly distributed random variable. Although the learning rate in our method is not scheduled, but is rather chosen as a random sample at every time step, we call our proposed method Probabilistic LR scheduler to keep in tune with the body of literature on LR schedulers. In order to represent our method in the conventional form of the stochastic gradient descent update, we split the learning rate $\eta_{t+1}$ into a constant learning rate $\eta_c$ and a random component, as $\eta_{t+1} = \eta_c + u_{t+1}$, where $u_{t+1} \sim \mathcal{U}[L_{min} - \eta_c, L_{max} - \eta_c]$. The stochastic gradient descent update using the proposed PLRS (referred to as SGD-PLRS) takes the form

$$\mathbf{x}_{t+1} = \mathbf{x}_t - (\eta_c + u_{t+1}) g(\mathbf{x}_t) = \mathbf{x}_t - \eta_c \nabla f(\mathbf{x}_t) - \mathbf{w}_t, \tag{3}$$

where we define $\mathbf{w}_t$ as

$$\mathbf{w}_t = \eta_c g(\mathbf{x}_t) - \eta_c \nabla f(\mathbf{x}_t) + u_{t+1} g(\mathbf{x}_t). \tag{4}$$

Here, $\nabla f(\mathbf{x}_t)$ refers to the true gradient, i.e., $\nabla f(\mathbf{x}_t) = \mathbb{E}[g(\mathbf{x}_t)]$. Note that in equation 3, the term $\mathbf{x}_t - \eta_c \nabla f(\mathbf{x}_t)$ resembles the vanilla gradient descent update and $\mathbf{w}_t$ encompasses the noise in the update; the noise is inclusive of both the randomness due to the stochastic gradient as well as the randomness from the proposed LR scheduler. We set $\eta_c = \frac{L_{min} + L_{max}}{2}$ so that the noise $\mathbf{w}_t$ is zero mean, which we prove later in Lemma 1.

**Remark 1.** *Note that a periodic LR scheduler such as triangular, or cosine annealing based scheduler can be considered as a single instance of our proposed PLRS. The range of values assigned to the learning rate $\eta_{t+1}$ is pre-determined in both cases. In fact, for any LR scheduler, the basic mechanism is to vary the learning rate between a low and a high value - the high learning rates help escape the saddle point by perturbing the iterate, whereas the low values help in convergence. This pattern of switching between high and low values can be achieved through both stochastic and deterministic mechanisms. While the current literature explores the deterministic route (without providing analysis), we propose and explore the stochastic variant here and also provide a detailed analysis.*

## 3 PRELIMINARIES AND DEFINITIONS

We denote the Hessian of a function $f : \mathbb{R}^d \to \mathbb{R}$ at $\mathbf{x} \in \mathbb{R}^d$ as $\mathbf{H}(\mathbf{x}) := \nabla^2 f(\mathbf{x})$ and the minimum eigenvalue of the Hessian as $\lambda_{min}(\mathbf{H}(\mathbf{x})) := \lambda_{min}(\nabla^2 f(\mathbf{x}))$ respectively.

**Definition 1.** *A function $f : \mathbb{R}^d \to \mathbb{R}$ is said to be $\beta$-smooth (also referred to as $\beta$-gradient Lipschitz) if, $\exists \beta \geq 0$ such that,*

$$\|\nabla f(\mathbf{x}) - \nabla f(\mathbf{y})\| \leq \beta \|\mathbf{x} - \mathbf{y}\|, \quad \forall \mathbf{x}, \mathbf{y} \in \mathbb{R}^d. \tag{5}$$

**Definition 2.** *A function $f : \mathbb{R}^d \to \mathbb{R}$ is said to be $\rho$-Hessian Lipschitz if, $\exists \rho \geq 0$ such that,*

$$\|\boldsymbol{H}(\boldsymbol{x}) - \boldsymbol{H}(\boldsymbol{y})\| \leq \rho \|\mathbf{x} - \mathbf{y}\|, \quad \forall \mathbf{x}, \mathbf{y} \in \mathbb{R}^d. \tag{6}$$

Informally, a function is said to be gradient/Hessian Lipschitz, if the rate of change of the gradient/Hessian with respect to its input is bounded by a constant, i.e., the gradient/Hessian will not change rapidly. We now proceed to define approximate first and second-order stationary points of a given function $f$.

**Definition 3.** *For a function $f : \mathbb{R}^d \to \mathbb{R}$ that is differentiable, we say $\mathbf{x} \in \mathbb{R}^d$ is a $\nu$- first-order stationary point ($\nu$-FOSP), if for a small positive value of $\nu$, $\|\nabla f(\mathbf{x})\| \leq \nu$.*

Before we define an $\epsilon$-second order stationary point, we define a saddle point.

**Definition 4.** *For a $\rho$-Hessian Lipschitz function $f : \mathbb{R}^d \to \mathbb{R}$ that is twice differentiable, we say $\boldsymbol{x} \in \mathbb{R}^d$ is a saddle point if,*

$$\|\nabla f(\mathbf{x})\| \leq \nu \quad and \quad \lambda_{min}(\boldsymbol{H}(\boldsymbol{x})) \leq -\gamma,$$

*where $\nu, \gamma > 0$ are arbitrary constants.*

For a convex function, it is sufficient if the algorithm is shown to converge to the $\nu$-FOSP as it would be the global minimum. However, in the case of a non-convex function, a point satisfying the condition for a $\nu$-FOSP may not necessarily be a local minimum, but could be a saddle point or a local maximum. Hence, the Hessian of the function is required to classify it as a second-order stationary point, as defined below. Note that, in our analysis, we prove convergence of SGD-PLRS to the approximate second-order stationary point.

**Definition 5.** *For a $\rho$-Hessian Lipschitz function $f : \mathbb{R}^d \to \mathbb{R}$ that is twice differentiable, we say $\mathbf{x} \in \mathbb{R}^d$ is a $\nu$-second-order stationary point ($\nu$-SOSP) if,*

$$\|\nabla f(\mathbf{x})\| \leq \nu \quad and \quad \lambda_{min}(\boldsymbol{H}(\boldsymbol{x})) \geq -\gamma, \tag{7}$$

*where $\nu, \gamma > 0$ are arbitrary constants.*

**Definition 6.** *A function $f : \mathbb{R}^d \to \mathbb{R}$ is said to possess the strict saddle property at all $\boldsymbol{x} \in \mathbb{R}^d$ if $\boldsymbol{x}$ fulfills any one of the following conditions: (i) $\|\nabla f(\mathbf{x})\| \geq \nu$, (ii) $\lambda_{min}(\boldsymbol{H}(\boldsymbol{x})) \leq -\gamma$, (iii) $\mathbf{x}$ is close to a local minimum.*

The strict saddle property ensures that an iterate stuck at a saddle point has a direction of escape.

**Definition 7.** *A function $f : \mathbb{R}^d \to \mathbb{R}$ is $\alpha-$strongly convex if $\lambda_{min}(\boldsymbol{H}(\boldsymbol{x})) \geq \alpha \quad \forall \mathbf{x} \in \mathbb{R}^d$.*

We now provide the formal definitions of two common terms in time complexity.

**Definition 8.** *A function $f(s)$ is said to be $O(g(s))$ if $\exists$ a constant $c > 0$ such that $|f(s)| \leq c|g(s)|$. Here $s \in S$ which is the domain of the functions $f$ and $g$.*

**Definition 9.** *A function $f(s)$ is said to be $\Omega(g(s))$ if $\exists$ a constant $c > 0$ such that $|f(s)| \geq c|g(s)|$.*

In our analysis, we introduce the notations $\tilde{O}(.)$ and $\tilde{\Omega}(.)$ which hide all factors (including $\beta$, $\rho$, $d$, and $\alpha$) except $\eta_c$, $L_{min}$ and $L_{max}$ in $O$ and $\Omega$ respectively.

# 4 PROOF OF CONVERGENCE

We present our convergence proofs to theoretically show that the proposed PLRS method converges to a $\nu$-SOSP in finite time. We first state the assumptions that are instrumental for our proofs.

**Assumptions 1.** *We now state the assumptions regarding the function $f : \mathbb{R}^d \to \mathbb{R}$ that we require for proving the theorems.*

**A1** *The function $f$ is $\beta$-smooth.*

**A2** *The function $f$ is $\rho$-Hessian Lipschitz.*

**A3** *The norm of the stochastic gradient noise is bounded i.e, $\|g(\mathbf{x}_t) - \nabla f(\mathbf{x}_t)\| \leq Q \quad \forall t \geq 0$. Further, $\mathbb{E}[Q^2] \leq \sigma^2$.*

**A4** *The function $f$ has strict saddle property.*

**A5** *The function $f$ is bounded i.e., $|f(\mathbf{x})| \leq B$, $\forall \mathbf{x} \in \mathbb{R}^d$.*

**A6** *The function $f$ is locally $\alpha-$strongly convex i.e, in the $\delta$-neighborhood of a locally optimal point $\mathbf{x}^*$ for some $\delta > 0$.*

**Remark 2.** *If $\nabla \tilde{f}(\tilde{\mathbf{x}}_t)$ and $\tilde{g}(\tilde{\mathbf{x}}_t)$ are the gradient and stochastic gradient of the second order Taylor approximation of $f$ about the iterate $\tilde{\mathbf{x}}_t$, from Assumption A3, it is implied that $\left\|\tilde{g}(\tilde{\mathbf{x}}_t) - \nabla \tilde{f}(\tilde{\mathbf{x}}_t)\right\| \leq \tilde{Q}$. Further, $\mathbb{E}[\tilde{Q}^2] \leq \tilde{\sigma}^2$.*

Note that these assumptions are similar to those in the perturbed gradient literature (Ge et al., 2015; Jin et al., 2017; 2021). We call attention to two significant differences in our approach compared to other perturbed gradient methods such as (Jin et al., 2017; Ge et al., 2015; Jin et al., 2021): (i) In contrast to the isotropic *additive* perturbation commonly added to the SGD update, we introduce randomness in our learning rate, manifested as *multiplicative* noise in the update. This makes the characterization of the total noise dependent on the gradient, making the analysis challenging. (ii) The magnitude of noise injected is computed through the smoothness constants in the work by Jin et al. (Jin et al., 2017; 2021); instead, we treat the parameters $L_{min}$ and $L_{max}$ as hyperparameters to be tuned. This enables our PLRS method to be easily applied to training deep neural networks where the computation of these smoothness constants could be infeasible due to sheer computational complexity.

We reiterate the update equations of the proposed SGD-PLRS.

$$\mathbf{x}_{t+1} = \mathbf{x}_t - \eta_c \nabla f(\mathbf{x}_t) - \mathbf{w}_t. \tag{3}$$

$$\mathbf{w}_t = \eta_c g(\mathbf{x}_t) - \eta_c \nabla f(\mathbf{x}_t) + u_{t+1} g(\mathbf{x}_t). \tag{4}$$

Note that the term $\mathbf{w}_t$ has zero mean and we state this formally in the lemma below.

**Lemma 1** (Zero mean property)**.** *The mean of $\mathbf{w}_{t-1}$ $\forall t \geq 1$ is 0.*

*Proof.*

$$\begin{aligned}\mathbb{E}[\mathbf{w}_{t-1}] &= \mathbb{E}\left[\eta_c g(\mathbf{x}_{t-1}) - \eta_c \nabla f(\mathbf{x}_{t-1})\right] + \mathbb{E}\left[u_t g(\mathbf{x}_{t-1})\right] \\ &= \mathbf{0} \qquad \forall t \geq 1.\end{aligned} \tag{8}$$

This follows as $\mathbb{E}[u_t] = \frac{L_{\min} + L_{\max} - 2\eta_c}{2} = 0$ and $\mathbb{E}[g(\mathbf{x}_{t-1})] = \nabla f(\mathbf{x}_{t-1})$. $\qquad \square$

For a function satisfying the Assumptions **A1**-**A6**, there are three possibilities for the iterate $\mathbf{x}_t$ with respect to the function's gradient and Hessian, namely, **B1:** Gradient is large; **B2:** Gradient is small and iterate is around a saddle point; **B3:** Gradient is small and iterate is around a $\nu$-SOSP.

We now present three theorems corresponding to each of these cases. Our first result pertains to the case **B1** where the gradient of the iterate is large.

**Theorem 1.** *Under the assumptions A1 and A3 with $L_{max} < \frac{1}{\beta}$, for any point $\mathbf{x}_t$ with $\|\nabla f(\mathbf{x}_t)\| \geq \sqrt{3\eta_c \beta \sigma^2}$ where $\sqrt{3\eta_c \beta \sigma^2} < \epsilon$, after one iteration, we have*

$$\mathbb{E}[f(\mathbf{x}_{t+1})] - f(\mathbf{x}_t) \leq -\tilde{\Omega}(L_{max}^2).$$

This theorem suggests that, for any iterate $\mathbf{x}_t$ for which the gradient is large, the expected functional value of the subsequent iterate $f(\mathbf{x}_{t+1})$ decreases, and the corresponding decrease $\mathbb{E}[f(\mathbf{x}_{t+1})] - f(\mathbf{x}_t)$ is in the order of $\tilde{\Omega}(L_{max}^2)$. The formal proof for this theorem can be found in Appendix A.

The next theorem corresponds to the case **B2** where the gradient is small and the Hessian is negative.

**Theorem 2.** *Consider $f$ satisfying Assumptions A1 - A5. Let $\{\mathbf{x}_t\}$ be the SGD iterates of the function $f$ using PLRS. Let $\|\nabla f(\mathbf{x}_0)\| \leq \sqrt{3\eta_c \beta \sigma^2} < \epsilon$ and $\lambda_{min}(\mathbf{H}(\mathbf{x}_0)) \leq -\gamma$ where $\epsilon, \gamma > 0$. Then, there exists a $T = \tilde{O}\left(L_{max}^{-1/4}\right)$ such that with probability at least $1 - \tilde{O}\left(L_{max}^{7/2}\right)$,*

$$\mathbb{E}[f(\mathbf{x}_T) - f(\mathbf{x}_0)] \leq -\tilde{\Omega}\left(L_{max}^{3/4}\right).$$

The formal proof of this theorem is provided in Appendix C. The sketch of the proof is given below.

**Proof Sketch** This theorem shows that the iterates obtained using PLRS escape from a saddle point $\mathbf{x}_0$ (where the gradient is small, and the Hessian has atleast one negative eigenvalue), i.e, it shows the decrease in the expected value of the function $f$ after $T = \tilde{O}\left(L_{max}^{-1/4}\right)$ iterations. Note that for a $\rho-$Hessian smooth function,

$$f(\mathbf{x}_T) \leq f(\mathbf{x}_0) + \nabla f(\mathbf{x}_0)^T(\mathbf{x}_T - \mathbf{x}_0) + \frac{1}{2}(\mathbf{x}_T - \mathbf{x}_0)^T\mathbf{H}(\mathbf{x}_0)(\mathbf{x}_T - \mathbf{x}_0) + \frac{\rho}{6}\left\|\mathbf{x}_T - \mathbf{x}_0\right\|^3. \quad (9)$$

To evaluate $\mathbb{E}[f(\mathbf{x}_T) - f(\mathbf{x}_0)]$ from equation 9, we require an analytical expression for $\mathbf{x}_T - \mathbf{x}_0$, which is not tractable. Hence, we employ the second-order Taylor approximation of the function $f$, which we denote as $\tilde{f}$. We then apply SGD-PLRS on $\tilde{f}$ to obtain $\tilde{\mathbf{x}}_T$. Following this, we write $\mathbf{x}_T - \mathbf{x}_0 = (\mathbf{x}_T - \tilde{\mathbf{x}}_T) + (\tilde{\mathbf{x}}_T - \mathbf{x}_0)$ and derive expressions for upper bounds on $\tilde{\mathbf{x}}_T - \mathbf{x}_0$ and $\mathbf{x}_T - \tilde{\mathbf{x}}_T$ which hold with high probability in Lemmas 2 and 3, respectively (given in Appendix B.1 and B.2).

We split the quadratic term in equation 9 into two parts corresponding to $\tilde{\mathbf{x}}_T - \mathbf{x}_0$ and $\mathbf{x}_T - \tilde{\mathbf{x}}_T$. We further decompose the term, say $\mathcal{Y} = (\tilde{\mathbf{x}}_T - \mathbf{x}_0)^T\mathbf{H}(\mathbf{x}_0)(\tilde{\mathbf{x}}_T - \mathbf{x}_0)$ into its eigenvalue components along each dimension with corresponding eigenvalues $\lambda_1, \ldots, \lambda_d$ of $\mathbf{H}(\mathbf{x}_0)$. Our main result in this theorem proves that the term $\mathcal{Y}$ dominates over all the other terms of equation 9, and that it is bounded by a negative value, thereby, proving $\mathbb{E}[f(\mathbf{x}_T)] \leq f(\mathbf{x}_0)$. This main result uses a two-pronged proof. Firstly, we use our assumption that the initial iterate $\mathbf{x}_0$ is at a saddle point and hence at least one of $\lambda_i, \quad 1 \leq i \leq d$ is negative. We formally show that the eigenvector corresponding to this eigenvalue points to the direction of escape. Secondly, we use the second order statistics of our noise, to show that the magnitude of $\mathcal{Y}$ is large enough to dominate over the other terms of equation 9. Note that our noise term involves the stochasticity in the gradient and the probabilistic learning rate. Hence, we have shown that the negative eigenvalue of the Hessian at a saddle point and the unique characterization of the noise is sufficient to force a descent along the negative curvature safely out of the region of the saddle point within $T$ iterations. ∎

As each SGD-PLRS update is noisy, we need to ensure that once we escape a saddle point and move towards a local minimum (case **B3**), we do not overshoot the minimum but rather, stay in the $\delta-$neighborhood of an SOSP, with high probability. We formalize this in Theorem 3.

**Theorem 3.** *Consider $f$ satisfying the assumptions **A1-A6**. Let the initial iterate $\boldsymbol{x}_0$ be $\delta$ close to a local minimum $\boldsymbol{x}^*$ such that $\|\boldsymbol{x}_0 - \boldsymbol{x}^*\| \leq \tilde{O}(\sqrt{L_{max}}) < \delta$. With probability at least $1 - \xi$, $\forall t \leq T$ where $T = \tilde{O}\left(\frac{1}{L_{max}^2}\log\frac{1}{\xi}\right)$,*

$$\|\boldsymbol{x}_t - \boldsymbol{x}^*\| \leq \tilde{O}\left(\sqrt{L_{max}\log\frac{1}{L_{max}\xi}}\right) < \delta$$

This theorem deals with the case that the initial iterate $\mathbf{x}_0$ is $\delta$-close to a local minimum $\mathbf{x}^*$ (case **B3**). We prove that the subsequent iterates are also in the same neighbourhood, i.e., $\delta$ close to the local minimum, with high probability. In other words, we prove that the sequence $\{\|\mathbf{x}_t - \mathbf{x}^*\|\}$ is bounded by $\delta$ for $t \leq T$. In the neighbourhood of the local minimum, gradients are small and subsequently, the change in iterates, $\mathbf{x}_t - \mathbf{x}_{t-1}$ are minute. Therefore, the iterates stay near the local minimum with high probability. It is worth noting that the nature of the noise, which is comprised of stochastic gradients (whose stochasticity is bounded by $Q$) multiplied with a bounded uniform random variable (owing to PLRS), aids in proving our result. We provide the formal proof in Appendix D.

## 5 EMPIRICAL EVALUATION

We conduct extensive empirical evaluations across diverse modalities and tasks, including: (a) image classification on benchmarks such as CIFAR-10, CIFAR-100 (Krizhevsky et al., 2009), and Tiny ImageNet (Le & Yang, 2015); (b) large-scale image classification on the ImageNet-1K dataset (Russakovsky et al., 2015); (c) natural language processing tasks, comprising question answering on SQuAD v1.1 (Rajpurkar et al., 2016) and machine translation on the IWSLT'14 dataset (Cettolo et al., 2014); and (d) automatic speech recognition on the CommonVoice 11.0 (Ardila et al., 2020)

Hindi dataset. We compare with the following baseline learning rate schedulers wherever applicable: (i) cosine annealing with warm restarts (Loshchilov & Hutter, 2017b), (ii) one-cycle scheduler (Smith & Topin, 2019), (iii) knee scheduler (Iyer et al., 2023), (iv) constant learning rate and (v) multi-step decay scheduler. We choose the parameters for these baseline schedulers as suggested in the original papers (further details of parameters are provided in Appendix F).

Further, in order to compare our proposed PLRS against the noisy SGD mechanism proposed by Ge et al. (Ge et al., 2015), we provide convergence results on the online tensor decomposition problem using the code provided by the authors in Appendix I. We conduct all our experiments on one NVIDIA GeForce RTX 2080 12GB GPU card and one NVIDIA A100 30GB GPU card.

**Hyperparameter tuning** To determine the parameters $L_{min}$ and $L_{max}$ for PLRS, we perform a range test, where we observe the training loss for a range of learning rates as is done in state-of-the-art LR schedulers such as one-cycle (Smith & Topin, 2019) and knee schedulers (Iyer et al., 2023). As the learning rate is gradually increased, we first observe a steady decrease in the training loss, then followed by a drastic increase. We note the learning rate at which there is an increase of training loss, say $\bar{L}$ and choose the maximum learning rate $L_{max}$ to be just below $\bar{L}$, where the loss is still decreasing. We then tune $L_{min}$ such that $0 < L_{min} < L_{max}$. Note that there is no extra tuning cost of $L_{min}$ and $L_{max}$ in comparison to state-of-the-art deterministic LR schedulers since all LR schedulers such as cosine, knee, cyclic, require an LR range test to set the parameters. Specifically, cosine LR scheduler requires the parameters minimum learning rate, frequency of restarts and a multiplicative factor; cyclic LR scheduler requires a base learning rate, maximum learning rate, mode of operation and the number of iterations to reach the maximum learning rate; knee LR scheduler requires the peak learning rate, number of explore iterations and the number of warmup iterations. In comparison, for our proposed probabilistic learning rate scheduler, we only require $L_{min}$ and $L_{max}$.

## 5.1 RESULTS ON IMAGE CLASSIFICATION TASKS

We run experiments for 500 epochs for the CIFAR datasets, for 100 epochs for the Tiny ImageNet dataset, and for 60 epochs on the ImageNet-1K dataset using the SGD optimizer for all schedulers [2]. We also set all other regularization parameters, such as weight decay and dampening, to zero. We use a batch size of 64 for DenseNet-40-12, 50 for ResNet-50, and 128 for the others.

| Scheduler | VGG-16 | | WRN-28-10 | |
|---|---|---|---|---|
| | Max acc. | Mean acc. (S.D.) | Max acc. | Mean acc. (S.D.) |
| Cosine | 96.87 | 96.09 (0.78) | 92.03 | 91.90 (0.13) |
| Knee | 96.87 | **96.35** (0.45) | 92.04 | 91.64 (0.63) |
| One-cycle | 90.62 | 89.06 (1.56) | 87.76 | 87.37 (0.35) |
| Constant | 96.09 | 96.06 (0.05) | 92.04 | 92.00 (0.08) |
| Multi-step | 92.97 | 92.45 (0.90) | 88.94 | 88.80 (0.21) |
| PLRS (ours) | **97.66** | 96.09 (1.56) | **94.00** | **93.97** (0.07) |

Table 1: Maximum and mean (with standard deviation) test accuracies over 3 runs for CIFAR-10.

**Results on CIFAR-10** We consider VGG-16 (Simonyan & Zisserman, 2015) and WRN-28-10 (Zagoruyko & Komodakis, 2016) architectures for training CIFAR-10 and use $L_{min} = 0.07$ and $L_{max} = 0.1$; and $L_{min} = 0.09$ and $L_{max} = 0.1$ respectively. We record the maximum and mean test accuracies across different LR schedulers in Table 1. The highest accuracy across schedulers is recorded in bold. For the VGG-16 network, we rank the highest in terms of maximum test accuracy. In terms of the mean test accuracy over 3 runs, the knee scheduler outperforms the rest. Note that the second highest mean test accuracy is achieved by both PLRS and the cosine annealing schedulers. Unsurprisingly, the constant scheduler has the lowest standard deviation. In the WRN-28-10 network, PLRS ranks the highest both in terms of maximum and mean test accuracies, with a $1.96\%$ improvement over the state-of-the-art Knee and constant LR schedulers in maximum test accuracy achieved. Further, we observe from the training loss plots that PLRS achieves fast convergence when compared to other schedulers. We give the plots in Appendix G.1.

---

[2]We provide results without momentum to be consistent with our theoretical framework. When we used the SGD optimizer with momentum for PLRS, we obtain results better than those reported without momentum.

| Scheduler | ResNet-110 | | DenseNet-40-10 | |
|---|---|---|---|---|
| | Max acc. | Mean acc. (S.D.) | Max acc. | Mean acc. (S.D.) |
| Cosine | 74.22 | 72.66 (1.56) | 64.34 | 64.10 (0.28) |
| Knee | 75.78 | 72.39 (2.96) | 65.18 | 64.83 (0.30) |
| One-cycle | 71.09 | 70.05 (1.19) | 64.21 | 59.21 (4.32) |
| Constant | 69.53 | 66.67 (2.51) | 64.8 | 64.49 (0.27) |
| Multi-step | 63.28 | 61.20 (2.39) | 29.14 | 29.01 (0.17) |
| PLRS (ours) | **77.34** | **74.61** (2.95) | **65.92** | **65.57** (0.31) |

Table 2: Maximum and mean (with standard deviation) test accuracies over 3 runs for CIFAR-100.

| Tiny ImageNet | | |
|---|---|---|
| Scheduler | Max acc. | Mean acc. (S.D) |
| Cosine | 62.13 | **62.03** (0.15) |
| Knee | 61.93 | 61.50 (0.42) |
| One-cycle | 52.24 | 51.99 (0.22) |
| Constant | 61.59 | 61.11 (0.42) |
| Multi-step | 61.28 | 61.20 (0.08) |
| PLRS (ours) | **62.34** | 61.90 (0.73) |
| ImageNet-1K | | |
| Scheduler | Top-1 acc. | Top-5 acc. |
| Baseline (Knee) | 65.21 | 85.78 |
| PLRS (ours) | **68.01** | **88.08** |

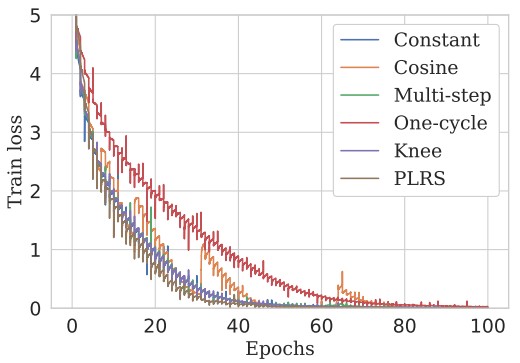

Table 3: Maximum and mean (with standard deviation) test accuracies over 3 runs for Tiny ImageNet; top-1 and top-5 accuracy for ImageNet-1K.

Figure 1: Training loss vs epochs for ResNet-50 with Tiny ImageNet.

**Results on CIFAR-100** For training CIFAR-100, we consider the networks ResNet-110 (He et al., 2016) and DenseNet-40-12 (Huang et al., 2017), and use $L_{min} = 0.07$ and $L_{max} = 0.1$ for the former, and $L_{min} = 0.1$ and $L_{max} = 0.2$ for the latter. The maximum and the mean test accuracies (with standard deviation) across 3 runs are provided in Table 2. For both ResNet-110 and DenseNet-40-12 networks, PLRS consistently outperforms all the other LR schedulers both in terms of maximum and mean test accuracies. Furthermore, from the training loss plots which are provided in Appendix G.2, PLRS converges faster than the other LR schedulers to a low train loss value. It does not have spikes (like the cosine LR scheduler), but converges in a smooth fashion to a low value.

**Results on Tiny ImageNet** We consider the Resnet-50 (He et al., 2016) architecture for training Tiny ImageNet and use $L_{min} = 0.35$ and $L_{max} = 0.4$. We present the maximum and mean test accuracies in Table 3. We provide the plot of training loss in Figure 1. PLRS performs the best in terms of maximum test accuracy. In terms of mean test accuracy, it ranks second next to cosine annealing by a close margin. It can be observed that PLRS achieves the fastest convergence to the lowest training loss compared to others. Moreover, it exhibits stable convergence, especially when compared cosine annealing, which experiences multiple spikes due to warm restarts.

**Results on ImageNet-1K** We train on the ImageNet-1K (Russakovsky et al., 2015) dataset for 60 epochs with the ResNet-50 architecture using the SGD optimizer without momentum or weight decay. With $L_{min}$ value of 0.05 and $L_{max}$ value of 0.11, and a batch size of 256, we achieve top-1 accuracy of 68.01, considerably outperforming the knee LR scheduler by 2.79% under similar settings as observed from Table 3.

**Sensitivity analysis** In order to determine how sensitive the maximum test accuracy is to the choice of $L_{min}$ and $L_{max}$, we conducted a hyper parameter sweep across a range of values for $L_{min}$ (0.01,0.03,0.05,0.07, 0.09) and $L_{max}$ (0.1, 0.2,0.3,0.4,0.5) for WRN-28-10 on the CIFAR-10 dataset, with the maximum test accuracy as the metric of interest. The average value obtained was 93.42 with a standard deviation of 0.47 and an inter-quartile range of 0.385, indicating that the val-

| Scheduler | F1 score | EM |
|---|---|---|
| Baseline | 88.66 (0.032) | 81.38 (0.02) |
| PLRS | 87.55 (0.117) | 79.775 (0.152) |

| Scheduler | BLEU | Eval ppl. |
|---|---|---|
| Baseline | 35.53 (0.06) | 4.86 (0.02) |
| PLRS | 35.37 (0.125) | **4.83** (0.02) |

Table 4: F1 score and Exact matches (EM) for SQuAD v1.1 dataset trained on BERT for 2 epochs, averaged over 3 runs.

Table 5: BLEU scores and evaluation perplexity comparison for IWSLT'14 trained on Transformer averaged over 3 runs.

ues are not spread out. Specifically, we obtain the maximum test accuracy value around 93% with multiple combinations of $(L_{min}, L_{max})$ such as $(0.01, 0.1)$, $(0.01, 0.2)$, $(0.01, 0.3)$, etc. Hence, the maximum test accuracy is relatively insensitive to $L_{min}$ and $L_{max}$ and tuning them, while recommended, may not be critical. We give detailed results of the sensitivity analysis for WRN-28-10 on CIFAR-10 as well as for DenseNet-40-12 on CIFAR-100 in Tables 6 and 7 of Appendix H.

### 5.2 RESULTS ON NLP TASKS

**Results on SQuAD v1.1** We finetune the pretrained BERT model (Devlin et al., 2019) on the SQuAD v1.1 dataset (Rajpurkar et al., 2016), which is a question-answer dataset. Using the AdamW optimizer (Loshchilov & Hutter, 2017a) with momentum parameters $\beta_1$ and $\beta_2$ set as 0.9 and 0.999 respectively, with all other parameters set as in Iyer et al. (2023), we obtain comparable values of F1-scores and exact matches (EM) to the state-of-the-art knee LR scheduler. With $L_{min}$ and $L_{max}$ values of 2e-5 and 3e-5, respectively, we give our result with baseline comparison in Table 4 after 2 epochs of training.

**Results on IWSLT'14** Experiments are conducted on the IWSLT'14 (DE-EN) dataset (Cettolo et al., 2014), which is a German to English machine translation dataset with the Transformer model (Vaswani et al., 2017). The transformer was trained with the AdamW optimizer with zero norm clipping, $\beta_1$ and $\beta_2$ values of 0.9 and 0.999 respectively, 0.3 dropout and 1e-4 weight decay for 50 epochs. With $L_{min}$ and $L_{max}$ values of 1.5e-4 and 4.5e-4, respectively, we perform competitively with the state-of-the-art knee LR scheduler as observed from Table 5.

### 5.3 RESULTS ON SPEECH RECOGNITION TASK

In order to evaluate our LR scheduler in the application of Automatic Speech Recognition, we finetune the Whisper-small (Radford et al., 2023) model on the CommonVoice 11.0 Hindi dataset (Ardila et al., 2020). We choose Hindi as it is the third most spoken language in the world [3]. The Whisper model is finetuned for a total of 5000 steps with training and evaluation batch sizes as 8, AdamW optimizer with $\beta_1$ and $\beta_2$ ad 0.9 and 0.999, and weight decay of 0.01 as per standard settings (Radford et al., 2023). We outperform the two LR schedulers with state-of-the-art results in Whisper finetuning, namely, linear decay (Radford et al., 2023) and cosine decay schedulers (Sharma et al., 2025), both starting with a base learning rate of 1e-5. We set $L_{min}$ and $L_{max}$ as 1e-6 and 1e-5 respectively. With PLRS, we obtain a word error rate (WER) of $16.10(0.0002)$, which is the mean (with standard deviation) of 3 runs, while we obtain a WER of $16.29(0.0015)$ and $16.35(0.0014)$ for the cosine and linear decay schedulers, respectively, outperforming them.

## 6 CONCLUDING REMARKS

We have proposed the novel idea of a probabilistic LR scheduler. The probabilistic nature of the scheduler helped us provide the first theoretical convergence proofs for SGD using LR schedulers. In our opinion, this is a significant step in the right direction to bridge the gap between theory and practice in the LR scheduler domain. Our empirical results show that our proposed LR scheduler performs competitively with the state-of-the-art cyclic schedulers, if not better, on a variety of image classification datasets, as well as natural language processing and speech recognition applications. This leads us to hypothesize that the proposed probabilistic LR scheduler acts as a super-class of LR schedulers encompassing both probabilistic and deterministic schedulers. Future research directions include further exploration of this hypothesis.

[3]https://www.icls.edu/blog/most-spoken-languages-in-the-world

## 7 REPRODUCIBILITY STATEMENT

For reproducibility, we provide the code as part of the supplementary material. Section 5 details the hyperparameters of our proposed learning rate scheduler, while Appendix F lists the hyperparameters used to obtain the baseline results. Additional information regarding the model architecture and training parameters is also provided in Section 5.

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

# Appendix

## A    PROOF OF THEOREM 1

**Theorem 4** (Theorem 1 restated). *Under the assumptions **A1** and **A3** with $L_{max} < \frac{1}{\beta}$, for any point $\mathbf{x}_t$ with $\|\nabla f(\mathbf{x}_t)\| \geq \sqrt{3\eta_c\beta\sigma^2}$ where $\sqrt{3\eta_c\beta\sigma^2} < \epsilon$ (satisfying **B1**), after one iteration we have,*

$$\mathbb{E}[f(\mathbf{x}_{t+1})] - f(\mathbf{x}_t) \leq -\tilde{\Omega}(L_{max}^2).$$

*Proof.* Using the second order Taylor series approximation for $f(\mathbf{x}_{t+1})$ around $\mathbf{x}_t$, where $\mathbf{x}_{t+1} = \mathbf{x}_t - \eta_c\nabla f(\mathbf{x}_t) - \mathbf{w}_t$, we have

$$f(\mathbf{x}_{t+1}) - f(\mathbf{x}_t) \leq \nabla f(\mathbf{x}_t)^T (\mathbf{x}_{t+1} - \mathbf{x}_t) + \frac{\beta}{2} \|\mathbf{x}_{t+1} - \mathbf{x}_t\|^2,$$

following the result from (Nesterov, 2014, Lemma 1.2.3). Taking expectation w.r.t. $\mathbf{w}_t$,

$$\mathbb{E}[f(\mathbf{x}_{t+1})] - f(\mathbf{x}_t) \leq \nabla f(\mathbf{x}_t)^T \mathbb{E}[\mathbf{x}_{t+1} - \mathbf{x}_t] + \frac{\beta}{2}\mathbb{E}[\|\mathbf{x}_{t+1} - \mathbf{x}_t\|^2]$$

$$= \nabla f(\mathbf{x}_t)^T\mathbb{E}[-\eta_c\nabla f(\mathbf{x}_t) - \mathbf{w}_t] + \frac{\beta}{2}\mathbb{E}[\|-\eta_c\nabla f(\mathbf{x}_t) - \mathbf{w}_t\|^2] \quad (10)$$

$$= -\eta_c \|\nabla f(\mathbf{x}_t)\|^2 + \frac{\beta}{2}\mathbb{E}[\eta_c^2 \|\nabla f(\mathbf{x}_t)\|^2 + \|\mathbf{w}_t\|^2],$$

since $\mathbb{E}[\mathbf{w}_t] = 0$ due to the zero mean property in Lemma 1. We focus on the last term in the next steps. Expanding $\|\mathbf{w}_t\|^2$,

$$\|\mathbf{w}_t\|^2 = (\eta_c g(\mathbf{x}_t) - \eta_c\nabla f(\mathbf{x}_t) + u_{t+1}g(\mathbf{x}_t))^T (\eta_c g(\mathbf{x}_t) - \eta_c\nabla f(\mathbf{x}_t) + u_{t+1}g(\mathbf{x}_t))$$

$$= \eta_c^2 \|g(\mathbf{x}_t)\|^2 - \eta_c^2 g(\mathbf{x}_t)^T\nabla f(\mathbf{x}_t) + \eta_c u_{t+1} \|g(\mathbf{x}_t)\|^2 - \eta_c^2\nabla f(\mathbf{x}_t)^T g(\mathbf{x}_t) + \eta_c^2 \|\nabla f(\mathbf{x}_t\|^2$$

$$- \eta_c u_{t+1}\nabla f(\mathbf{x}_t)^T g(\mathbf{x}_t) + \eta_c u_{t+1} \|g(\mathbf{x}_t)\|^2 - \eta_c u_{t+1}g(\mathbf{x}_t)^T\nabla f(\mathbf{x}_t) + u_{t+1}^2 \|g(\mathbf{x}_t)\|^2.$$

Taking expectation with respect to $\mathbf{x}_t$ and noting that $\mathbb{E}[u_{t+1}] = 0$ and $\mathbb{E}[g(\mathbf{x}_t)] = \nabla f(\mathbf{x}_t)$,[4]

$$\mathbb{E}[\|\mathbf{w}_t\|^2] = \eta_c^2\mathbb{E}[\|g(\mathbf{x}_t)\|^2] - \eta_c^2 \|\nabla f(\mathbf{x}_t)\|^2 + \mathbb{E}[u_{t+1}^2]\mathbb{E}[\|g(\mathbf{x}_t)\|^2]. \quad (11)$$

Now, as per assumption **A3**,

$$\|g(\mathbf{x}_t) - \nabla f(\mathbf{x}_t)\|^2 \leq Q^2$$

$$\|g(\mathbf{x}_t)\|^2 + \|\nabla f(\mathbf{x}_t)\|^2 - 2g(\mathbf{x}_t)^T\nabla f(\mathbf{x}_t) \leq Q^2$$

$$\|g(\mathbf{x}_t)\|^2 \leq Q^2 - \|\nabla f(\mathbf{x}_t)\|^2 + 2g(\mathbf{x}_t)^T\nabla f(\mathbf{x}_t)$$

$$\mathbb{E}[\|g(\mathbf{x}_t)\|^2] \leq \mathbb{E}[Q^2] - \|\nabla f(\mathbf{x}_t)\|^2 + 2 \|\nabla f(\mathbf{x}_t)\|^2 \leq \sigma^2 + \|\nabla f(\mathbf{x}_t)\|^2, \quad (12)$$

as $\mathbb{E}[Q^2] \leq \sigma^2$. Applying equation 12 to equation 11,

$$\mathbb{E}[\|\mathbf{w}_t\|^2] \leq \eta_c^2\sigma^2 + \eta_c^2 \|\nabla f(\mathbf{x}_t)\|^2 - \eta_c^2 \|\nabla f(\mathbf{x}_t)\|^2 + \mathbb{E}[u_{t+1}^2]\sigma^2 + \mathbb{E}[u_{t+1}^2] \|\nabla f(\mathbf{x}_t)\|^2$$

$$= \eta_c^2\sigma^2 + \mathbb{E}[u_{t+1}^2]\sigma^2 + \mathbb{E}[u_{t+1}^2] \|\nabla f(\mathbf{x}_t)\|^2 \quad (13)$$

$$= \eta_c^2\sigma^2 + \frac{(L_{max} - L_{min})^2\sigma^2}{12} + \frac{(L_{max} - L_{min})^2 \|\nabla f(\mathbf{x}_0)\|^2}{12},$$

---

[4]Note that there are two random variables in $\mathbf{w}_t$ which are the stochastic gradient $g(\mathbf{x}_t)$ and the uniformly distributed LR $u_{t+1}$ due to our proposed LR scheduler. Hence, the expectation is with respect to both these variables. Also note that $u_{t+1}$ and $g(\mathbf{x}_t)$ are independent of each other.

since the second moment of a uniformly distributed random variable in the interval $[L_{min} - \eta_c, L_{max} - \eta_c]$ is given by $\frac{(L_{max}-L_{min})^2}{12}$. Using equation 13 in equation 10 and $\eta_c = \frac{L_{min}+L_{max}}{2}$,

$$\mathbb{E}[f(\mathbf{x}_{t+1})] - f(\mathbf{x}_t) \leq -\eta_c \|\nabla f(\mathbf{x}_t)\|^2 + \frac{\beta}{2}\eta_c^2 \|\nabla f(\mathbf{x}_t)\|^2 + \frac{\beta\eta_c^2\sigma^2}{2} + \frac{\beta(L_{max}-L_{min})^2\sigma^2}{24}$$
$$+ \frac{\beta(L_{max}-L_{min})^2 \|\nabla f(\mathbf{x}_t)\|^2}{24}$$
$$\leq -\eta_c \|\nabla f(\mathbf{x}_t)\|^2 + \frac{\beta}{2}\eta_c^2 \|\nabla f(\mathbf{x}_t)\|^2 + \frac{\beta\eta_c^2\sigma^2}{2} + \frac{\beta\eta_c^2\sigma^2}{6} + \frac{\beta\eta_c^2 \|\nabla f(\mathbf{x}_0)\|^2}{6}$$
$$= -\|\nabla f(\mathbf{x}_t)\|^2 \left(\eta_c - \frac{2\beta\eta_c^2}{3}\right) + \frac{2\beta\eta_c^2\sigma^2}{3}$$

Now, applying our initial assumption that $\|\nabla f(\mathbf{x}_t)\| \geq \sqrt{3\eta_c\beta\sigma^2}$, we have,

$$\mathbb{E}[f(\mathbf{x}_{t+1})] - f(\mathbf{x}_t) \leq -3\eta_c\beta\sigma^2 \left(\eta_c - \frac{2\beta\eta_c^2}{3}\right) + \frac{2\beta\eta_c^2\sigma^2}{3} = -3\eta_c^2\beta\sigma^2 + \frac{6\beta^2\eta_c^3\sigma^2}{3} + \frac{2\beta\eta_c^2\sigma^2}{3}$$

Since $L_{max} < \frac{1}{\beta}$ and $\eta_c = \frac{L_{min}+L_{max}}{2}$, we have $\eta_c\beta < L_{max}\beta < 1$. Finally,

$$\mathbb{E}[f(\mathbf{x}_{t+1})] - f(\mathbf{x}_t) \leq -3\eta_c^2\beta\sigma^2 + \frac{6\beta\eta_c^2\sigma^2}{3} + \frac{2\beta\eta_c^2\sigma^2}{3} = -\frac{\beta\eta_c^2\sigma^2}{3}$$
$$= -\tilde{\Omega}(\eta_c^2),$$

which proves the theorem. □

## B ADDITIONAL RESULTS NEEDED TO PROVE THEOREM 2

Here, we state and prove two lemmas that are instrumental in the proof of Theorem 2.

### B.1 PROOF OF LEMMA 2

In the following Lemma, we prove that the gradients of a second order approximation of $f$ are probabilistically bounded for all $t \leq T$ and its iterates as we apply SGD-PLRS are also bounded when the initial iterate $\mathbf{x}_0$ is a saddle point.

**Lemma 2.** *Let $f$ satisfy Assumptions A1 - A4. Let $\tilde{f}$ be the second order Taylor approximation of $f$ and let $\tilde{\mathbf{x}}_t$ be the iterate at time step $t$ obtained using the SGD update equation as in equation 3 on $\tilde{f}$; let $\tilde{\mathbf{x}}_0 = \mathbf{x}_0$, $\|\nabla f(\mathbf{x}_0)\| \leq \epsilon$ and the minimum eigenvalue of the Hessian of $f$ at $\mathbf{x}_0$ be $\lambda_{min}(\mathbf{H}(\mathbf{x}_0)) = -\gamma_o$ where $\gamma_o > 0$. With probability at least $1 - \tilde{O}(L_{max}^{15/4})$, we have*

$$\left\|\nabla\tilde{f}(\tilde{\mathbf{x}}_t)\right\| \leq \tilde{O}\left(\frac{1}{L_{max}^{0.5}}\right), \quad \|\tilde{\mathbf{x}}_t - \mathbf{x}_0\| \leq \tilde{O}\left(L_{max}^{3/8}\log\left(\frac{1}{L_{max}}\right)\right) \quad \forall t \leq T = \tilde{O}\left(L_{max}^{-1/4}\right).$$

*Proof.* As $\tilde{f}$ is the second order Taylor series approximation of $f$, we have

$$\tilde{f}(\tilde{\mathbf{x}}) = f(\mathbf{x}_0) + \nabla f(\mathbf{x}_0)^T(\tilde{\mathbf{x}} - \mathbf{x}_0) + \frac{1}{2}(\tilde{\mathbf{x}} - \mathbf{x}_0)^T\mathbf{H}(\mathbf{x}_0)(\tilde{\mathbf{x}} - \mathbf{x}_0).$$

Taking derivative w.r.t. $\tilde{\mathbf{x}}$, we have $\nabla\tilde{f}(\tilde{\mathbf{x}}) = \nabla f(\mathbf{x}_0) + \mathbf{H}(\mathbf{x}_0)(\tilde{\mathbf{x}} - \mathbf{x}_0)$. Now, note that $\nabla\tilde{f}(\tilde{\mathbf{x}}_{t-1}) = \nabla f(\mathbf{x}_0) + \mathbf{H}(\mathbf{x}_0)(\tilde{\mathbf{x}}_{t-1} - \mathbf{x}_0) = K(\mathbf{x}_0) + \mathbf{H}(\mathbf{x}_0)\tilde{\mathbf{x}}_{t-1}$, where $K(\mathbf{x}_0) = \nabla f(\mathbf{x}_0) - \mathbf{H}(\mathbf{x}_0)\mathbf{x}_0 = \nabla\tilde{f}(\tilde{\mathbf{x}}_{t-1}) - \mathbf{H}(\mathbf{x}_0)\tilde{\mathbf{x}}_{t-1}$. Therefore,

$$\nabla\tilde{f}(\tilde{\mathbf{x}}_t) = K(\mathbf{x}_0) + \mathbf{H}(\mathbf{x}_0)\tilde{\mathbf{x}}_t = \nabla\tilde{f}(\tilde{\mathbf{x}}_{t-1}) - \mathbf{H}(\mathbf{x}_0)\tilde{\mathbf{x}}_{t-1} + \mathbf{H}(\mathbf{x}_0)\tilde{\mathbf{x}}_t$$
$$= \nabla\tilde{f}(\tilde{\mathbf{x}}_{t-1}) + \mathbf{H}(\mathbf{x}_0)(\tilde{\mathbf{x}}_t - \tilde{\mathbf{x}}_{t-1}). \tag{14}$$

Next, using the SGD-PLRS update and rearranging,

$$\nabla\tilde{f}(\tilde{\mathbf{x}}_t) = \nabla\tilde{f}(\tilde{\mathbf{x}}_{t-1}) - \mathbf{H}(\mathbf{x}_0)(\eta_c\nabla\tilde{f}(\tilde{\mathbf{x}}_{t-1}) + \tilde{\mathbf{w}}_{t-1})$$
$$= (I - \eta_c\mathbf{H}(\mathbf{x}_0))\nabla\tilde{f}(\tilde{\mathbf{x}}_{t-1}) - \mathbf{H}(\mathbf{x}_0)\tilde{\mathbf{w}}_{t-1}, \tag{15}$$

where $I$ denotes the $d \times d$ identity matrix. Next, unrolling the term $\nabla \tilde{f}(\tilde{\mathbf{x}}_{t-1})$ recursively,

$$\nabla \tilde{f}(\tilde{\mathbf{x}}_t) = (I - \eta_c \mathbf{H}(\mathbf{x}_0))^t \nabla \tilde{f}(\tilde{\mathbf{x}}_0) - \mathbf{H}(\mathbf{x}_0) \sum_{\tau=0}^{t-1} (I - \eta_c \mathbf{H}(\mathbf{x}_0))^{t-\tau-1} \tilde{\mathbf{w}}_\tau. \tag{16}$$

Using the triangle and Cauchy-Schwartz inequalities,

$$\left\| \nabla \tilde{f}(\tilde{\mathbf{x}}_t) \right\| \leq \left\| (I - \eta_c \mathbf{H}(\mathbf{x}_0))^t \nabla \tilde{f}(\tilde{\mathbf{x}}_0) \right\| + \left\| \mathbf{H}(\mathbf{x}_0) \sum_{\tau=0}^{t-1} (I - \eta_c \mathbf{H}(\mathbf{x}_0))^{t-\tau-1} \tilde{\mathbf{w}}_\tau \right\|$$

$$\leq \left\| (I - \eta_c \mathbf{H}(\mathbf{x}_0))^t \right\| \left\| \nabla \tilde{f}(\tilde{\mathbf{x}}_0) \right\| + \| \mathbf{H}(\mathbf{x}_0) \| \left\| \sum_{\tau=0}^{t-1} (I - \eta_c \mathbf{H}(\mathbf{x}_0))^{t-\tau-1} \tilde{\mathbf{w}}_\tau \right\| \tag{17}$$

Note that the norm over the matrices refers to the matrix-induced norm. Since $\mathbf{H}(\mathbf{x}_0)$ is a real symmetric matrix, the induced norm gives the maximum eigenvalue of $\mathbf{H}(\mathbf{x}_0)$ i.e, $\lambda_{max}(\mathbf{H}(\mathbf{x}_0)) \leq \beta$ by our $\beta$-smoothness assumption **A1**. In the case of $(I - \eta_c \mathbf{H}(\mathbf{x}_0))$ the induced norm gives $(1 - \eta_c \lambda_{min}(\mathbf{H}(\mathbf{x}_0)))$ which is $(1 + \eta_c \gamma_o)$ as per our assumption that $\lambda_{min}(\mathbf{H}(\mathbf{x}_0)) = -\gamma_o$. Also recall that $\left\| \nabla \tilde{f}(\tilde{\mathbf{x}}_0) \right\| \leq \epsilon$. Now equation 17 becomes,

$$\left\| \nabla \tilde{f}(\tilde{\mathbf{x}}_t) \right\| \leq (1 + \eta_c \gamma_o)^t \epsilon + \beta \left\| \sum_{\tau=0}^{t-1} (I - \eta_c \mathbf{H}(\mathbf{x}_0))^{t-\tau-1} \tilde{\mathbf{w}}_\tau \right\|,$$

$$\leq (1 + \eta_c \gamma_o)^t \epsilon + \beta \sum_{\tau=0}^{t-1} (1 + \eta_c \gamma_o)^{t-\tau-1} \| \tilde{\mathbf{w}}_\tau \| . \tag{18}$$

Now, expanding the noise term $\tilde{\mathbf{w}}_\tau$,

$$\left\| \nabla \tilde{f}(\tilde{\mathbf{x}}_t) \right\| = (1 + \eta_c \gamma_o)^t \epsilon + \beta \sum_{\tau=0}^{t-1} (1 + \eta_c \gamma_o)^{t-\tau-1} \left\| \eta_c \tilde{g}(\tilde{\mathbf{x}}_\tau) - \eta_c \nabla \tilde{f}(\tilde{\mathbf{x}}_\tau) + u_{\tau+1} \tilde{g}(\tilde{\mathbf{x}}_\tau) \right\|$$

Now recall from our assumption **A3** that $\left\| \tilde{g}(\tilde{\mathbf{x}}_\tau) - \nabla \tilde{f}(\tilde{\mathbf{x}}_\tau) \right\| \leq \tilde{Q}$. Hence,

$$\left\| \nabla \tilde{f}(\tilde{\mathbf{x}}_t) \right\| \leq (1 + \eta_c \gamma_o)^t \epsilon + \beta \sum_{\tau=0}^{t-1} (1 + \eta_c \gamma_o)^{t-\tau-1} \left( \eta_c \tilde{Q} + |u_{\tau+1}| \left\| \tilde{g}(\tilde{\mathbf{x}}_\tau) - \nabla \tilde{f}(\tilde{\mathbf{x}}_\tau) + \nabla \tilde{f}(\tilde{\mathbf{x}}_\tau) \right\| \right)$$

$$\leq (1 + \eta_c \gamma_o)^t \epsilon + \beta \sum_{\tau=0}^{t-1} (1 + \eta_c \gamma_o)^{t-\tau-1} \left( \eta_c \tilde{Q} + |u_{\tau+1}| \left( \tilde{Q} + \left\| \nabla \tilde{f}(\tilde{\mathbf{x}}_\tau) \right\| \right) \right)$$

Using $\left\| \nabla \tilde{f}(\tilde{\mathbf{x}}_0) \right\| \leq \epsilon$ and $\left\| \nabla \tilde{f}(\tilde{\mathbf{x}}_1) \right\| \leq (1 + \eta_c \gamma_o)\epsilon + \epsilon + 2\tilde{Q}$, it can be proved by induction that the general expression for $t \geq 2$ is given by,

$$\left\| \nabla \tilde{f}(\tilde{\mathbf{x}}_t) \right\| \leq 10\tilde{Q} \sum_{\tau=0}^{\frac{t(t-1)}{2}} (1 + \eta_c \gamma_o)^\tau \tag{19}$$

We give the proof of equation 19 by induction in Appendix E. Next, we prove the bound on $\tilde{\mathbf{x}}_t - \tilde{\mathbf{x}}_0$. Using the SGD-PLRS update,

$$\tilde{\mathbf{x}}_t - \tilde{\mathbf{x}}_0 = -\sum_{\tau=0}^{t-1} \left( \eta_c \nabla \tilde{f}(\tilde{\mathbf{x}}_\tau) + \tilde{\mathbf{w}}_\tau \right)$$

$$= -\sum_{\tau=0}^{t-1} \left( \eta_c \left( (I - \eta_c \mathbf{H}(\mathbf{x}_0))^\tau \nabla \tilde{f}(\tilde{\mathbf{x}}_0) - \mathbf{H}(\mathbf{x}_0) \sum_{\tau'=0}^{\tau-1} (I - \eta_c \mathbf{H}(\mathbf{x}_0))^{\tau-\tau'-1} \tilde{\mathbf{w}}_{\tau'} \right) + \tilde{\mathbf{w}}_\tau \right) \tag{20a}$$

$$= -\sum_{\tau=0}^{t-1} \eta_c (I - \eta_c \mathbf{H}(\mathbf{x}_0))^\tau \nabla f(\mathbf{x}_0) - \sum_{\tau=0}^{t-1} (I - \eta_c \mathbf{H}(\mathbf{x}_0))^{t-\tau-1} \tilde{\mathbf{w}}_\tau, \tag{20b}$$

where the equation equation 20a is obtained by using equation 16. We obtain equation 20b by using the summation of geometric series as $\mathbf{H}(\mathbf{x}_0)$ is invertible by the strict saddle property. As $\tilde{\mathbf{x}}_0 = \mathbf{x}_0$, we can write $\nabla \tilde{f}(\tilde{\mathbf{x}}_0) = \nabla f(\mathbf{x}_0)$. Taking norm,

$$
\|\tilde{\mathbf{x}}_t - \tilde{\mathbf{x}}_0\| \leq \left\| \sum_{\tau=0}^{t-1} \eta_c (I - \eta_c \mathbf{H}(\mathbf{x}_0))^\tau \nabla f(\mathbf{x}_0) \right\| + \left\| \sum_{\tau=0}^{t-1} (I - \eta_c \mathbf{H}(\mathbf{x}_0))^{t-\tau-1} \tilde{\mathbf{w}}_\tau \right\|
$$

$$
\leq \sum_{\tau=0}^{t-1} \| \eta_c (I - \eta_c \mathbf{H}(\mathbf{x}_0))^\tau \nabla f(\mathbf{x}_0) \| + \sum_{\tau=0}^{t-1} \left\| (I - \eta_c \mathbf{H}(\mathbf{x}_0))^{t-\tau-1} \tilde{\mathbf{w}}_\tau \right\| \quad (21)
$$

$$
\leq \eta_c \epsilon \sum_{\tau=0}^{t-1} (1 + \eta_c \gamma_o)^\tau + \sum_{\tau=0}^{t-1} (1 + \eta_c \gamma_o)^{t-\tau-1} \| \tilde{\mathbf{w}}_\tau \|.
$$

In equation 21, it can be seen that the first term is arbitrarily small by the initial assumption and that the second term decides the order of $\|\tilde{\mathbf{x}}_t - \tilde{\mathbf{x}}_0\|$. Hence, in order to bound $\|\tilde{\mathbf{x}}_t - \tilde{\mathbf{x}}_0\|$ probabilistically, it is sufficient to bound the second term, $\sum_{\tau=0}^{t-1} (1 + \eta_c \gamma_o)^{t-\tau-1} \| \tilde{\mathbf{w}}_\tau \|$. Now,

$$
\sum_{\tau=0}^{t-1} (1 + \eta_c \gamma_o)^{t-\tau-1} \| \tilde{\mathbf{w}}_\tau \| = \sum_{\tau=0}^{t-1} (1 + \eta_c \gamma_o)^{t-\tau-1} \left\| \eta_c \tilde{g}(\tilde{\mathbf{x}}_\tau) - \eta_c \nabla \tilde{f}(\tilde{\mathbf{x}}_\tau) + u_{\tau+1} \tilde{g}(\tilde{\mathbf{x}}_\tau) \right\|
$$

$$
= \sum_{\tau=0}^{t-1} (1 + \eta_c \gamma_o)^{t-\tau-1} \left( \eta_c \tilde{Q} + |u_{\tau+1}| \left\| \tilde{g}(\tilde{\mathbf{x}}_\tau) - \nabla \tilde{f}(\tilde{\mathbf{x}}_\tau) + \nabla \tilde{f}(\tilde{\mathbf{x}}_\tau) \right\| \right)
$$

$$
= \sum_{\tau=0}^{t-1} (1 + \eta_c \gamma_o)^{t-\tau-1} \tilde{Q} (\eta_c + |u_{\tau+1}|) + \sum_{\tau=0}^{t-1} (1 + \eta_c \gamma_o)^{t-\tau-1} |u_{\tau+1}| \left\| \nabla \tilde{f}(\tilde{\mathbf{x}}_\tau) \right\|
$$

Now, using $\left\| \nabla \tilde{f}(\tilde{\mathbf{x}}_0) \right\| \leq \epsilon$, $\left\| \nabla \tilde{f}(\tilde{\mathbf{x}}_1) \right\| \leq (1 + \eta_c \gamma_o)\epsilon + \epsilon + 2\tilde{Q}$ and equation 19 we write,

$$
\sum_{\tau=0}^{t-1} (1 + \eta_c \gamma_o)^{t-\tau-1} \| \tilde{\mathbf{w}}_\tau \| \leq \sum_{\tau=0}^{t-1} (1 + \eta_c \gamma_o)^{t-\tau-1} \tilde{Q} (\eta_c + |u_{\tau+1}|) + (1 + \eta_c \gamma_o)^{t-1} |u_1| \epsilon +
$$

$$
(1 + \eta_c \gamma_o)^{t-2} |u_2| \left( (1 + \eta_c \gamma_o)\epsilon + \epsilon + 2\tilde{Q} \right) + \sum_{\tau=2}^{t-1} (1 + \eta_c \gamma_o)^{t-\tau-1} |u_{\tau+1}| 10\tilde{Q} \sum_{\tau'=0}^{\frac{\tau(\tau-1)}{2}} (1 + \eta_c \gamma_o)^{\tau'}
$$

$$(22)$$

It can be observed from equation 22 that the last term dominates the expression of and hence, it determines the order of $\|\tilde{\mathbf{x}}_t - \tilde{\mathbf{x}}_0\|$. We now apply Hoeffding's inequality to derive a probabilistic bound on $\|\tilde{\mathbf{x}}_t - \tilde{\mathbf{x}}_0\|$. According to Hoeffding's inequality for any summation $S_n = X_1 + \cdots + X_n$ such that $a_i \leq X_i \leq b_i$, $\mathbb{P}(S_n - \mathbb{E}[S_n] \geq \delta) \leq \exp\left( \frac{-2\delta^2}{\sum_{i=1}^n (b_i - a_i)^2} \right)$. Now, setting $T = \tilde{O}\left( L_{max}^{-1/4} \right)$ from equation 41 and assuming $\eta_c \leq \eta_{max} \leq \frac{\sqrt{2}-1}{\gamma'}, \gamma_o \leq \gamma'$, the squared bound of the summation $\sum_{\tau=2}^{t-1} (1 + \eta_c \gamma_o)^{t-\tau-1} |u_{\tau+1}| 10\tilde{Q} \sum_{\tau'=0}^{\frac{\tau(\tau-1)}{2}} (1 + \eta_c \gamma_o)^{\tau'} \leq \tilde{O}\left( L_{max}^{3/4} \right)$, Setting $\delta = \tilde{O}\left( \sqrt{L_{max}^{3/4}} \log\left( \frac{1}{L_{max}} \right) \right)$, for some $t \leq T$,

$$
\mathbb{P}\left( \sum_{\tau=2}^{t-1} (1 + \eta_c \gamma_o)^{t-\tau-1} |u_{\tau+1}| 10\tilde{Q} \sum_{\tau'=0}^{\frac{\tau(\tau-1)}{2}} (1 + \eta_c \gamma_o)^{\tau'} \geq \tilde{O}\left( L_{max}^{3/8} \log\left( \frac{1}{L_{max}} \right) \right) \right)
$$

$$
\leq \tilde{O}(L_{max}^4).
$$

Taking the union bound over all $t \leq T$,

$$
\mathbb{P}\left( \forall t \leq T, \quad \sum_{\tau=2}^{t-1} (1 + \eta_c \gamma_o)^{t-\tau-1} |u_{\tau+1}| 10\tilde{Q} \sum_{\tau'=0}^{\frac{\tau(\tau-1)}{2}} (1 + \eta_c \gamma_o)^{\tau'} \geq \tilde{O}\left( L_{max}^{3/8} \log\left( \frac{1}{L_{max}} \right) \right) \right)
$$

$$
\leq \tilde{O}\left( L_{max}^{15/4} \right),
$$

which completes our proof. $\qquad \square$

## B.2 PROOF OF LEMMA 3

This lemma is used to derive an expression for a high probability upper bound of $\|\mathbf{x}_t - \tilde{\mathbf{x}}_t\|$ and $\left\|\nabla f(\mathbf{x}_t) - \nabla \tilde{f}(\tilde{\mathbf{x}}_t)\right\|$.

**Lemma 3.** *Let $f : \mathbb{R}^d \to \mathbb{R}$ satisfy Assumptions A1 - A4. Let $\tilde{f}$ be the second order Taylor's approximation of $f$ and let $\boldsymbol{x}_t, \tilde{\boldsymbol{x}}_t$ be the iterates at time step $t$ obtained using the SGD-PLRS update on $f$, $\tilde{f}$ respectively; let $\tilde{\boldsymbol{x}}_0 = \boldsymbol{x}_0$ and $\|\nabla f(\boldsymbol{x}_0)\| \leq \epsilon$. Let the minimum eigenvalue of the Hessian at $\boldsymbol{x}_0$ be $\lambda_{min}(\nabla^2(f(\boldsymbol{x}_0))) = -\gamma_o$, where $\gamma_o > 0$. Then $\forall t \leq T = O\left(L_{max}^{-1/4}\right)$, with a probability of at least $1 - \tilde{O}(L_{max}^{7/2})$,*

$$\|\boldsymbol{x}_t - \tilde{\boldsymbol{x}}_t\| \leq O\left(L_{max}^{3/4}\right) \quad \text{and} \quad \left\|\nabla f(\boldsymbol{x}_t) - \nabla \tilde{f}(\tilde{\boldsymbol{x}}_t)\right\| \leq O\left(L_{max}^{3/8} \log \frac{1}{L_{max}}\right).$$

*Proof.* The expression for $\mathbf{x}_t - \tilde{\mathbf{x}}_t$ can be written as,

$$\mathbf{x}_t - \tilde{\mathbf{x}}_t = (\mathbf{x}_t - \mathbf{x}_0) - (\tilde{\mathbf{x}}_t - \mathbf{x}_0)$$

$$= -\sum_{\tau=0}^{t-1} \left(\eta_c \nabla f(\mathbf{x}_\tau) + \mathbf{w}_\tau\right) - \left(-\sum_{\tau=0}^{t-1} \left(\eta_c \nabla \tilde{f}(\tilde{\mathbf{x}}_\tau) + \tilde{\mathbf{w}}_\tau\right)\right) = -\sum_{\tau=0}^{t-1} \left(\eta_c \Delta_\tau + (\mathbf{w}_\tau - \tilde{\mathbf{w}}_\tau)\right). \tag{23}$$

where we define $\Delta_t = \nabla f(\mathbf{x}_t) - \nabla \tilde{f}(\tilde{\mathbf{x}}_t)$. Now in order to bound $\|\mathbf{x}_t - \tilde{\mathbf{x}}_t\|$, we derive expressions for both $\mathbf{w}_\tau - \tilde{\mathbf{w}}_\tau$ and $\Delta_\tau$. We initially focus on the term $\mathbf{w}_\tau - \tilde{\mathbf{w}}_\tau$.

$$\mathbf{w}_\tau - \tilde{\mathbf{w}}_\tau = \eta_c g(\mathbf{x}_\tau) - \eta_c \nabla f_\tau + u_{\tau+1} g(\mathbf{x}_\tau) - \left(\eta_c \tilde{g}(\tilde{\mathbf{x}}_\tau) - \eta_c \nabla \tilde{f}(\tilde{\mathbf{x}}_\tau) + u_{\tau+1} \tilde{g}(\tilde{\mathbf{x}}_\tau)\right)$$

$$= (u_{\tau+1} + \eta_c) \left(\left(g(\mathbf{x}_\tau) - \nabla f(\mathbf{x}_\tau)\right) - \left(\tilde{g}(\tilde{\mathbf{x}}_\tau) - \nabla \tilde{f}(\tilde{\mathbf{x}}_\tau)\right)\right) + u_{\tau+1} \Delta_\tau. \tag{24}$$

Taking norm on both sides,

$$\|\mathbf{w}_\tau - \tilde{\mathbf{w}}_\tau\| \leq |u_{\tau+1} + \eta_c| \left(Q + \tilde{Q}\right) + |u_{\tau+1}| \|\Delta_\tau\| \tag{25}$$

Using equation 24 and equation 25 in equation 23, and assumption **A3** that stochastic noise is bounded, and applying norm,

$$\|\mathbf{x}_t - \tilde{\mathbf{x}}_t\| = \left\|-\sum_{\tau=0}^{t-1} \left(\eta_c \Delta_\tau + (\mathbf{w}_\tau - \tilde{\mathbf{w}}_\tau)\right)\right\| \leq \sum_{\tau=0}^{t-1} \|\eta_c \Delta_\tau + (\mathbf{w}_\tau - \tilde{\mathbf{w}}_\tau)\|$$

$$\leq \sum_{\tau=0}^{t-1} (\eta_c + |u_{\tau+1}|) \left(\|\Delta_\tau\| + Q + \tilde{Q}\right) \tag{26}$$

Next, we focus on providing a bound for $\|\Delta_t\|$. Recall that $\Delta_t = \nabla f(\mathbf{x}_t) - \nabla \tilde{f}(\tilde{\mathbf{x}}_t)$. The gradient can be written as (Nesterov, 2014),

$$\nabla f(\mathbf{x}_t) = \nabla f(\mathbf{x}_{t-1}) + (\mathbf{x}_t - \mathbf{x}_{t-1}) \left(\int_0^1 \mathbf{H}(\mathbf{x}_{t-1} + v(\mathbf{x}_t - \mathbf{x}_{t-1})) dv\right)$$

$$= \nabla f(\mathbf{x}_{t-1}) + (\mathbf{x}_t - \mathbf{x}_{t-1}) \left(\int_0^1 \left(\mathbf{H}(\mathbf{x}_{t-1} + v(\mathbf{x}_t - \mathbf{x}_{t-1})) + \mathbf{H}(\mathbf{x}_{t-1}) - \mathbf{H}(\mathbf{x}_{t-1})\right) dv\right)$$

$$= \nabla f(\mathbf{x}_{t-1}) + \mathbf{H}(\mathbf{x}_{t-1})(\mathbf{x}_t - \mathbf{x}_{t-1}) + \theta_{t-1},$$

where $\theta_{t-1} = \left(\int_0^1 \left(\mathbf{H}(\mathbf{x}_{t-1} + v(\mathbf{x}_t - \mathbf{x}_{t-1})) - \mathbf{H}(\mathbf{x}_{t-1})\right) dv\right) (\mathbf{x}_t - \mathbf{x}_{t-1})$. Let $H_{t-1}^{'} = \mathbf{H}(\mathbf{x}_{t-1}) - \mathbf{H}(\mathbf{x}_0)$. Using the SGD-PLRS update,

$$\nabla f(\mathbf{x}_t) = \nabla f(\mathbf{x}_{t-1}) - (H_{t-1}^{'} + \mathbf{H}(\mathbf{x}_0))(\eta_c \nabla f(\mathbf{x}_{t-1}) + \mathbf{w}_{t-1}) + \theta_{t-1}$$

$$= \nabla f(\mathbf{x}_{t-1})(I - \eta_c \mathbf{H}(\mathbf{x}_0)) - \mathbf{H}(\mathbf{x}_0)\mathbf{w}_{t-1} - \eta_c H_{t-1}^{'} \nabla f(\mathbf{x}_{t-1}) - H_{t-1}^{'} \mathbf{w}_{t-1} + \theta_{t-1}, \tag{27}$$

From equation 14 in the proof of Lemma 2,

$$\nabla \tilde{f}(\tilde{\mathbf{x}}_t) = \nabla \tilde{f}(\tilde{\mathbf{x}}_{t-1}) + \mathbf{H}(\mathbf{x}_0)(\tilde{\mathbf{x}}_t - \tilde{\mathbf{x}}_{t-1}). \tag{28}$$

Subtracting equation 28 from equation 27, we obtain $\Delta_t$ as,

$$\Delta_t = \nabla f(\mathbf{x}_{t-1})(I - \eta_c \mathbf{H}(\mathbf{x}_0)) - \mathbf{H}(\mathbf{x}_0)\mathbf{w}_{t-1} - \eta_c H'_{t-1} \nabla f(\mathbf{x}_{t-1}) - H'_{t-1}\mathbf{w}_{t-1} + \theta_{t-1}$$
$$- \nabla \tilde{f}(\tilde{\mathbf{x}}_t) - \mathbf{H}(\mathbf{x}_0)(\tilde{\mathbf{x}}_t - \tilde{\mathbf{x}}_{t-1})$$
$$= (I - \eta_c \mathbf{H}(\mathbf{x}_0))\Delta_{t-1} - \mathbf{H}(\mathbf{x}_0)\left(\mathbf{w}_{t-1} - \tilde{\mathbf{w}}_{t-1}\right) - H'_{t-1}\left(\eta_c \Delta_{t-1} + \eta_c \nabla \tilde{f}(\tilde{\mathbf{x}}_{t-1})\right)$$
$$- H'_{t-1}\mathbf{w}_{t-1} + \theta_{t-1}, \tag{29}$$

We now have an expression for $\Delta_t$. However, the derived expression is recursive and contains $\Delta_{t-1}$. We focus on eliminating the recursive dependence and obtain a stand-alone bound for $\|\Delta_t\| \; \forall t \leq T$. Now, we bound each of the five terms (we term them $T_1, \cdots, T_5$) of equation 29. First, let us define the events,

$$R_t = \left\{ \forall \tau \leq t, \quad \left\|\nabla \tilde{f}(\tilde{\mathbf{x}}_\tau)\right\| \leq \tilde{O}\left(\frac{1}{\sqrt{L_{max}}}\right), \quad \|\tilde{\mathbf{x}}_\tau - \mathbf{x}_0\| \leq \tilde{O}\left(L_{max}^{3/8} \log\left(\frac{1}{L_{max}}\right)\right) \right\}$$

$$C_t = \left\{ \forall \tau \leq t, \quad \|\Delta_\tau\| \leq \mu L_{max}^{3/8} \log\left(\frac{1}{L_{max}}\right) \right\}.$$

It can be seen that $R_t \subset R_{t-1}$ and $C_t \subset C_{t-1}$. Note that, from Lemma 2, we know the probabilistic characterization of $R_t$. We comment on the parameter $\mu$ later in the proof. Now, we derive bounds for each term of $\Delta_t$ *conditioned* on the event $R_{t-1} \cap C_{t-1}$ for time $t \leq T = O\left(L_{max}^{-1/4}\right)$.

$$T_1: \quad \|(I - \eta_c \mathbf{H}(\mathbf{x}_0))\Delta_{t-1}\| \leq \|\Delta_{t-1}\| + \|-\eta_c \mathbf{H}(\mathbf{x}_0)\Delta_{t-1}\|$$
$$\leq \mu L_{max}^{3/8} \log\left(\frac{1}{L_{max}}\right) + \tilde{O}\left(\mu L_{max}^{11/8} \log\left(\frac{1}{L_{max}}\right)\right) \tag{30}$$
$$= \tilde{O}\left(\mu L_{max}^{3/8} \log\left(\frac{1}{L_{max}}\right)\right),$$

where equation 30 follows from the definition of event $C_{t-1}$. Note that the first term in equation 30 governs the order of the expression (as $0 \leq L_{max} \leq 1$).

$$T_2: \quad \|\mathbf{H}(\mathbf{x}_0)\left(\mathbf{w}_{t-1} - \tilde{\mathbf{w}}_{t-1}\right)\| \leq \|\mathbf{H}(\mathbf{x}_0)\| \|\mathbf{w}_{t-1} - \tilde{\mathbf{w}}_{t-1}\|$$
$$\leq \|\mathbf{H}(\mathbf{x}_0)\| \left( |u_{\tau+1} + \eta_c| \left(Q + \tilde{Q}\right) + |u_{\tau+1}| \|\Delta_\tau\| \right)$$
$$\leq \tilde{O}(L_{max}) + \tilde{O}\left(\mu L_{max}^{11/8} \log\left(\frac{1}{L_{max}}\right)\right) = \tilde{O}(L_{max}),$$

where the substitution follows from equation 25. To bound $T_3$ and $T_4$, we first bound $H'_{t-1}$,

$$\left\|H'_{t-1}\right\| = \|\mathbf{H}(\mathbf{x}_{t-1}) - \mathbf{H}(\mathbf{x}_0)\| \leq \rho \|\mathbf{x}_{t-1} - \mathbf{x}_0\| \tag{31a}$$
$$\leq \rho \left(\|\mathbf{x}_{t-1} - \tilde{\mathbf{x}}_{t-1}\| + \|\tilde{\mathbf{x}}_{t-1} - \mathbf{x}_0\|\right)$$
$$\leq \rho \left(\sum_{\tau=0}^{t-1} (\eta_c + |u_{\tau+1}|)\left(\|\Delta_\tau\| + Q + \tilde{Q}\right)\right) + \rho \tilde{O}\left(L_{max}^{3/8} \log \frac{1}{L_{max}}\right) \tag{31b}$$
$$= \tilde{O}\left(\frac{1}{L_{max}^{1/4}}\right)\tilde{O}\left(\mu L_{max}^{11/8} \log \frac{1}{L_{max}}\right) + \tilde{O}\left(\frac{1}{L_{max}^{1/4}}\right)\tilde{O}(L_{max}) + \tilde{O}\left(L_{max}^{3/8} \log \frac{1}{L_{max}}\right) \tag{31c}$$
$$\leq \tilde{O}(L_{max}^{3/4}) + \tilde{O}\left(L_{max}^{3/8} \log \frac{1}{L_{max}}\right) \leq \tilde{O}\left(L_{max}^{3/8} \log \frac{1}{L_{max}}\right), \tag{31d}$$

where equation 31a follows from the assumption **A2** while equation 31b follows from equation 26. We use the bounds defined for events $R_{t-1} \cap C_{t-1}$ in equation 31b and equation 31c. Now, using

the bound for $\left\|H'_{t-1}\right\|$, $T_3$ can be bounded as follows.

$$T_3: \quad \left\|H'_{t-1}\eta_c(\Delta_{t-1}+\nabla\tilde{f}(\tilde{\mathbf{x}}_{t-1}))\right\| \leq \eta_c\left\|H'_{t-1}\Delta_{t-1}\right\| + \eta_c\left\|H'_{t-1}\nabla\tilde{f}(\tilde{\mathbf{x}}_{t-1})\right\|$$

$$\leq O(L_{max})\tilde{O}\left(L_{max}^{3/8}\log\frac{1}{L_{max}}\right)\mu L_{max}^{3/8}\log\frac{1}{L_{max}}$$

$$+ O(L_{max})\tilde{O}\left(L_{max}^{3/8}\log\frac{1}{L_{max}}\right)\tilde{O}\left(\frac{1}{\sqrt{L_{max}}}\right)$$

$$= \tilde{O}\left(L_{max}^{7/8}\log\frac{1}{L_{max}}\right),$$

where we use the bounds in the event $R_{t-1}\cap C_{t-1}$ and equation 31d.

$$T_4: \quad \left\|H'_{t-1}\mathbf{w}_{t-1}\right\| \leq \left\|H'_{t-1}\right\|\|\mathbf{w}_{t-1}\| = \left\|H'_{t-1}\right\|\|\eta_c g(\mathbf{x}_{t-1}) - \eta_c\nabla f(\mathbf{x}_{t-1} + u_t g(\mathbf{x}_t)\|$$

$$\leq \left\|H'_{t-1}\right\|(\eta_c Q + |u_t|Q + |u_t|\|\nabla f(\mathbf{x}_{t-1})\|) \tag{32a}$$

$$= (\eta_c + |u_t|)Q\left\|H'_{t-1}\right\| + |u_t|\left\|H'_{t-1}\right\|\|\Delta_{t-1}\| + |u_t|\left\|H'_{t-1}\right\|\left\|\nabla\tilde{f}(\tilde{\mathbf{x}}_{t-1})\right\|$$

$$= \tilde{O}\left(L_{max}^{11/8}\log\frac{1}{L_{max}}\right) + \tilde{O}\left(\mu L_{max}^{14/8}\log^2\frac{1}{L_{max}}\right) + \tilde{O}\left(L_{max}^{7/8}\log\frac{1}{L_{max}}\right)$$
$$\tag{32b}$$

$$= \tilde{O}\left(L_{max}^{7/8}\log\frac{1}{L_{max}}\right),$$

where we use assumption **A3** in equation 32a and the bounds of $R_{t-1}\cap C_{t-1}$ and equation 31d in equation 32b.

$$T_5: \quad \|\theta_{t-1}\| = \left\|\left(\int_0^1 \left(\mathbf{H}(\mathbf{x}_{t-1}+v(\mathbf{x}_t-\mathbf{x}_{t-1})) - \mathbf{H}(\mathbf{x}_{t-1})\right)dv\right)(\mathbf{x}_t-\mathbf{x}_{t-1})\right\|$$

$$\leq \left(\int_0^1 \rho\|\mathbf{x}_{t-1}+v(\mathbf{x}_t-\mathbf{x}_{t-1})-\mathbf{x}_{t-1}\|\,dv\right)\|\mathbf{x}_t-\mathbf{x}_{t-1}\| \tag{33a}$$

$$\leq \frac{\rho}{2}\|\mathbf{x}_t-\mathbf{x}_{t-1}\|^2 \leq \frac{\rho}{2}\|-\eta_c\nabla f(\mathbf{x}_{t-1})-\mathbf{w}_{t-1}\|^2$$

$$\leq \frac{\rho}{2}\|-\eta_c\nabla f(\mathbf{x}_{t-1})-\eta_c g(\mathbf{x}_{t-1})+\eta_c\nabla f(\mathbf{x}_{t-1})-u_t g(\mathbf{x}_{t-1})\|^2$$

$$\leq \frac{\rho|\eta_c+u_t|^2}{2}\left(Q^2 + \|\nabla f(\mathbf{x}_{t-1})\|^2 + 2Q\|\nabla f(\mathbf{x}_{t-1})\|\right)$$

$$= \frac{\rho|\eta_c+u_t|^2}{2}\left(Q^2 + \|\Delta_{t-1}\|^2 + \left\|\nabla\tilde{f}(\tilde{\mathbf{x}}_{t-1})\right\|^2 + 2\|\Delta_{t-1}\|\left\|\nabla\tilde{f}(\tilde{\mathbf{x}}_{t-1})\right\|\right.$$

$$\left. + 2Q\|\Delta_{t-1}\| + 2Q\left\|\nabla\tilde{f}(\tilde{\mathbf{x}}_{t-1})\right\|\right)$$

$$= \tilde{O}(L_{max}^2) + \tilde{O}\left(\mu^2 L_{max}^{11/4}\log^2\frac{1}{L_{max}}\right) + \tilde{O}(L_{max}) + \tilde{O}\left(\mu L_{max}^{15/8}\log\frac{1}{L_{max}}\right)$$

$$+ \tilde{O}\left(\mu L_{max}^{19/8}\log\frac{1}{L_{max}}\right) + \tilde{O}(L_{max}^{3/2}) = \tilde{O}(L_{max}). \tag{33b}$$

Here, we use assumption **A3** and the bounds of the event $R_{t-1}\cap C_{t-1}$ in equation 33b. Note that we have derived bounds so far conditioned on the event $R_{t-1}\cap C_{t-1}$. We now include this conditioning explicitly in our notations going forward.

To characterize $\|\Delta_t\|^2$, we construct a supermartingale process; and to do so, we focus on finding $\mathbb{E}[\|\Delta_t\|^2\mathbf{1}_{R_{t-1}\cap C_{t-1}}]$ using the bounds derived for the terms $T_1,\cdots,T_5$. Later, we use the Azuma-

Hoeffding inequality to obtain a probabilistic bound of $\|\Delta_t\|$.

$$
\begin{aligned}
\mathbb{E}[\|\Delta_t\|^2 \mathbf{1}_{R_{t-1}\cap C_{t-1}}|S_{t-1}] \leq &\left[ (1+\eta_c\gamma_o)^2 \|\Delta_{t-1}\|^2 + \tilde{O}\left(\mu L_{max}^{3/8}\log\frac{1}{L_{max}}\right)\tilde{O}\left(L_{max}^{7/8}\log\frac{1}{L_{max}}\right) \right. \\
&+ \tilde{O}\left(\mu L_{max}^{3/8}\log\frac{1}{L_{max}}\right)\tilde{O}(L_{max}) + \tilde{O}(L_{max}^2) \\
&\left. + \tilde{O}\left(L_{max}^{7/8}\log\frac{1}{L_{max}}\right)\tilde{O}(L_{max}) + \tilde{O}\left(L_{max}^{7/4}\log^2\frac{1}{L_{max}}\right) \right]\mathbf{1}_{R_{t-1}\cap C_{t-1}} \\
\leq &\left[ (1+\eta_c\gamma_o)^2 \|\Delta_{t-1}\|^2 + \tilde{O}\left(\mu L_{max}^{7/8}\log\frac{1}{L_{max}}\right) \right]\mathbf{1}_{R_{t-1}\cap C_{t-1}}
\end{aligned}
$$
$$(34)$$

Now, let

$$
G_t = (1+\eta_c\gamma_o)^{-2t}\left[ \|\Delta_t\|^2 + \tilde{O}\left(\mu L_{max}^{7/8}\log\frac{1}{L_{max}}\right) \right]. \tag{35}
$$

Now, in order to prove the process $G_t\mathbf{1}_{R_{t-1}\cap C_{t-1}}$ is a supermartingale, we prove that $\mathbb{E}[G_t\mathbf{1}_{R_{t-1}\cap C_{t-1}}|S_{t-1}] \leq G_{t-1}\mathbf{1}_{R_{t-2}\cap C_{t-2}}$. We define a filtration $S_t = s\{\mathbf{w}_0,\ldots,\mathbf{w}_{t-1}\}$ where $s\{.\}$ denotes a sigma-algebra field.

$$
\mathbb{E}[G_t\mathbf{1}_{R_{t-1}\cap C_{t-1}}|S_{t-1}]
$$
$$
\leq (1+\eta_c\gamma_o)^{-2t}\left( (1+\eta_c\gamma_o)^2 \|\Delta_{t-1}\|^2 + 2\tilde{O}\left(\mu L_{max}^{7/8}\log\frac{1}{L_{max}}\right) \right)\mathbf{1}_{R_{t-1}\cap C_{t-1}} \tag{36a}
$$
$$
\leq (1+\eta_c\gamma_o)^{-2t}\left( (1+\eta_c\gamma_o)^2 \|\Delta_{t-1}\|^2 + 2(1+\eta_c\gamma_o)^2\tilde{O}\left(\mu L_{max}^{7/8}\log\frac{1}{L_{max}}\right) \right)\mathbf{1}_{R_{t-1}\cap C_{t-1}} \tag{36b}
$$
$$
= (1+\eta_c\gamma_o)^{-2(t-1)}\left( \|\Delta_{t-1}\|^2 + \tilde{O}\left(\mu L_{max}^{7/8}\log\frac{1}{L_{max}}\right) \right)\mathbf{1}_{R_{t-1}\cap C_{t-1}}
$$
$$
= G_{t-1}\mathbf{1}_{R_{t-1}\cap C_{t-1}} \leq G_{t-1}\mathbf{1}_{R_{t-2}\cap C_{t-2}}.
$$

To obtain equation 36a, we use equation 34 to find $\mathbb{E}[G_t\mathbf{1}_{R_{t-1}\cap C_{t-1}}|S_{t-1}]$. In equation 36b, we upper bound by the multiplication of a positive term $(1+\eta_c\gamma_o)^2$. Therefore, $G_t\mathbf{1}_{R_{t-1}\cap C_{t-1}}$ is a supermartingale.

$$
\begin{aligned}
\|\Delta_t\|^2 - \mathbb{E}[\|\Delta_t\|^2|S_{t-1}]\mathbf{1}_{R_{t-1}\cap C_{t-1}} \leq & -2\|(I-\eta_c\mathbf{H}(\mathbf{x}_0))\Delta_{t-1}\| \|\mathbf{H}(\mathbf{x}_0)(\mathbf{w}_{t-1}-\tilde{\mathbf{w}}_{t-1})\| \\
& -2\|(I-\eta_c\mathbf{H}(\mathbf{x}_0))\Delta_{t-1}\| \|H'_{t-1}\mathbf{w}_{t-1}\| + 2\|(I-\eta_c\mathbf{H}(\mathbf{x}_0))\Delta_{t-1}\| \|\theta_{t-1}\| \\
& + \|\mathbf{H}(\mathbf{x}_0)(\mathbf{w}_{t-1}-\tilde{\mathbf{w}}_{t-1})\|^2 + \|H'_{t-1}\mathbf{w}_{t-1}\|^2 + 2\|\mathbf{H}(\mathbf{x}_0)(\mathbf{w}_{t-1}-\tilde{\mathbf{w}}_{t-1})\| \|H'_{t-1}\mathbf{w}_{t-1}\| \\
& + 2\|\mathbf{H}(\mathbf{x}_0)(\mathbf{w}_{t-1}-\tilde{\mathbf{w}}_{t-1})\| \|H'_{t-1}(\eta_c\Delta_{t-1}+\eta_c\nabla\tilde{f}(\tilde{\mathbf{x}}_{t-1}))\| \\
& - 2\|\mathbf{H}(\mathbf{x}_0)(\mathbf{w}_{t-1}-\tilde{\mathbf{w}}_{t-1})\| \|\theta_{t-1}\| + 2\|H'_{t-1}(\eta_c\Delta_{t-1}+\eta_c\nabla\tilde{f}(\tilde{\mathbf{x}}_{t-1}))\| \|H'_{t-1}\mathbf{w}_{t-1}\| \\
& - 2\|H'_{t-1}(\eta_c\Delta_{t-1}+\eta_c\nabla\tilde{f}(\tilde{\mathbf{x}}_{t-1}))\| \|\theta_{t-1}\| - 2\|H'_{t-1}\mathbf{w}_{t-1}\| \|\theta_{t-1}\| + \|\theta_{t-1}\|^2 \\
= & \tilde{O}\left(\mu L_{max}^{11/8}\log\frac{1}{L_{max}}\right) + \tilde{O}\left(\mu L_{max}^{10/8}\log^2\frac{1}{L_{max}}\right) + \tilde{O}(L_{max}^2) + \tilde{O}\left(L_{max}^{15/8}\log\frac{1}{L_{max}}\right) \\
& + \tilde{O}\left(L_{max}^{7/4}\log^2\frac{1}{L_{max}}\right) \leq \tilde{O}\left(\mu L_{max}^{7/8}\log\frac{1}{L_{max}}\right)
\end{aligned}
$$

Note that the above expression is obtained by the observation that the only random terms of $\Delta_t$ conditioned on the filtration $S_{t-1} = s\{\mathbf{w}_0,\mathbf{w}_1,\ldots,\mathbf{w}_{t-2}\}$ are $\mathbf{H}(\mathbf{x}_0)(\mathbf{w}_{t-1}-\tilde{\mathbf{w}}_{t-1})$, $H'_{t-1}\mathbf{w}_{t-1}$ and $\theta_{t-1}$(see equation 33a). Hence, we cancel out the deterministic terms in $\|\Delta_t\|^2$ and $\mathbb{E}\|\Delta_t\|^2$ and neglect the negative terms while upper bounding.

The Azuma-Hoeffding inequality for martingales and supermartingales (Hoeffding, 1994) states that if $\{G_t \mathbf{1}_{R_{t-1} \cap C_{t-1}}\}$ is a supermartingale and $|G_t \mathbf{1}_{R_{t-1} \cap C_{t-1}} - G_{t-1} \mathbf{1}_{R_{t-2} \cap C_{t-2}}| \le c_t$ almost surely, then for all positive integers $t$ and positive reals $\delta$,

$$\mathbb{P}(G_t \mathbf{1}_{R_{t-1} \cap C_{t-1}} - G_0 \mathbf{1}_{R_{-1} \cap C_{-1}} \ge \delta) \le \exp\left(-\frac{\delta^2}{2 \sum_{\tau=0}^{t-1} c_\tau^2}\right).$$

The bound of $|G_t \mathbf{1}_{R_{t-1} \cap C_{t-1}} - G_{t-1} \mathbf{1}_{R_{t-2} \cap C_{t-2}}|$ can be obtained using the definition of the process $G_t$ in equation 35. Recollecting our assumption that $\eta_c \le \eta_{max} \le \frac{\sqrt{2}-1}{\gamma'}, \gamma_o \le \gamma'$, we see that $(1 + \eta_c \gamma_o)^{-2t} \le \tilde{O}(1)$. Therefore,

$$|G_t \mathbf{1}_{R_{t-1} \cap C_{t-1}} - \mathbb{E}[G_t \mathbf{1}_{R_{t-1} \cap C_{t-1}} | S_{t-1}]| = (1 + \eta_c \gamma_o)^{-2t} \left| \|\Delta_t\|^2 - \mathbb{E}[\|\Delta_t\|^2 | S_{t-1}] \right| \mathbf{1}_{R_{t-1} \cap C_{t-1}}$$

$$\le \tilde{O}\left(\mu L_{max}^{7/8} \log \frac{1}{L_{max}}\right).$$

We denote the bound obtained for $|G_t \mathbf{1}_{R_{t-1} \cap C_{t-1}} - \mathbb{E}[G_t \mathbf{1}_{R_{t-1} \cap C_{t-1}} | S_{t-1}]|$ as $c_{t-1}$. Now, let $\delta = \sqrt{\sum_{\tau=0}^{t-1} c_\tau^2} \log \frac{1}{L_{max}}$ in the Azuma-Hoeffding inequality. Now, for any $t \le T = O\left(L_{max}^{-1/4}\right)$,

$$\delta = \sqrt{O\left(\frac{1}{L_{max}^{1/4}}\right) \tilde{O}\left(\mu^2 L_{max}^{7/4} \log^2 \frac{1}{L_{max}}\right)} \log \frac{1}{L_{max}} = \tilde{O}\left(\mu L_{max}^{3/4} \log^2 \frac{1}{L_{max}}\right).$$

$$\mathbb{P}\left(G_t \mathbf{1}_{R_{t-1} \cap C_{t-1}} - G_0.1 \ge \tilde{O}\left(\mu L_{max}^{3/4} \log^2 \frac{1}{L_{max}}\right)\right) \le \exp\left(-\tilde{\Omega}\left(\log^2 \frac{1}{L_{max}}\right)\right)$$

$$\le \tilde{O}(L_{max}^4).$$

After taking union bound $\forall\, t \le T$,

$$\mathbb{P}\left(\forall\, t \le T,\ G_t \mathbf{1}_{R_{t-1} \cap C_{t-1}} - G_0 \ge \tilde{O}\left(\mu L_{max}^{3/4} \log^2 \frac{1}{L_{max}}\right)\right) \le \tilde{O}(L_{max}^{15/4}).$$

We represent the hidden constants in $\tilde{O}\left(\mu L_{max}^{3/4} \log^2 \frac{1}{L_{max}}\right)$ by $\tilde{c}$ and choose $\mu$ such that $\mu < \tilde{c}$. Then, the following equation holds true.

$$\mathbb{P}\left(G_t \mathbf{1}_{R_{t-1} \cap C_{t-1}} - G_0 \ge \mu^2 L_{max}^{3/4} \log^2 \frac{1}{L_{max}}\right) \le \tilde{O}(L_{max}^{15/4}).$$

Hence we can write,

$$\mathbb{P}\left(R_{t-1} \cap C_{t-1} \cap \left\{\|\Delta_t\| \ge \mu L_{max}^{3/8} \log \frac{1}{L_{max}}\right\}\right) \le \tilde{O}(L_{max}^{15/4}). \tag{37}$$

We need the probability of the event $C_t, \forall t \le T$ in order to prove the lemma. From Lemma 2, we get the probability of the event $\bar{R}_t$ as $\tilde{O}(L_{max}^{15/4})$. Then,

$$\mathbb{P}\left(C_{t-1} \cap \left\{\|\Delta_t\| \ge \mu L_{max}^{3/8} \log \frac{1}{L_{max}}\right\}\right) = \mathbb{P}\left(R_{t-1} \cap C_{t-1} \cap \left\{\|\Delta_t\| \ge \mu L_{max}^{3/8} \log \frac{1}{L_{max}}\right\}\right)$$

$$+ \mathbb{P}\left(\bar{R}_{t-1} \cap C_{t-1} \cap \left\{\|\Delta_t\| \ge \mu L_{max}^{3/8} \log \frac{1}{L_{max}}\right\}\right)$$

$$\le \tilde{O}(L_{max}^{15/4}) + \mathbb{P}(\bar{R}_{t-1}) \le \tilde{O}(L_{max}^{15/4}), \tag{38}$$

where the first term of equation 38 follows from equation 37. The second term of equation 38 can be bounded by $\mathbb{P}(\bar{R}_{t-1})$ which is known by Lemma 2. Finally,

$$\mathbb{P}(\bar{C}_t) = \mathbb{P}\left(C_{t-1} \cap \left\{\|\Delta_t\| \ge \mu L_{max}^{3/8} \log \frac{1}{L_{max}}\right\}\right) + \mathbb{P}(\bar{C}_{t-1}) \le \tilde{O}(L_{max}^{15/4}) + \mathbb{P}(\bar{C}_{t-1}).$$

The probability $\mathbb{P}(\bar{C}_{t-1})$ can be found as,

$$\mathbb{P}(\bar{C}_{t-1}) = \mathbb{P}\left( C_{t-2} \cap \left\{ \|\Delta_{t-1}\| \ge \mu L_{max}^{3/8} \log \frac{1}{L_{max}} \right\} \right) + \mathbb{P}(\bar{C}_{t-2})$$

$$= \mathbb{P}\left( C_{t-2} \cap \left\{ \|\Delta_{t-1}\| \ge \mu L_{max}^{3/8} \log \frac{1}{L_{max}} \right\} \right) + \dots$$

$$+ \mathbb{P}\left( C_0 \cap \left\{ \|\Delta_1\| \ge \mu L_{max}^{3/8} \log \frac{1}{L_{max}} \right\} \right) + \mathbb{P}(\bar{C}_0).$$

As $T = O\left( L_{max}^{-1/4} \right)$, $\mathbb{P}(\bar{C}_T) \le \tilde{O}\left( L_{max}^{7/2} \right)$. From equation 26,

$$\|\mathbf{x}_t - \tilde{\mathbf{x}}_t\| \le \sum_{\tau=0}^{t-1} (\eta_c + |u_{\tau+1}|) \left( \|\Delta_\tau\| + Q + \tilde{Q} \right)$$

$$\le O\left( \frac{1}{L_{max}^{1/4}} \right) \left( \tilde{O}(L_{max}) \mu L_{max}^{3/8} \log \frac{1}{L_{max}} + \tilde{O}(L_{max}) \right)$$

$$= O\left( \mu L_{max}^{9/8} \log \frac{1}{L_{max}} \right) + \tilde{O}(L_{max}^{3/4}) \le \tilde{O}(L_{max}^{3/4})$$

This completes our proof. $\qquad\square$

## C  PROOF OF THEOREM 2

**Theorem 5.** *(Theorem 2 restated) Consider $f$ satisfying Assumptions A1 - A5. Let $\tilde{f}$ be the second order Taylor approximation of $f$; let $\{x_t\}$ and $\{\tilde{x}_t\}$ be the corresponding SGD iterates using PLRS, with $\tilde{x}_0 = x_0$. Let $x_0$ correspond to B2, i.e., $\|\nabla f(x_0)\| \le \epsilon$ and $\lambda_{min}(\boldsymbol{H}(x_0)) \le -\gamma$ where $\epsilon, \gamma > 0$. Then, there exists a $T = \tilde{O}\left( L_{max}^{-1/4} \right)$ such that with probability at least $1 - \tilde{O}\left( L_{max}^{7/2} \right)$,*

$$\mathbb{E}[f(x_T) - f(x_0)] \le -\tilde{\Omega}\left( L_{max}^{3/4} \right).$$

*Proof.* In this proof, we consider the case when the initial iterate $\mathbf{x}_0$ is at a saddle point (corresponding to **B2**). This theorem shows that the SGD-PLRS algorithm escapes the saddle point in $T$ steps where $T = \tilde{O}\left( L_{max}^{-1/4} \right)$.

We use the Taylor series approximation in order to make the problem tractable. Similar to the SGD-PLRS updates for the function $f$, the SGD update on the function $\tilde{f}$ can be given as,

$$\tilde{\mathbf{x}}_t = \tilde{\mathbf{x}}_{t-1} - \eta_c \nabla \tilde{f}(\tilde{\mathbf{x}}_{t-1}) - \tilde{\mathbf{w}}_{t-1}, \quad \tilde{\mathbf{w}}_{t-1} = \eta_c \tilde{g}(\tilde{\mathbf{x}}_{t-1}) - \eta_c \nabla \tilde{f}(\tilde{\mathbf{x}}_{t-1}) + u_t \tilde{g}(\tilde{\mathbf{x}}_{t-1}).$$

As the function $f$ is $\rho$-Hessian, using (Nesterov, 2014, Lemma 1.2.4) and the Taylor series expansion one obtains, $f(\mathbf{x}) \le f(\mathbf{x}_0) + \nabla f(\mathbf{x}_0)^T (\mathbf{x} - \mathbf{x}_0) + \frac{1}{2} (\mathbf{x} - \mathbf{x}_0)^T \mathbf{H}(\mathbf{x}_0)(\mathbf{x} - \mathbf{x}_0) + \frac{\rho}{6} \|\mathbf{x} - \mathbf{x}_0\|^3$. Let $\tilde{\boldsymbol{\kappa}} = \tilde{\mathbf{x}}_T - \mathbf{x}_0$, $\boldsymbol{\kappa} = \mathbf{x}_T - \tilde{\mathbf{x}}_T$. Note that $\tilde{\boldsymbol{\kappa}} + \boldsymbol{\kappa} = \mathbf{x}_T - \mathbf{x}_0$. Then, replacing $\mathbf{x}$ by $\mathbf{x}_T$,

$$f(\mathbf{x}_T) - f(\mathbf{x}_0) \le \nabla f(\mathbf{x}_0)^T (\mathbf{x}_T - \mathbf{x}_0) + \frac{1}{2} (\mathbf{x}_T - \mathbf{x}_0)^T \mathbf{H}(\mathbf{x}_0)(\mathbf{x}_T - \mathbf{x}_0) + \frac{\rho}{6} \|\mathbf{x}_T - \mathbf{x}_0\|^3$$

$$= \nabla f(\mathbf{x}_0)^T (\tilde{\boldsymbol{\kappa}} + \boldsymbol{\kappa}) + \frac{1}{2} (\tilde{\boldsymbol{\kappa}} + \boldsymbol{\kappa})^T \mathbf{H}(\mathbf{x}_0)(\tilde{\boldsymbol{\kappa}} + \boldsymbol{\kappa}) + \frac{\rho}{6} \|\tilde{\boldsymbol{\kappa}} + \boldsymbol{\kappa}\|^3$$

$$= \left( \nabla f(\mathbf{x}_0)^T \tilde{\boldsymbol{\kappa}} + \frac{1}{2} \tilde{\boldsymbol{\kappa}}^T \mathbf{H}(\mathbf{x}_0) \tilde{\boldsymbol{\kappa}} \right) + \left( \nabla f(\mathbf{x}_0)^T \boldsymbol{\kappa} + \tilde{\boldsymbol{\kappa}}^T \mathbf{H}(\mathbf{x}_0) \boldsymbol{\kappa} + \frac{1}{2} \boldsymbol{\kappa}^T \mathbf{H}(\mathbf{x}_0) \boldsymbol{\kappa} \right.$$

$$\left. + \frac{\rho}{6} \|\tilde{\boldsymbol{\kappa}} + \boldsymbol{\kappa}\|^3 \right).$$

Let the first term be $\tilde{\zeta} = \nabla f(\mathbf{x}_0)^T \tilde{\boldsymbol{\kappa}} + \frac{1}{2} \tilde{\boldsymbol{\kappa}}^T \mathbf{H}(\mathbf{x}_0) \tilde{\boldsymbol{\kappa}}$ and the second term be $\zeta = \nabla f(\mathbf{x}_0)^T \boldsymbol{\kappa} + \tilde{\boldsymbol{\kappa}}^T \mathbf{H}(\mathbf{x}_0) \boldsymbol{\kappa} + \frac{1}{2} \boldsymbol{\kappa}^T \mathbf{H}(\mathbf{x}_0) \boldsymbol{\kappa} + \frac{\rho}{6} \|\tilde{\boldsymbol{\kappa}} + \boldsymbol{\kappa}\|^3$. Hence $f(\mathbf{x}_T) - f(\mathbf{x}_0) \le \tilde{\zeta} + \zeta$. In order to prove the theorem, we require an upper bound on $\mathbb{E}[f(\mathbf{x}_T) - f(\mathbf{x}_0)]$.

Now, we introduce two mutually exclusive events $C_t$ and $\bar{C}_t$ so that $\mathbb{E}[f(\mathbf{x}_T) - f(\mathbf{x}_0)]$ can be written in terms of events $C_t$ and $\bar{C}_t$ as,

$$
\begin{aligned}
\mathbb{E}[f(\mathbf{x}_T) - f(\mathbf{x}_0)] &= \mathbb{E}[f(\mathbf{x}_T) - f(\mathbf{x}_0)](\mathbb{E}[\mathbf{1}_{C_T}] + \mathbb{E}[\mathbf{1}_{\bar{C}_T}]) \\
&= \mathbb{E}[(f(\mathbf{x}_T) - f(\mathbf{x}_0))\mathbf{1}_{C_T}] + \mathbb{E}[(f(\mathbf{x}_T) - f(\mathbf{x}_0))\mathbf{1}_{\bar{C}_T}] \\
&\leq \mathbb{E}[\tilde{\zeta}\mathbf{1}_{C_T}] + \mathbb{E}[\zeta\mathbf{1}_{C_T}] + \mathbb{E}[(f(\mathbf{x}_T) - f(\mathbf{x}_0))\mathbf{1}_{\bar{C}_T}] \\
&= \mathbb{E}[\tilde{\zeta}] + \mathbb{E}[\zeta\mathbf{1}_{C_T}] + \mathbb{E}[(f(\mathbf{x}_T) - f(\mathbf{x}_0))\mathbf{1}_{\bar{C}_T}] - \mathbb{E}[\tilde{\zeta}\mathbf{1}_{\bar{C}_T}].
\end{aligned}
$$

Let $K_1 = \mathbb{E}[\tilde{\zeta}]$, $K_2 = \mathbb{E}[\zeta\mathbf{1}_{C_T}]$ and $K_3 = \mathbb{E}[(f(\mathbf{x}_T) - f(\mathbf{x}_0))\mathbf{1}_{\bar{C}_T}] - \mathbb{E}[\tilde{\zeta}\mathbf{1}_{\bar{C}_T}]$. In the remainder of the proof, we focus on deriving the bounds for individual terms, $K_1$, $K_2$ and $K_3$, and then finally put them together to obtain the result of the theorem.

## C.1 Bounding $K_1$

Using equation 20b from the proof of Lemma 2 in Appendix B.1, we obtain the bound for the term $K_1 = \mathbb{E}[\tilde{\zeta}]$ as,

$$
\begin{aligned}
\mathbb{E}[\tilde{\zeta}] &= \mathbb{E}\left[\nabla f(\mathbf{x}_0)^T(\tilde{\mathbf{x}}_T - \mathbf{x}_0) + \frac{1}{2}(\tilde{\mathbf{x}}_T - \mathbf{x}_0)^T\mathbf{H}(\mathbf{x}_0)(\tilde{\mathbf{x}}_T - \mathbf{x}_0)\right] \\
&= \mathbb{E}\left[\nabla f(\mathbf{x}_0)^T\left(-\sum_{\tau=0}^{T-1}\eta_c(I - \eta_c\mathbf{H}(\mathbf{x}_0))^\tau\nabla f(\mathbf{x}_0) - \sum_{\tau=0}^{T-1}(I - \eta_c\mathbf{H}(\mathbf{x}_0))^{T-\tau-1}\tilde{\mathbf{w}}_\tau\right)\right] \\
&\quad + \frac{1}{2}\mathbb{E}\left[\left(-\sum_{\tau=0}^{T-1}\eta_c(I - \eta_c\mathbf{H}(\mathbf{x}_0))^\tau\nabla f(\mathbf{x}_0) - \sum_{\tau=0}^{T-1}(I - \eta_c\mathbf{H}(\mathbf{x}_0))^{T-\tau-1}\tilde{\mathbf{w}}_\tau\right)^T\mathbf{H}(\mathbf{x}_0)\right. \\
&\quad \left.\left(-\sum_{\tau=0}^{T-1}\eta_c(I - \eta_c\mathbf{H}(\mathbf{x}_0))^\tau\nabla f(\mathbf{x}_0) - \sum_{\tau=0}^{T-1}(I - \eta_c\mathbf{H}(\mathbf{x}_0))^{T-\tau-1}\tilde{\mathbf{w}}_\tau\right)\right].
\end{aligned}
$$

Since $\tilde{\mathbf{w}}_\tau = \mathbf{0}$, all the terms with $\mathbb{E}[\tilde{\mathbf{w}}_\tau]$ will go to zero. Hence we obtain,

$$
\begin{aligned}
\mathbb{E}[\tilde{\zeta}] &= \nabla f(\mathbf{x}_0)^T\left(-\sum_{\tau=0}^{T-1}\eta_c(I - \eta_c\mathbf{H}(\mathbf{x}_0))^\tau\nabla f(\mathbf{x}_0)\right) + \\
&\quad \frac{1}{2}\left(-\sum_{\tau=0}^{T-1}\eta_c(I - \eta_c\mathbf{H}(\mathbf{x}_0))^\tau\nabla f(\mathbf{x}_0)\right)^T\mathbf{H}(\mathbf{x}_0)\left(-\sum_{\tau=0}^{T-1}\eta_c(I - \eta_c\mathbf{H}(\mathbf{x}_0))^\tau\nabla f(\mathbf{x}_0)\right) \\
&\quad + \frac{1}{2}\mathbb{E}\left[\left(-\sum_{\tau=0}^{T-1}(I - \eta_c\mathbf{H}(\mathbf{x}_0))^{T-\tau-1}\tilde{\mathbf{w}}_\tau\right)^T\mathbf{H}(\mathbf{x}_0)\left(-\sum_{\tau=0}^{T-1}(I - \eta_c\mathbf{H}(\mathbf{x}_0))^{T-\tau-1}\tilde{\mathbf{w}}_\tau\right)\right].
\end{aligned}
$$

Let $\lambda_1, \ldots, \lambda_d$ be the eigenvalues of the Hessian matrix at $\mathbf{x}_0$, $\mathbf{H}(\mathbf{x}_0)$. Now, we simplify similar to Ge et al. (Ge et al., 2015) as,

$$
\begin{aligned}
\mathbb{E}[\tilde{\zeta}] &= -\sum_{i=1}^d\sum_{\tau=0}^{T-1}\eta_c(1 - \eta_c\lambda_i)^\tau|\nabla_i f(\mathbf{x}_0)|^2 + \frac{1}{2}\sum_{i=1}^d\lambda_i\sum_{\tau=0}^{T-1}\eta_c^2(1 - \eta_c\lambda_i)^{2\tau}|\nabla_i f(\mathbf{x}_0)|^2 \\
&\quad + \frac{1}{2}\sum_{i=1}^d\lambda_i\sum_{\tau=0}^{T-1}(1 - \eta_c\lambda_i)^{2(T-\tau-1)}\mathbb{E}[|\tilde{\mathbf{w}}_{\tau,i}|^2].
\end{aligned}
$$

Note that for the case of very small gradients (as per our initial conditions), $|\nabla_i f(\mathbf{x}_0)|^2 \leq \|\nabla f(\mathbf{x}_0)\| \leq \epsilon$. Therefore, the first and second terms can be made arbitrarily small so that they do not contribute to the order of the equation. Hence, we focus on the third term. We first characterize $\mathbb{E}[|\tilde{\mathbf{w}}_{\tau,i}|^2]$ as follows. Since the norm of the stochastic noise is bounded as per the assumption **A3**,

we assume that $\tilde{g}_i(\tilde{\mathbf{x}}_t) - \nabla_i \tilde{f}(\tilde{\mathbf{x}}_t) \leq \tilde{q}$ and $\mathbb{E}[\tilde{q}] \leq \tilde{\sigma}^2$.

$$\tilde{\mathbf{w}}_{\tau,i} = \eta_c \tilde{g}_i(\tilde{\mathbf{x}}_t) - \eta_c \nabla_i \tilde{f}(\tilde{\mathbf{x}}_t) + u_{t+1}\tilde{g}_i(\tilde{\mathbf{x}}_t)$$

$$\leq \eta_c \tilde{q} + u_{t+1}\left(\tilde{g}_i(\tilde{\mathbf{x}}_t) - \nabla_i \tilde{f}(\tilde{\mathbf{x}}_t) + \nabla_i \tilde{f}(\tilde{\mathbf{x}}_t)\right)$$

$$\leq \tilde{q}(\eta_c + u_{t+1}) + u_{t+1}\nabla_i \tilde{f}(\tilde{\mathbf{x}}_t)$$

$$|\tilde{\mathbf{w}}_{\tau,i}|^2 \leq \left(\tilde{q}(\eta_c + u_{t+1}) + u_{t+1}\nabla_i \tilde{f}(\tilde{\mathbf{x}}_t)\right)^2$$

$$= \tilde{q}^2(\eta_c^2 + 2\eta_c u_{t+1} + u_{t+1}^2) + 2\tilde{q}\eta_c u_{t+1}\nabla_i \tilde{f}(\tilde{\mathbf{x}}_t) + 2\tilde{q}u_{t+1}^2\nabla_i \tilde{f}(\tilde{\mathbf{x}}_t) + u_{t+1}^2\left|\nabla_i \tilde{f}(\tilde{\mathbf{x}}_t)\right|^2.$$

Taking expectation with respect to $\tilde{q}$ and the uniformly distributed random variable $u_{t+1}$ and recalling that $\mathbb{E}[u_{t+1}] = 0$, we set expectation over linear functions of $u_{t+1}$ to zero.

$$\mathbb{E}[|\tilde{\mathbf{w}}_{\tau,i}|^2] \leq \tilde{\sigma}^2\eta_c^2 + \tilde{\sigma}^2\mathbb{E}[u_{t+1}^2] + 2\tilde{\sigma}^2\mathbb{E}[u_{t+1}^2]\nabla_i \tilde{f}(\tilde{\mathbf{x}}_t) + \mathbb{E}[u_{t+1}^2]\left|\nabla_i \tilde{f}(\tilde{\mathbf{x}}_t)\right|^2$$

$$\leq \tilde{O}(L_{max}^2) + \tilde{O}(L_{max}^2) + \tilde{O}(L_{max}^2)\tilde{O}\left(\frac{1}{\sqrt{L_{max}}}\right) + \tilde{O}(L_{max}^2)\tilde{O}\left(\frac{1}{L_{max}}\right) \quad (39)$$

$$= \tilde{O}(L_{max}^2) + \tilde{O}(L_{max}^{1.5}) + \tilde{O}(L_{max}) = \tilde{O}(L_{max}^2).$$

Here, we use $\mathbb{E}[u_{t+1}^2] = \frac{(L_{max}-L_{min})^2}{12} = \tilde{O}(L_{max}^2)$. From equation 19 in the proof of Lemma 2 (Appendix B.1), $\left\|\nabla \tilde{f}(\tilde{\mathbf{x}}_t)\right\| \leq 10\tilde{Q}\sum_{\tau=0}^{\frac{t(t-1)}{2}}(1+\eta_c\gamma_o)^\tau = \tilde{O}\left(\frac{1}{\sqrt{L_{max}}}\right)$ as $t \leq T = \tilde{O}\left(L_{max}^{-1/4}\right)$. Also, note that $\tilde{q}$ and $u_{t+1}$ are independent of each other. As $\lambda_{min}(\mathbf{H}(\mathbf{x}_0)) = -\gamma_o$,

$$\frac{1}{2}\sum_{i=1}^{d}\lambda_i\sum_{\tau=0}^{T-1}(1-\eta_c\lambda_i)^{2(T-\tau-1)}\mathbb{E}[|\tilde{\mathbf{w}}_{\tau,i}|^2]$$

$$\leq \frac{1}{2}\sum_{i=1}^{d}\lambda_i\sum_{\tau=0}^{T-1}(1+\eta_c\gamma_o)^{2\tau}\mathbb{E}[|\tilde{\mathbf{w}}_{\tau,i}|^2] \leq \frac{\tilde{O}(L_{max})}{2}\sum_{i=1}^{d}\lambda_i\sum_{\tau=0}^{T-1}(1+\eta_c\gamma_o)^{2\tau} \quad (40a)$$

$$= \frac{\tilde{O}(L_{max})}{2}\left(-\gamma_o\sum_{\tau=0}^{T-1}(1+\eta_c\gamma_o)^{2\tau} + (d-1)\lambda_{max}(\mathbf{H}(\mathbf{x}_0))\sum_{\tau=0}^{T-1}(1+\eta_c\gamma_o)^{2\tau}\right), \quad (40b)$$

where we use the upper bound of $\mathbb{E}[|\tilde{\mathbf{w}}_{\tau,i}|^2]$ obtained from equation 39 in equation 40a. We use the fact that one of the eigenvalues of $\mathbf{H}(\mathbf{x}_0)$ is $-\gamma_o$ and then upper bound the other eigenvalues by the maximum eigenvalue $\lambda_{max}(\mathbf{H}(\mathbf{x}_0))$ in equation 40b.

Let $\eta_c \leq \eta_{max} \leq \frac{\sqrt{2}-1}{\gamma'}$ where $\gamma \leq \gamma_o \leq \gamma'$. As $\sum_{\tau=0}^{T-1}(1+\eta_c\gamma_o)^{2\tau}$ is a monotonically increasing sequence, we choose the smallest $T$ that satisfies $\frac{d}{\eta_c^{1/4}\gamma_o} \leq \sum_{\tau=0}^{T-1}(1+\eta_c\gamma_o)^{2\tau}$. Therefore, $\sum_{\tau=0}^{T-2}(1+\eta_c\gamma_o)^{2\tau} \leq \frac{d}{\eta_c^{1/4}\gamma_o}$. Now,

$$\sum_{\tau=0}^{T-1}(1+\eta_c\gamma_o)^{2\tau} = 1 + (1+\eta_c\gamma_o)^2\sum_{\tau=0}^{T-2}(1+\eta_c\gamma_o)^{2\tau} \leq 1 + \frac{2d}{\eta_c^{1/4}\gamma_o},$$

which follows from our constraints that $\eta_c < \frac{\sqrt{2}-1}{\gamma'}$ and $\gamma_o \leq \gamma'$ making $(1+\eta_c\gamma)^2 \leq \left(1+\frac{\sqrt{2}-1}{\gamma'}\gamma'\right)^2 \leq 2$. Further using $\eta_c\gamma_o \leq \eta_c^{1/4}\gamma_o \leq \frac{\sqrt{2}-1}{\gamma'}\gamma' < d$,

$$\frac{d}{\eta_c^{1/4}\gamma_o} \leq \sum_{\tau=0}^{T-1}(1+\eta_c\gamma_o)^{2\tau} \leq 1 + \frac{2d}{\eta_c^{1/4}\gamma_o} \leq \frac{3d}{\eta_c^{1/4}\gamma_o} \quad (41)$$

Hence the order of $T$ is given by $T = O\left(\frac{\log d}{L_{max}^{1/4}\gamma_o}\right)$. We hide the dependence on $d$ when we use $T = \tilde{O}\left(L_{max}^{-1/4}\right)$. Using equation 41 it can be proved that,

$$\frac{1}{2}\sum_{i=1}^{d}\lambda_i\sum_{\tau=0}^{T-1}(1-\eta_c\lambda_i)^{2(T-\tau-1)}\mathbb{E}[|\tilde{\mathbf{w}}_{\tau,i}|^2] \leq -\tilde{O}(L_{max}^{3/4}).$$

## C.2   BOUNDING $K_2$ AND $K_3$

We define the event $C_T$ as, $C_T = \left\{ \forall t \leq T, \|\tilde{\kappa}\| \leq \tilde{O}\left(L_{max}^{3/8} \log \frac{1}{L_{max}}\right), \|\kappa\| \leq \tilde{O}(L_{max}^{3/4}) \right\}$. From Lemma 2 and Lemma 3 in Appendix B.1 and B.2 respectively, we know that with probability $\mathbb{P}(C_T) \geq 1 - \tilde{O}\left(L_{max}^{7/2}\right)$, the term $\|\tilde{\kappa}\|$ can be bounded by $\tilde{O}\left(L_{max}^{3/8} \log \frac{1}{L_{max}}\right)$ and $\|\kappa\|$ can be bounded by $\tilde{O}(L_{max}^{3/4}), \forall t \leq T = O\left(L_{max}^{-1/4}\right)$.

Now, to complete the proof of Theorem 2, we need to show that the term $K_1$ dominates both $K_2$ and $K_3$. Hence, we obtain the bound for the term $K_2$ as,

$$\mathbb{E}[\zeta \mathbf{1}_{C_T}] = \mathbb{E}\left[\nabla f(\mathbf{x}_0)^T \kappa + \tilde{\kappa}^T \mathbf{H}(\mathbf{x}_0)\kappa + \frac{1}{2}\kappa^T \mathbf{H}(\mathbf{x}_0)\kappa + \frac{\rho}{6}\|\tilde{\kappa} + \kappa\|^3\right]\mathbb{P}(C_T)$$

$$\leq \tilde{O}\left(L_{max}^{3/8} \log \frac{1}{L_{max}}\right)\tilde{O}(L_{max}^{3/4})\mathbb{P}(C_T) = \tilde{O}\left(L_{max}^{9/8} \log \frac{1}{L_{max}}\right)\mathbb{P}(C_T).$$

Finally, we bound the term $K_3$ as follows.

$$\mathbb{E}[(f(\mathbf{x}_T) - f(\mathbf{x}_0))\mathbf{1}_{\bar{C}_T}] - \mathbb{E}[\tilde{\zeta}\mathbf{1}_{\bar{C}_T}] \leq \tilde{O}(1)\mathbb{P}(\bar{C}_T) \leq \tilde{O}\left(L_{max}^{7/2}\right),$$

where the inequality arises from the boundedness of the function. Comparing the bounds of the terms $K_1$, $K_2$, and $K_3$, we find that $K_1$ dominates, which completes the proof. $\qquad\square$

# D   PROOF OF THEOREM 3

**Theorem 6.** *(Theorem 3 restated) Consider $f$ satisfying the assumptions **A1-A6**. Let the initial iterate $\boldsymbol{x}_0$ be $\delta$ close to a local minimum $\boldsymbol{x}^*$ such that $\|\boldsymbol{x}_0 - \boldsymbol{x}^*\| \leq \tilde{O}(\sqrt{L_{max}}) < \delta$. With probability at least $1 - \xi$, $\forall t \leq T$ where $T = \tilde{O}\left(\frac{1}{L_{max}^2} \log \frac{1}{\xi}\right)$,*

$$\|\boldsymbol{x}_t - \boldsymbol{x}^*\| \leq \tilde{O}\left(\sqrt{L_{max} \log \frac{1}{L_{max}\xi}}\right) < \delta$$

*Proof.* This theorem handles the case when the iterate is close to the local minimum (case **B3**). We aim to show that the iterate does not leave the neighbourhood of the minimum for $t \leq \tilde{O}\left(\frac{1}{L_{max}^2} \log \frac{1}{\xi}\right)$. By assumption **A6**, if $\mathbf{x}_t$ is $\delta$ close to the local minimum $\mathbf{x}^*$, the function is locally $\alpha$- strongly convex. We define event $D_t = \{\forall \tau \leq t, \|\mathbf{x}_\tau - \mathbf{x}^*\| \leq \mu\sqrt{L_{max} \log \frac{1}{L_{max}\xi}} < \delta\}$. Let $L_{max} < \frac{r}{\log \xi^{-1}}$ where $r < \log \xi^{-1}$. It can be seen that $D_{t-1} \subset D_t$. Conditioned on event $D_t$, and using $\alpha-$strong convexity of $f$, $(\nabla f(\mathbf{x}_t) - \nabla f(\mathbf{x}^*))^T(\mathbf{x}_t - \mathbf{x}^*)\mathbf{1}_{D_t} \geq \alpha \|\mathbf{x}_t - \mathbf{x}^*\|^2 \mathbf{1}_{D_t}$. As $\nabla f(\mathbf{x}^*) = 0$, it becomes, $\nabla f(\mathbf{x}_t)^T(\mathbf{x}_t - \mathbf{x}^*)\mathbf{1}_{D_t} \geq \alpha \|\mathbf{x}_t - \mathbf{x}^*\|^2 \mathbf{1}_{D_t}$. We define a filtration $S_t = s\{\mathbf{w}_0, \dots, \mathbf{w}_{t-1}\}$ in order to construct a supermartingale and use the Azuma-Hoeffding inequality where $s\{.\}$ denotes a sigma-algebra field. Now, assuming $L_{max} < \frac{\alpha}{\beta^2}$,

$$\mathbb{E}[\|\mathbf{x}_t - \mathbf{x}^*\|^2 \mathbf{1}_{D_{t-1}}|S_{t-1}] = \mathbb{E}[\|\mathbf{x}_{t-1} - \eta_c\nabla f(\mathbf{x}_{t-1}) - \mathbf{w}_{t-1} - \mathbf{x}^*\|^2 |S_{t-1}]\mathbf{1}_{D_{t-1}}$$

$$= \mathbb{E}[\|(\mathbf{x}_{t-1} - \mathbf{x}^*) - \eta_c\nabla f(\mathbf{x}_{t-1}) - \mathbf{w}_{t-1}\|^2 |S_{t-1}]\mathbf{1}_{D_{t-1}}$$

$$= [\|\mathbf{x}_{t-1} - \mathbf{x}^*\|^2 - 2\eta_c(\mathbf{x}_{t-1} - \mathbf{x}^*)^T\nabla f(\mathbf{x}_{t-1}) + \eta_c^2\|\nabla f(\mathbf{x}_{t-1})\|^2 + \mathbb{E}[\|\mathbf{w}_{t-1}\|^2]]\mathbf{1}_{D_{t-1}} \quad \text{(42a)}$$

$$\leq [\|\mathbf{x}_{t-1} - \mathbf{x}^*\|^2 - 2\eta_c\alpha\|\mathbf{x}_{t-1} - \mathbf{x}^*\|^2 + \eta_c^2\beta^2\|\mathbf{x}_{t-1} - \mathbf{x}^*\|^2 + \mathbb{E}[\|\mathbf{w}_{t-1}\|^2]]\mathbf{1}_{D_{t-1}} \quad \text{(42b)}$$

We use $\mathbb{E}[\mathbf{w}_t] = 0$ in equation 42a. We use the $\beta$-smoothness and $\alpha-$convexity assumptions of $f$ in equation 42b. Now, using $\mathbf{w}_{t-1} = \eta_c g(\mathbf{x}_{t-1}) - \eta_c\nabla f(\mathbf{x}_{t-1}) + u_t g(\mathbf{x}_{t-1})$, we compute $\mathbb{E}[\|\mathbf{w}_{t-1}\|^2]$

as,

$$\mathbb{E}[\|\mathbf{w}_{t-1}\|^2]$$

$$= \mathbb{E}\left[\eta_c^2 \|g(\mathbf{x}_{t-1}) - \nabla f(\mathbf{x}_{t-1})\|^2 + 2\eta_c u_t \left(g(\mathbf{x}_{t-1}) - \nabla f(\mathbf{x}_{t-1})\right)^T g(\mathbf{x}_{t-1}) + u_t^2 \|g(\mathbf{x}_{t-1})\|^2\right]$$

$$\leq \eta_c^2 \sigma^2 + \mathbb{E}[u_t^2]\mathbb{E}[\|g(\mathbf{x}_{t-1})\|^2] \leq \eta_c^2 \sigma^2 + \mathbb{E}[u_t^2](\sigma^2 + \|\nabla f(\mathbf{x}_{t-1})\|^2)$$

$$\leq \eta_c^2 \sigma^2 + \mathbb{E}[u_t^2]\sigma^2 + \mathbb{E}[u_t^2]\beta^2 \|\mathbf{x}_{t-1} - \mathbf{x}^*\|^2$$

$$\leq \sigma^2 \left(\eta_c^2 + \frac{2L_{max}^2}{3} - \frac{2L_{max}\eta_c}{3}\right) + \beta^2 \|\mathbf{x}_{t-1} - \mathbf{x}^*\|^2 \left(\frac{2L_{max}^2}{3} - \frac{2L_{max}\eta_c}{3}\right).$$

$$(43)$$

As $\eta_c = \frac{L_{min}+L_{max}}{2}$, $L_{min} = 2\eta_c - L_{max}$. Hence, we write $\mathbb{E}[u_t^2] = \frac{(L_{max}-L_{min})^2}{12} = \frac{4(L_{max}-\eta_c)^2}{12} = \frac{L_{max}^2 + \eta_c^2 - 2L_{max}\eta_c}{3} < \frac{2L_{max}^2}{3} - \frac{2L_{max}\eta_c}{3}$ in equation 43. Using equation 43 in equation 42b,

$$\mathbb{E}[\|\mathbf{x}_t - \mathbf{x}^*\|^2 \mathbf{1}_{D_{t-1}}|S_{t-1}] \leq \left[\|\mathbf{x}_{t-1} - \mathbf{x}^*\|^2 \left(1 - 2\eta_c\alpha + \eta_c^2\beta^2 + \frac{2L_{max}^2\beta^2}{3} - \frac{2L_{max}\eta_c\beta^2}{3}\right)\right.$$

$$\left. + \sigma^2 \left(\eta_c^2 + \frac{2L_{max}^2}{3} - \frac{2L_{max}\eta_c}{3}\right)\right]\mathbf{1}_{D_{t-1}}$$

$$\leq \left[\|\mathbf{x}_{t-1} - \mathbf{x}^*\|^2 \left(1 + \eta_c\alpha + \frac{2L_{max}\alpha}{3}\right) + \sigma^2 \left(L_{max}^2 + \frac{2L_{max}^2}{3}\right)\right]\mathbf{1}_{D_{t-1}}$$

$$\leq \left[\|\mathbf{x}_{t-1} - \mathbf{x}^*\|^2 \left(1 + L_{max}\alpha + \frac{2L_{max}\alpha}{3}\right) + \sigma^2 \left(L_{max}^2 + \frac{2L_{max}^2}{3}\right)\right]\mathbf{1}_{D_{t-1}}$$

$$= \left[\|\mathbf{x}_{t-1} - \mathbf{x}^*\|^2 \left(1 + \frac{5L_{max}\alpha}{3}\right) + \frac{5L_{max}^2\sigma^2}{3}\right]\mathbf{1}_{D_{t-1}}.$$

We use $L_{max} < \frac{\alpha}{\beta^2}$. Let $J_t = \left(1 + \frac{5\alpha L_{max}}{3}\right)^{-t} \left(\|\mathbf{x}_t - \mathbf{x}^*\|^2 + \frac{L_{max}\sigma^2}{\alpha}\right)$. We prove $J_t \mathbf{1}_{D_{t-1}}$ is a supermartingale process as follows.

$$\mathbb{E}\left[\left(1 + \frac{5\alpha L_{max}}{3}\right)^{-t} \left(\|\mathbf{x}_t - \mathbf{x}^*\|^2 + \frac{L_{max}\sigma^2}{\alpha}\right)\middle| S_{t-1}\right]\mathbf{1}_{D_{t-1}} \leq$$

$$\left(1 + \frac{5\alpha L_{max}}{3}\right)^{-t} \left[\|\mathbf{x}_{t-1} - \mathbf{x}^*\|^2 \left(1 + \frac{5L_{max}\alpha}{3}\right) + \frac{5L_{max}^2\sigma^2}{3} + \frac{L_{max}\sigma^2}{\alpha}\right]\mathbf{1}_{D_{t-1}}$$

$$= \left(1 + \frac{5\alpha L_{max}}{3}\right)^{-(t-1)} \left[\|\mathbf{x}_{t-1} - \mathbf{x}^*\|^2 + \frac{L_{max}\sigma^2}{\alpha}\right]\mathbf{1}_{D_{t-1}} = J_{t-1}\mathbf{1}_{D_{t-1}} \leq J_{t-1}\mathbf{1}_{D_{t-2}}.$$

Hence $J_t \mathbf{1}_{D_{t-1}}$ is a supermartingale. In order to use the Azuma-Hoeffding inequality, we bound $|J_t \mathbf{1}_{D_{t-1}} - \mathbb{E}[J_t \mathbf{1}_{D_{t-1}}|S_{t-1}]|$ as,

$$|J_t \mathbf{1}_{D_{t-1}} - \mathbb{E}[J_t \mathbf{1}_{D_{t-1}}|S_{t-1}]| = \left(1 + \frac{5\alpha L_{max}}{3}\right)^{-t} \left[\|\mathbf{x}_t - \mathbf{x}^*\|^2 - \mathbb{E}[\|\mathbf{x}_t - \mathbf{x}^*\|^2 |S_{t-1}]\right]\mathbf{1}_{D_{t-1}}$$

$$\leq \left(1 + \frac{5\alpha L_{max}}{3}\right)^{-t} \left[2\|\mathbf{x}_{t-1} - \eta_c\nabla f(\mathbf{x}_{t-1}) - \mathbf{x}^*\| \|\mathbf{w}_{t-1}\| + \|\mathbf{w}_{t-1}\|^2 + \right.$$

$$\sigma^2 \left(\eta_c^2 + \frac{2L_{max}^2}{3} - \frac{2L_{max}\eta_c}{3}\right) + \beta^2 \|\mathbf{x}_{t-1} - \mathbf{x}^*\|^2 \left(\frac{2L_{max}^2}{3} - \frac{2L_{max}\eta_c}{3}\right)\bigg]\mathbf{1}_{D_{t-1}},$$

$$(44)$$

where we use equation 43 in equation 44 for the term $\mathbb{E}[\|\mathbf{w}_{t-1}\|^2]$. Now, we compute $\|\mathbf{w}_{t-1}\|$ using assumption **A3** as follows.

$$\|\mathbf{w}_{t-1}\| = \|\eta_c g(\mathbf{x}_{t-1}) - \eta_c\nabla f(\mathbf{x}_{t-1}) + u_t g(\mathbf{x}_{t-1})\|$$
$$\leq \eta_c Q + |u_t|(Q + \|\nabla f(\mathbf{x}_{t-1})\|) \leq Q(\eta_c + |u_t|) + |u_t|\beta \|\mathbf{x}_{t-1} - \mathbf{x}^*\|. \qquad (45)$$

Using equation 45 in equation 44 and the bound of the event $D_{t-1}$,

$$|J_t \mathbf{1}_{D_{t-1}} - \mathbb{E}[J_t \mathbf{1}_{D_{t-1}} | S_{t-1}]|$$

$$\leq \left(1 + \frac{5\alpha L_{max}}{3}\right)^{-t} \left[ 2\|\mathbf{x}_{t-1} - \mathbf{x}^*\| \left(Q(\eta_c + |u_t|) + |u_t|\beta \|\mathbf{x}_{t-1} - \mathbf{x}^*\|\right)\right.$$

$$+ \left(Q(\eta_c + |u_t|) + |u_t|\beta \|\mathbf{x}_{t-1} - \mathbf{x}^*\|\right)^2 + \sigma^2 \left(\eta_c^2 + \frac{2L_{max}^2}{3} - \frac{2L_{max}\eta_c}{3}\right)$$

$$+ \beta^2 \|\mathbf{x}_{t-1} - \mathbf{x}^*\|^2 \left(\frac{2L_{max}^2}{3} - \frac{2L_{max}\eta_c}{3}\right)\bigg] \mathbf{1}_{D_{t-1}}$$

$$= \left(1 + \frac{5\alpha L_{max}}{3}\right)^{-t} \left[ \tilde{O}\left(\mu L_{max}^{1.5} \log^{0.5} \frac{1}{L_{max}\xi}\right) + \tilde{O}\left(\mu^2 L_{max}^2 \log \frac{1}{L_{max}\xi}\right) + 2\tilde{O}(L_{max}^2)\right.$$

$$+ \tilde{O}\left(\mu L_{max}^{2.5} \log^{0.5} \frac{1}{L_{max}\xi}\right) + 2\tilde{O}\left(\mu^2 L_{max}^3 \log \frac{1}{L_{max}\xi}\right)\bigg]$$

$$\leq \left(1 + \frac{5\alpha L_{max}}{3}\right)^{-t} \tilde{O}\left(\mu L_{max}^{1.5} \log^{0.5} \frac{1}{L_{max}\xi}\right) = d_t$$

We denote the bound of $|J_t \mathbf{1}_{D_{t-1}} - \mathbb{E}[J_t \mathbf{1}_{D_{t-1}} | S_{t-1}]|$ as $d_t$.

Let $b_t = \sqrt{\sum_{\tau=1}^t d_\tau^2} = \sqrt{\sum_{\tau=1}^t \left(1 + \frac{5\alpha L_{max}}{3}\right)^{-2\tau} \tilde{O}\left(\mu L_{max}^{1.5} \log^{0.5} \frac{1}{L_{max}\xi}\right)}$. Now,

$$\sqrt{\sum_{\tau=1}^t \left(1 + \frac{5\alpha L_{max}}{3}\right)^{-2\tau} \tilde{O}\left(\mu L_{max}^{1.5} \log^{0.5} \frac{1}{L_{max}\xi}\right)}$$

$$\leq \sqrt{\frac{1}{1 - \left(1 + \frac{5\alpha L_{max}}{3}\right)^{-2}}} \tilde{O}\left(\mu L_{max}^{1.5} \log^{0.5} \frac{1}{L_{max}\xi}\right)$$

$$= \sqrt{\frac{\tilde{O}(1)}{\tilde{O}(L_{max})}} \tilde{O}\left(\mu L_{max}^{1.5} \log^{0.5} \frac{1}{L_{max}\xi}\right) = \tilde{O}\left(\mu L_{max} \log^{0.5} \frac{1}{L_{max}\xi}\right).$$

Hence $b_t$ is of the order $\tilde{O}\left(\mu L_{max} \log^{0.5} \frac{1}{L_{max}\xi}\right)$. By the Azuma Hoeffding inequality,

$$\mathbb{P}\left(J_t \mathbf{1}_{D_{t-1}} - J_0 \geq b_t \log^{0.5} \frac{1}{L_{max}\xi}\right) \leq \exp\left(-\tilde{\Omega}\left(\log \frac{1}{L_{max}\xi}\right)\right) \leq \tilde{O}(L_{max}^3 \xi),$$

which leads to,

$$\mathbb{P}\left(J_t \mathbf{1}_{D_{t-1}} - J_0 \geq \tilde{O}\left(\mu L_{max} \log \frac{1}{L_{max}\xi}\right)\right) \leq \tilde{O}(L_{max}^3 \xi).$$

Hence we can write,

$$\mathbb{P}\left(D_{t-1} \cap \left\{\|\mathbf{x}_t - \mathbf{x}^*\|^2 \geq \tilde{O}\left(\mu L_{max} \log \frac{1}{L_{max}\xi}\right)\right\}\right) \leq \tilde{O}(L_{max}^3 \xi)$$

For some constant $\tilde{b}$ independent of $L_{max}$ and $\xi$ we can write,

$$\mathbb{P}\left(D_{t-1} \cap \left\{\|\mathbf{x}_t - \mathbf{x}^*\|^2 \geq \tilde{b}\mu L_{max} \log \frac{1}{L_{max}\xi}\right\}\right) \leq \tilde{O}(L_{max}^3 \xi)$$

By choosing $\mu < \tilde{b}$,

$$\mathbb{P}\left(D_{t-1} \cap \left\{\|\mathbf{x}_t - \mathbf{x}^*\| \geq \mu\sqrt{L_{max} \log \frac{1}{L_{max}\xi}}\right\}\right) \leq \tilde{O}(L_{max}^3 \xi)$$

$$\mathbb{P}(\bar{D}_t) = \mathbb{P}\left(D_{t-1} \cap \left\{\|\mathbf{x}_t - \mathbf{x}^*\| \geq \mu\sqrt{L_{max} \log \frac{1}{L_{max}\xi}}\right\}\right) + \mathbb{P}(\bar{D}_{t-1})$$

$$\leq \tilde{O}(L_{max}^3 \xi) + \mathbb{P}(\bar{D}_{t-1})$$

Iteratively unrolling the above equation, we obtain $\mathbb{P}(\bar{D}_t) \leq t\tilde{O}(L_{max}^3 \xi)$. Choosing $t = \tilde{O}\left(\frac{1}{L_{max}^2} \log \frac{1}{\xi}\right)$, $\mathbb{P}(\bar{D}_t) \leq \tilde{O}\left(L_{max}\xi \log \frac{1}{\xi}\right)$. As $L_{max} < \tilde{O}\left(\frac{1}{\log \frac{1}{\xi}}\right)$, $\mathbb{P}(\bar{D}_t) \leq \tilde{O}(\xi)$. $\qquad \square$

# E PROOF USING INDUCTION

In the proof of Lemma 2 in Appendix B.1, we state that equation 19 can be proved by induction for $t \geq 2$. We restate the equation here and provide the corresponding proof by induction.

$$\text{Induction hypothesis:} \quad \left\| \nabla \tilde{f}(\tilde{\mathbf{x}}_t) \right\| \leq 10\tilde{Q} \sum_{\tau=0}^{\frac{t(t-1)}{2}} (1 + \eta_c \gamma_o)^\tau. \tag{46}$$

Recollect from that equation 15 that $\nabla \tilde{f}(\tilde{\mathbf{x}}_t) = (I - \eta_c \mathbf{H}(\mathbf{x}_0)) \nabla \tilde{f}(\tilde{\mathbf{x}}_{t-1}) - \mathbf{H}(\mathbf{x}_0) \tilde{\mathbf{w}}_{t-1}$. Taking matrix induced norm on both sides,

$$\left\| \nabla \tilde{f}(\tilde{\mathbf{x}}_{t+1}) \right\| \leq (1 + \eta_c \gamma_o) \left\| \nabla \tilde{f}(\tilde{\mathbf{x}}_t) \right\| + \beta \left\| \tilde{\mathbf{w}}_t \right\|$$

$$= ((1 + \eta_c \gamma_o) + \beta |u_{t+1}|) \left\| \nabla \tilde{f}(\tilde{x}_t) \right\| + \beta \tilde{Q}(\eta_c + |u_{t+1}|), \tag{47}$$

since, $\left\| \tilde{g}(\tilde{\mathbf{x}}_t) - \nabla \tilde{f}(\tilde{\mathbf{x}}_t) \right\| \leq \tilde{Q}$. Note that $\left\| \nabla \tilde{f}(\tilde{\mathbf{x}}_t) \right\| \leq \epsilon$, $|u_t| \leq L_{max}$ and $\beta L_{max} < 1$ hold for all $t$. Therefore, at $t = 1$,

$$\left\| \nabla \tilde{f}(\tilde{\mathbf{x}}_1) \right\| \leq ((1 + \eta_c \gamma_o) + \beta |u_1|) \epsilon + \beta \tilde{Q}(\eta_c + |u_1|) \leq (1 + \eta_c \gamma_o)\epsilon + \epsilon + 2\tilde{Q}.$$

Now, we prove the hypothesis in equation 46 for $t = 2$. From equation 47, for an arbitrarily small $\epsilon$,

$$\left\| \nabla \tilde{f}(\tilde{\mathbf{x}}_2) \right\| \leq ((1 + \eta_c \gamma_o) + \beta |u_2|) \left\| \nabla \tilde{f}(\tilde{\mathbf{x}}_1) \right\| + \beta \tilde{Q}(\eta_c + |u_2|)$$

$$\leq (1 + \eta_c \gamma_o)^2 \epsilon + 2(1 + \eta_c \gamma_o)\epsilon + \epsilon + 2\tilde{Q}(1 + \eta_c \gamma_o) + 4\tilde{Q}$$

$$\leq 2\epsilon \sum_{\tau=0}^{2} (1 + \eta_c \gamma_o)^\tau + 4\tilde{Q} \sum_{\tau=0}^{1} (1 + \eta_c \gamma_o)^\tau \leq 10\tilde{Q} \sum_{\tau=0}^{\frac{2(2-1)}{2}} (1 + \eta_c \gamma_o)^\tau.$$

We have shown that the induction hypothesis holds for $t = 2$. Now, assuming that it holds for any $t$, we need to prove that it holds for $t + 1$. We know from equation 47, when the hypothesis is assumed to hold for $t$,

$$\left\| \nabla \tilde{f}(\tilde{\mathbf{x}}_{t+1}) \right\| \leq ((1 + \eta_c \gamma_o) + \beta |u_{t+1}|) 10\tilde{Q} \sum_{\tau=0}^{\frac{t(t-1)}{2}} (1 + \eta_c \gamma_o)^\tau + \beta \tilde{Q}(\eta_c + |u_{t+1}|)$$

$$\leq (1 + \eta_c \gamma_o) 10\tilde{Q} \sum_{\tau=0}^{\frac{t(t-1)}{2}} (1 + \eta_c \gamma_o)^\tau + 10\tilde{Q} \sum_{\tau=0}^{\frac{t(t-1)}{2}} (1 + \eta_c \gamma_o)^\tau + \beta \tilde{Q}(\eta_c + |u_{t+1}|)$$

$$\leq 20\tilde{Q} \sum_{\tau=0}^{\frac{t(t-1)}{2}+1} (1 + \eta_c \gamma_o)^\tau$$

If we prove $20\tilde{Q} \sum_{\tau=0}^{\frac{t(t-1)}{2}+1} (1 + \eta_c \gamma_o)^\tau \leq 10\tilde{Q} \sum_{\tau=0}^{\frac{t(t+1)}{2}} (1 + \eta_c \gamma_o)^\tau$, the induction proof is complete. Now, we need to prove

$$20\tilde{Q} \sum_{\tau=0}^{\frac{t^2-t}{2}+1} (1 + \eta_c \gamma_o)^\tau \leq 10\tilde{Q} \sum_{\tau=0}^{\frac{t^2+t}{2}} (1 + \eta_c \gamma_o)^\tau$$

$$\leq 10\tilde{Q} \sum_{\tau=0}^{\frac{t^2-t}{2}+1} (1 + \eta_c \gamma_o)^\tau + 10\tilde{Q} \sum_{\tau=\frac{t^2-t}{2}+2}^{\frac{t^2+t}{2}} (1 + \eta_c \gamma_o)^\tau.$$

Therefore we need to show that,

$$\underbrace{\sum_{\tau=0}^{\frac{t^2-t}{2}+1} (1 + \eta_c \gamma_o)^\tau}_{S_1} \leq \underbrace{\sum_{\tau=\frac{t^2-t}{2}+2}^{\frac{t^2+t}{2}} (1 + \eta_c \gamma_o)^\tau}_{S_2}. \tag{48}$$

Now, summing up the geometric series $S_1$, $\sum_{\tau=0}^{\frac{t^2-t}{2}+1}(1+\eta_c\gamma_o)^\tau = \frac{(1+\eta_c\gamma_o)^{\frac{t^2-t}{2}+2}-1}{\eta_c\gamma_o}$. Using change of variable in $S_2$ of equation 48 as $m = \tau - \left(\frac{t^2-t}{2}+2\right)$,

$$\sum_{m=0}^{t-2}(1+\eta_c\gamma_o)^{\frac{t^2-t}{2}+m+2} = (1+\eta_c\gamma_o)^{\frac{t^2-t}{2}+2}\frac{(1+\eta_c\gamma_o)^{t-1}-1}{\eta_c\gamma_o}.$$

Therefore, we now need to prove,

$$(1+\eta_c\gamma_o)^{\frac{t^2-t}{2}+2} - 1 \le (1+\eta_c\gamma_o)^{\frac{t^2-t}{2}+2}\left((1+\eta_c\gamma_o)^{t-1}-1\right)$$
$$\Rightarrow 2(1+\eta_c\gamma_o)^{\frac{t^2-t}{2}+2} \le (1+\eta_c\gamma_o)^{\frac{t^2-t}{2}+t+1} + 1 \tag{49}$$

We further prove equation 49 by induction as follows. For $t = 2$, $2(1+\eta_c\gamma_o)^3 \le (1+\eta_c\gamma_o)^4 + 1$. Let us assume the following expression holds for time step $t$.

$$2(1+\eta_c\gamma_o)^{\frac{t^2-t}{2}+2} \le (1+\eta_c\gamma_o)^{\frac{t^2-t}{2}+t+1} \tag{50}$$

Now, we prove for the time step $t+1$,

$$2(1+\eta_c\gamma_o)^{\frac{t(t+1)}{2}+2} = 2(1+\eta_c\gamma_o)^{\frac{t(t-1)}{2}+t+2} \le (1+\eta_c\gamma_o)^{\frac{t^2-t}{2}+t+1+t}$$
$$= (1+\eta_c\gamma_o)^{\frac{t(t+1)}{2}+t+1} \le (1+\eta_c\gamma_o)^{\frac{t(t+1)}{2}+t+2}, \tag{51}$$

where we use $\frac{t(t-1)}{2} + t = \frac{t(t+1)}{2}$ and apply our assumption equation 50 in equation 51. We have proved $2(1+\eta_c\gamma_o)^{\frac{t^2-t}{2}+2} \le (1+\eta_c\gamma_o)^{\frac{t^2-t}{2}+t+1} \le (1+\eta_c\gamma_o)^{\frac{t^2-t}{2}+t+1} + 1$. This concludes our proof of equation 46.

## F    Choice of parameters for other LR schedulers

1. Cosine annealing (Loshchilov & Hutter, 2017b): There are 3 parameters namely, initial restart interval, a multiplicative factor and minimum learning rate. The authors propose an initial restart interval of 1, a factor of 2 for subsequent restarts, with a minimum learning rate of $1e-4$, which we use in our comparisons.

2. Knee (Iyer et al., 2023): The total number of epochs is divided into those that correspond to the "explore" epochs and "exploit" epochs. During the explore epochs, the learning rate is kept at a constant high value, while from the beginning of the exploit epochs, it is linearly decayed. We use the suggested setting of 100 initial explore epochs with a learning rate of 0.1 followed by a linear decay for the rest of the epochs. For training ImageNet-1K, we use the suggested setting of 30 explore epochs. For fine-tuning BERT on SQuAD v1.1 dataset, we use a base learning rate of $3e-5$ for 1 explore epoch and then decay, for a total of 2 epochs. For training the Transformer model on the IWSLT'14 dataset, a seed learning rate of $3e-4$ is used for 40 explore epochs.

3. One cycle (Smith & Topin, 2019): We perform the learning rate range test for our networks as suggested by the authors. For the range test, the learning rate is gradually increased during which the training loss explodes. The learning rate at which it explodes is noted and the maximum learning rate (the learning rate at the middle of the triangular cycle) is fixed to be before that. We linearly increase the learning rate for the initial $45\%$ of the total epochs up to the maximum learning rate determined by the range test, followed by a linear decay for the next $45\%$ of the total epochs. We then decay it further up to a divisive factor of 10 for the rest of the epochs, which is the suggested setting. Note that the one cycle LR scheduler relies heavily on regularization parameters like weight decay and momentum.

4. Constant: To compare with a constant learning rate, we choose 0.05 for the VGG-16 architecture and 0.1 for the remaining architectures as done in our other baselines(Smith, 2017; Loshchilov & Hutter, 2017b).

5. Multi step: For the multi-step decay scheduler, our choice of the decay rate and time is based on the standard repositories for the architectures. [5]. Specifically, we decay the learning rate by a factor of 10 at the the epochs 100 and 150 for ResNet-110 and ResNet-50. In the case of DenseNet-40-12, we decay by a factor of 10 at the epochs 150 and 225. For VGG-16, we decay by a factor of 10 every 30 epochs. In the case of WRN, we fix a learning rate of 0.2 for the initial 60 epochs, decay it by $0.2^2$ for the next 60 epochs, and by $0.2^3$ for the rest of the epochs.

## G  TRAIN LOSS PLOTS

### G.1  PLOTS OF CIFAR-10

To study the convergence of the schedulers we plot the training loss across epochs in Figure 2. We observe that our proposed PLRS achieves one of the fastest rates of convergence in terms of the training loss compared across all the schedulers for both networks. Note that the cosine annealing scheduler records several spikes across the training.

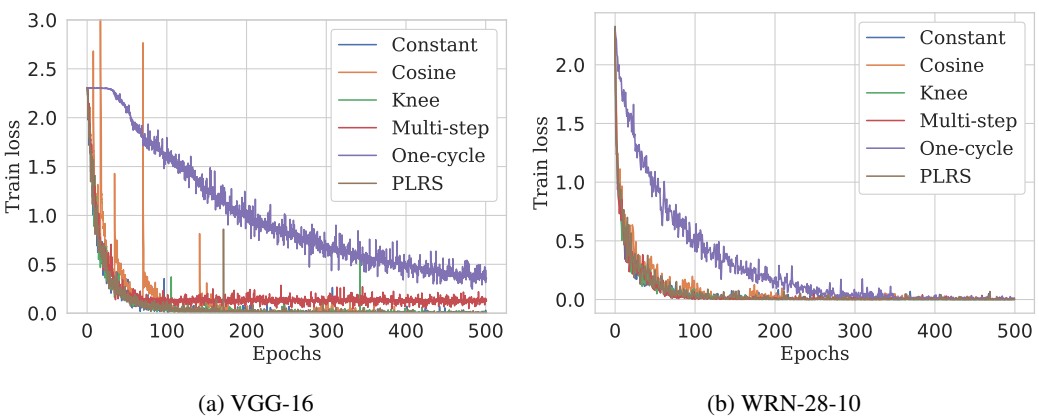

(a) VGG-16                    (b) WRN-28-10

Figure 2: Training loss vs epochs for VGG-16 and WRN-28-10 for CIFAR-10.

### G.2  PLOTS OF CIFAR-100

We plot the training loss in Figure 3. For ResNet-110, both PLRS and knee LR scheduler converge to a low training loss around 150 epochs. While cosine annealing LR scheduler also seems to converge fast, it experiences sharp spikes along the curve during the restarts. For DenseNet-40-12, PLRS converges faster to a lower training loss compared to the other schedulers. Specifically, the train loss converges around 150 and 200 epochs for ResNet-110 and DenseNet-40-12 respectively.

---

[5]ResNet:https://github.com/akamaster/pytorch_resnet_cifar10,
DenseNet:https://github.com/andreasveit/densenet-pytorch,
VGG:https://github.com/chengyangfu/pytorch-vgg-cifar10,
WRN:https://github.com/meliketoy/wide-resnet.pytorch

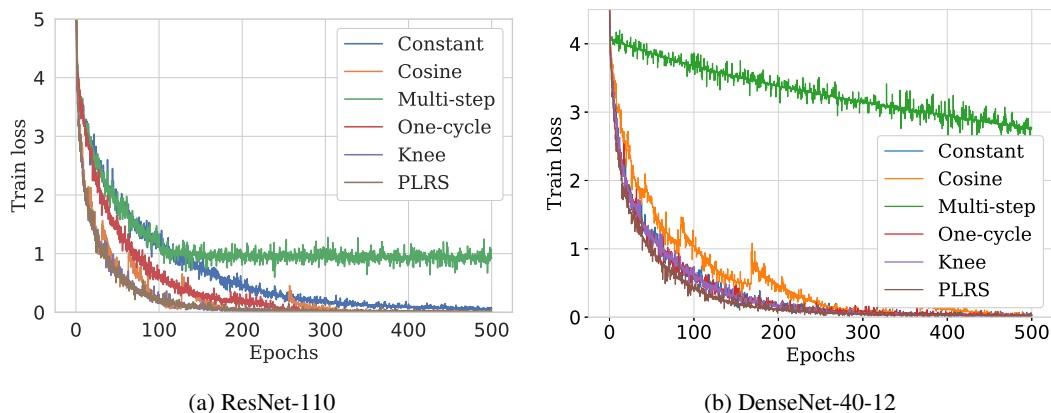

(a) ResNet-110           (b) DenseNet-40-12

Figure 3: Training loss vs epochs for ResNet-110 and DenseNet-40-12 on CIFAR-100.

## H  SENSITIVITY ANALYSIS

We perform sensitivity analysis of the parameters $L_{min}$ and $L_{max}$ on the maximum test accuracy. We vary the parameters and record the highest test accuracy achieved for various combinations of $L_{min}$ and $L_{max}$ for the WRN-28-10 network trained on the CIFAR-10 dataset and the DenseNet-40-10 network trained on the CIFAR-100 dataset respectively and give a subset of the results in Tables 6 and 7. It can be observed that over a range of combinations of $L_{min}$ and $L_{max}$, the maximum test accuracy remains $\sim 93$ for CIFAR-10 and $\sim 65$ for CIFAR-100, indicating that even if the settings of $L_{min}$ and $L_{max}$ are not tuned extensively, one can still achieve state-of-the-art results.

| $\mathbf{L_{max}}$ | $\mathbf{L_{min}}$ | Max acc. |
|---|---|---|
| 0.1 | 0.01 | 93.77 |
| 0.1 | 0.03 | 93.31 |
| 0.1 | 0.05 | 93.58 |
| 0.2 | 0.01 | 93.87 |
| 0.2 | 0.03 | 93.29 |
| 0.2 | 0.05 | 92.73 |
| 0.3 | 0.01 | 93.55 |
| 0.3 | 0.03 | 93.63 |
| 0.3 | 0.05 | 93.57 |

Table 6: Sensitivity analysis for WRN-28-10 on CIFAR-10

| $\mathbf{L_{max}}$ | $\mathbf{L_{min}}$ | Max acc. |
|---|---|---|
| 0.5 | 0.09 | 65.83 |
| 0.5 | 0.07 | 64.32 |
| 0.5 | 0.05 | 65.41 |
| 0.5 | 0.01 | 65.18 |
| 0.4 | 0.07 | 65.72 |
| 0.4 | 0.05 | 65.72 |
| 0.4 | 0.01 | 64.39 |
| 0.3 | 0.03 | 64.39 |
| 0.3 | 0.01 | 64.94 |

Table 7: Sensitivity analysis for DenseNet-40-10 on CIFAR-100.

## I  ONLINE TENSOR DECOMPOSITION

We follow the experimental setup in (Ge et al., 2015), where their proposed projected noisy gradient descent is applied to orthogonal tensor decomposition. A brief description of the online tensor decomposition problem is given below.

Consider a tensor $T$ which has an orthogonal decomposition,

$$T = \sum_{i=1}^{d} a_i^{\otimes 4}, \tag{52}$$

where $a_i$'s are orthonormal vectors. The goal of performing the tensor decomposition is to find the orthonormal components, given the tensor. The objective function is defined to reduce the correlation between the components:

$$\min_{\forall i, \|u_i\|=1} \sum_{i \neq j} T(u_i, u_i, u_j, u_j) \tag{53}$$

We plot the normalized reconstruction error, $\left\| T - \sum_{i=1}^{d} u_i^{\otimes 4} \right\|_F^2 / \|T\|_F^2$ in Figure 4, where $\|.\|_F$ denotes the Frobenius norm.

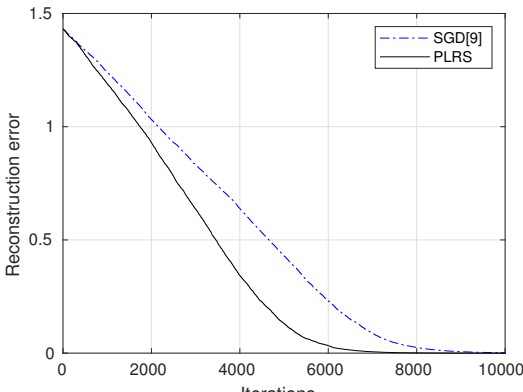

Figure 4: Reconstruction error for online tensor decomposition

We tune the learning rate parameters $L_{min}$ and $L_{max}$ to 0.007 and 0.01 respectively to obtain the convergence plot with PLRS. We compare against the plot in Figure 1.a of (Ge et al., 2015). We note that the proposed Uniform LR produces faster and smoother convergence when compared to the unit sphere noise proposed in the Noisy SGD algorithm. As mentioned in (Ge et al., 2015), the plot may vary depending on the instance of initialization; however, it converges consistently across all runs.

Additionally, we implemented stochastic gradient descent with additive noise in the neural network setting. However, its performance was suboptimal even with extensive tuning of hyperparameters.

## J  LLM USAGE

We make use of LLMs for grammar, punctuation and phrasing suggestions.

