# OpenReview forum: "Randomness Helps Rigor: A Probabilistic Learning Rate Scheduler Bridging Theory and Deep Learning Practice"
_ICLR.cc/2026/Conference — Submitted to ICLR 2026_

### Official Review · Reviewer_eUrV · 2025-10-25

**Soundness:** 3
**Presentation:** 3
**Contribution:** 2
**Rating:** 4
**Confidence:** 3

**Summary:**

Naive/stochastic gradient descent uses a fixed learning rate eta. Convergence to a local minimum is not guaranteed with a fixed learning rate. If the learning rate is too small, naive/stochastic gradient descent can get stuck in stationary points that are not local minima. If the learning rate is too large, convergence is not guaranteed. Typical approaches to address this are learning rate schedules that alternate between large and small values for eta. Past methods are deterministic. This paper proposes a stochastic method where the learning rate is sampled from a uniform distribution in [l, h]. Conditions are given, along with a proof, where convergence to a local minimum is guaranteed. Experiments are provided giving empirical evidence of the value of the new method.

This is outside my area of expertise. Thus, I am not familiar with the literature on the topic. I leave it to other reviewers to assess whether this is novel in the context of relevant literature. I am also not a math wizard. Thus, I did not slog through the proofs. I don't believe that is necessary given my comments below. Nonetheless, I trust the authors on the validity of the proofs. If there are errors (and I don't know whether there are or claim that there are), they must be fixable, as I believe that the general intuition is sound. I leave it to others to check the proofs.

**Strengths:**

I believe that the general intuition/idea of randomly choosing eta, instead of having a deterministic schedule, can lead to convergence guarantees. Generally, randomized algorithms have demonstrated similar benefits over deterministic algorithms for many problems.

The field of deep learning has lots of aspects that are black art, where people use general intuition to solve problems, rather than methods with provable properties. Choosing the optimization method and its associate hyperparameters is one of them. The field would greatly benefit from replacing this black art with methods that offer guarantees .

Thus, I like and encourage this work, subject to my concerns below.

**Weaknesses:**

1. This is reminiscent to simulated annealing in many ways. While simulated annealing attempts to find global optima and this is to provide convergence guarantees to local optima, they are otherwise similar. This should be discussed and relevant results brought to bear.
2. Why sample eta from a uniform distribution? It would seem to me that, generally, to guarantee convergence, you want an eta at the low end of the distribution. You only need it rarely at the high end to avoid stationary points that are not local minima. So why not sample from a long tailed monotonically decreasing distribution, such as an exponential? That would seem to me to be sufficient to guarantee convergence and may be more general and faster. An interesting question would be what kind of distribution is necessary to guarantee convergence and what the relationship would be between that distribution and speed of convergence.
3. All optimization methods have some hyperparameters. This one too. For any method (this one too), convergence is guaranteed (and in fact only happens) only for certain hyperparameter values. So, from a practical perspective, no advantage is demonstrated of this method over any other. What one needs to show is some empirical evidence of the following form: it is easier or less error prone to select the hyperparameters for this approach than any others. If a nonexpert user arbitrarily selects l and h are they more likely to just stumble upon ones that lead to (fast) convergence. Are there easy rules of thumb that one can use to select l and h that lead to (fast) convergence.
4. The empirical experiments don't add much to the paper without (3).
5. In the black art of deep learning, convergence is not necessarily important or even desirable. Things like early stopping explicitly choose to not seek convergence in order to avoid overfitting and get better generalization. What you really want is some sort of guarantee on solving the underlying problem.

**Questions:**

None.

---

> ### Author Response · Authors · 2025-11-16
> **Response to reviewer eUrV**
>
> We thank the reviewer for their efforts spent in reviewing our paper. We address their concerns below pointwise.
>
> W1: PLRS and simulated annealing (SA) both introduce randomness to help escape saddle regions of the loss landscape, but they do so in fundamentally different ways. PLRS perturbs the update by randomly scaling the stochastic gradient with an i.i.d. learning rate variable, resulting in multiplicative noise that preserves the structure of SGD. In contrast, simulated annealing perturbs the iterate itself, independent of the gradient, based on Markov chain sampling.
>
> SA relies on a decreasing temperature schedule to guarantee global convergence, while PLRS uses the randomness from non-decreasing probabilistic learning rate schedule for which we provide saddle point escape guarantees.
> Further, the randomness that is introduced in PLRS does not change the direction of the iterate since it is a scalar random variable, while in SA the direction is also randomized. Hence, we had not compared with SA, as it does not fit in the framework of SGD.
>
> W2: We thank the reviewer for raising this thoughtful point. Indeed, there is substantial recent evidence in machine learning that periodically varying the learning rate, **alternating between large and small values**, can accelerate training and improve both optimization and generalization. This includes cyclical learning rates (Smith, 2017) [1] and super-convergence (Smith & Topin, 2019) [2], both of which explicitly rely on intermittent use of relatively large learning rates. Motivated by this trend, our goal in introducing PLRS was to provide **convergence guarantees for optimization dynamics that allow both small and large learning rates to be selected stochastically**, thereby capturing a broad generalization of such cyclical schedules.
>
> For this reason, we focused on the simplest non-degenerate distribution; a uniform distribution, as a clean way to mathematically model the stochastic alternation between low and high learning rates. Investigating long-tailed or monotonically decaying distributions (e.g., exponential) is an interesting direction, but it is non-trivial: the technical challenges in our proofs arise precisely from the coupling between multiplicative noise and the gradient, and extending these arguments to heavy-tailed would require substantial new analysis. We therefore view the design of such distributions and their relationship to convergence speed as an important direction for future work, and we appreciate the reviewer for highlighting it.
>
> W3: We thank the reviewer for this valuable comment. In practice, we select $(\ell, h)$ (i.e., $⁡L_{\min}, L_{\max}$​) using the standard **learning rate range test**, as described in Section 5 under Hyperparameter Tuning. This provides a simple and well-established procedure for choosing the learning-rate interval.
>
> Moreover, even when the user selects different values, our **sensitivity analysis** (Section 5.1, Sensitivity Analysis) demonstrates that the resulting maximum test accuracy varies only mildly. We point out one relevant result from the paper: for WRN trained on the CIFAR-10 dataset, we consider multiple varying combinations of hyperparameters (without tuning) that a nonexpert might choose. We observe that the standard deviation of the accuracy obtained with these different combinations is only 0.47. This empirically shows that our method is robust to the choice of learning-rate bounds, and that PLRS remains stable and effective across a wide range of hyperparameter settings even without performing the range test.
>
> W5: Convergence to an $\epsilon$-second order stationary point does mean that we are closer to reaching our objective of a lower functional value, the function being the loss value in neural networks. Further, it is empirically observed that the proposed PLRS schedule with convergence guarantees reaches a better test accuracy in many experiments performed, which will not be the case in overfitting regimes.
>
> We provide a convergence guarantee not to run to full convergence, but to show the proposed scheduler is stable, principled, and not prone to divergence. Providing such guarantees is standard in learning-rate scheduler work and avoids relying solely on heuristics and helps us position our paper on theoretical footing.
>
> Our analysis characterizes the algorithm’s behavior if run longer, but does not prescribe a stopping rule. Therefore it is compatible with more practice-oriented approaches such as Early stopping and other generalization-oriented practices.
>
> References:
>
> [1] Leslie N Smith. Cyclical learning rates for training neural networks. In 2017 IEEE winter conference on applications of computer vision, pp. 464–472, 2017.
>
> [2] Leslie N Smith and Nicholay Topin. Super-convergence: Very fast training of neural networks using large learning rates. In Artificial intelligence and machine learning for multi-domain operations applications, pp. 369–386, 2019.

---

### Official Review · Reviewer_FLZt · 2025-10-30

**Soundness:** 2
**Presentation:** 2
**Contribution:** 3
**Rating:** 4
**Confidence:** 2

**Summary:**

The paper proposes a randomized learning rate (LR) scheduler that modulates (multiplies) learning rates by uniform random noise in order to improve convergence in theory and practice. The paper consists of a theoretical part, where convergence properties are examined under assumptions on smoothness, bounded values and other analytical assumptions, and a practical part, where experiments are conducted that compare the generalization performance reached after a certain amount of training to competing LR-scheduling schemes (such as the famous 1-cycle). The experiments show favorable results for the new method, as it mostly outperforms the competition.

**Strengths:**

The paper proposes a new LR-scheduling method that is easy to implement and seems to provide some practical gains over commonly practice scheduling techniques, which is impressive given the overall amount of research that has been conducted to put these schemes into popularity. As far as I was able to check, the results reported for the "smaller" benchmarks like CIFAR-X are close to actual SOTA for the architectures (for ImageNet, I am not sure, see below).
The paper also makes some effort to provide theoretical justifications for convergence guarantees (in particular, concerning avoidance of saddle points).
The method also seems to have some merits from an intuitive point of view: Noisy gradients can help with generalization in multiple ways (batch noise creates drift away from non-generalizing local minima, gradient noise can affect margins), so, at least when looked at it from the surface, randomizing step sizes could help in some way.

**Weaknesses:**

I should first say that I have not actively worked on convergence analysis for any such numerical scheme; so my opinion is not an expert one. That said, I found the theoretical part less convincing than the practical results.
First of all, a number of strong assumptions have to be made, many of which are not met in the practical examples (ReLU networks employed in VGG or ResNet-Variants are only continuous but do not have any higher-order smoothness properties, and up to the classification layers, rectifier networks are not bounded)
It is also my understanding that practical networks are not strongly convex in the stated sense for typical training data but contain degnerate "valleys" of constant, locally optimal training loss (which seems to be somewhat related to why they generalize so well). I would assume that stronger assumptions are just a common modeling tool used in related literature; so this might still be a useful modeling assumption even if not fully realistic. In that case, it would be nice to position this pronouncedly as such.
Then, in the proof sketches (I have not tried to verify the actual proofs in the appendix), it seems to me that these assumptions play a critical role; in particular, in my (maybe superficial understanding) saddle points are escaped because if there is one direction left with an escape gradient, smoothness transfers this into the neighborhood of other points and directions (otherwise, the Hessian-level smoothness would not hold). However, this mechanism could and would not work at all in a non-smooth network with ReLUs; so the model seems to be insufficient to explain the practical success of the method. I was also not able to see where the main idea - randomization - actually brings in its benefits; wouldn't the same arguments also hold for a deterministic scheme with a small but fixed LR?

I would not be surprised that my understaning ist rather incomplete due to limited prior knowledge on these kind of issues, but it would be very helpful to end Section 3 (Theory) with one or two paragraphs tying the theoretical models more closely to effects that one would expect to see and help in a practical scenario.

Finally, concerning the theory part, I had also some issues with the formulation of some of the definitions. It seems to me that some of those only make sense with some implicit assumptions added that are not stated in the text:

Definition 4 seems to define a saddle point as a point with a negative eigenvalue in the Hessian (due to the constants being arbitrary, no other restrictions would apply for a twice-differentiable function). I would guess that the existence of positive eigenvalues is implied, and constants will later be looked at and judged.

Definition 6: f posses the strict saddle property at all x if x fullfills... Probably that should be f? But then, how can x be close to a local minimum when we are talking about the whole function? I could not find an interpretation of the text that seemed useful to me.

In the practical part, I found the results encouraging, but I was a bit surprised about the ImageNet 1-K performance: The original 2016 ResNet Paper by He reports 5.25% top-5 and 20.74% top-1 error, clearly better than Knee and PLRS. While the 2016 paper might have used more epochs, it is still worth discussing SOTA for ImageNet (I consider this benchmark important as it is more realistic; obviously, computational costs might be prohibitive for parameter tuning so that a lower-level baseline is all one can get, but this should be discussed/explained).

Overall, there seems to be some potential, and the parts I am critical of are in an area where I am not very experienced; due to open questions on my side I would for now maintain a slightly skeptical rating at rather low confidence.

**Questions:**

I had some questions about the theoretical part (see above). The most important aspect for me would be to understand more clearly "why" the method should work. Can you connect the formal analysis to actual advantages in a (realistic model of a) practical scenario? In particular, why is the specific type of randomization so useful that it can beat all the other methods?

---

> ### Author Response · Authors · 2025-11-18
> **Response to reviewer FLZt**
>
> **Part 1/2**
>
> We thank the reviewer for the thoughtful and detailed feedback. We address the concerns below.
>
> **Assumptions vs. Practical Networks**
>
> We acknowledge the gap between our theoretical assumptions and practical neural architectures such as ReLU-based VGG/ResNet models, which lack higher-order smoothness and boundedness. This gap, however, is common across the entire literature on analyzing SGD escape from saddle points, including Jin et al. (2017), Ge et al. (2015), Allen-Zhu & Li (2018), and Daneshmand et al. (2018) [1,2,3,4]; all of which make similar smoothness and regularity assumptions to obtain provable guarantees, showing results on neural networks [4] even though theoretical analysis is constrained with such assumptions.
> Our aim is not to theoretically model deep network optimization exactly, but to study a broader function class for which meaningful analysis is mathematically possible. Despite the theoretical mismatch, our empirical results consistently show strong performance, suggesting that the mechanism is not constrained by the theoretical assumptions.
>
> **Local Strong Convexity**
>
> The reviewer notes that practical networks are not strongly convex. Our analysis does not assume global strong convexity. In Theorem 3, we only require **local strong convexity in a small $\delta$-neighborhood** around a local minimum; an assumption standard in non-convex optimization to ensure that, after escaping saddle points, the iterates remain in the basin of attraction with high probability.
>
> **Role of Randomization**
>
> Classical convergence guarantees typically rely on a small fixed learning rate or a decaying schedule. However, recent empirical successes, such as cyclical schedules (Smith, 2017) [5] and warm restarts (Loshchilov & Hutter, 2017) [6], show that alternating between large and small learning rates often outperforms both fixed and decaying schemes, though no theoretical explanations exist.
> Our stochastic learning-rate scheme models this behavior while enabling rigorous probabilistic analysis. Randomization is essential: it provides a non-degenerate noise source (nonzero variance, finite moments) that allows us to derive high-probability guarantees that deterministic alternating schemes cannot currently provide. Exploring other distributions (e.g., heavy-tailed) is an interesting direction for future work.
>
> **Clarifications on Definitions**
>
> Definition 4: It follows the standard characterization of saddle points used in non-convex optimization, combining (i) a small gradient norm with (ii) negative curvature. This excludes both local minima (non-negative Hessian) and non-stationary points (large gradient).
>
> Definition 6: The reviewer is correct; the strict saddle property is a property of the function f, not the point x. We will revise the phrasing accordingly to reflect this better.
>
> **ImageNet-1K Performance**
>
> The original ResNet paper (He et al., 2016) [7] trains for $60\times 10^4$ iterations (~120 epochs), which is **twice** the training budget used in our experiments. This difference largely explains their superior top-1 and top-5 accuracy, as longer training is well known to improve final performance.
>
> **Practical Implications of the Theory**
>
> As suggested, we will add a short discussion at the end of Section 3 explaining how the theoretical behavior (e.g., increased probability of escaping saddle points under multiplicative randomness) helps interpret the empirical improvements observed in deep learning scenarios. We shall add the paragraph below:
>
> *Our theoretical results show that introducing multiplicative randomness into the learning rate through a probabilistic LR scheduler increases the probability of escaping saddle points. In practical deep networks, where such regions are common, this behavior provides a plausible explanation for the observed empirical gains: the random fluctuations help the optimizer explore more effectively and avoid saddle points. Empirical evidence supports our theoretical findings, demonstrating comparable or better performance when compared against the baseline LR schedulers.*
>
> References:
>
> [1] Jin, C., Ge, R., Netrapalli, P., Kakade, S. M., & Jordan, M. I. (2017, July). How to escape saddle points efficiently. In International conference on machine learning (pp. 1724-1732). PMLR.
>
> [2] Rong Ge, Furong Huang, Chi Jin, and Yang Yuan. Escaping from saddle points—online stochastic gradient for tensor decomposition. In COLT, 2015.
>
> [3] Allen-Zhu, Z., & Li, Y. (2018). Neon2: Finding local minima via first-order oracles. Advances in Neural Information Processing Systems, 31.
>
> [4] Daneshmand, H., Kohler, J., Lucchi, A., & Hofmann, T. (2018, July). Escaping saddles with stochastic gradients. In International Conference on Machine Learning (pp. 1155-1164). PMLR.
>
> [5] Leslie N Smith. Cyclical learning rates for training neural networks. In 2017 IEEE winter conference on applications of computer vision, pp. 464–472, 2017.

---

> > ### Author Response · Authors · 2025-11-18
> > **Response to reviewer FLZt**
> >
> > **Part 2/2**
> >
> > [6] Loshchilov, I., & Hutter, F. (2017). SGDR: Stochastic Gradient Descent with Warm Restarts. International Conference on Learning Representations (ICLR 2017). (arXiv:1608.03983).
> >
> > [7] He, K., Zhang, X., Ren, S., & Sun, J. (2016). Deep residual learning for image recognition. In Proceedings of the IEEE conference on computer vision and pattern recognition (pp. 770-778).

---

> > > ### Comment · Reviewer_FLZt · 2025-11-26
> > >
> > > Dear Authors,
> > >
> > > thanks for the feedback. It now seems to me even more so that the theoretical part is central to the contribution of this submission. Thus, I would emphasize at this point that the reviews of colleagues with a more in-depth knowledge in that area should receive more weight.
> > >
> > > - Rev. FLZt

---

### Official Review · Reviewer_oxWh · 2025-11-01

**Soundness:** 3
**Presentation:** 3
**Contribution:** 3
**Rating:** 6
**Confidence:** 3

**Summary:**

The paper addresses the theory–practice gap that arises because modern learning‑rate (LR) schedulers used in deep learning are non‑monotone, while classical analyses assume constant/decaying LR. Proposed method: a probabilistic LR scheduler (PLRS) that samples the LR each step from [Lmin, Lmax], equivalently constant‑step GD with zero‑mean noise whose scheduler component is multiplicative. Under standard smoothness/strict‑saddle/local‑convexity assumptions, the analysis provides three results: expected descent for large gradients, finite‑time escape from strict saddles, and stability near local minima over a time window. Empirically, PLRS is validated across vision (CIFAR‑10/100, Tiny‑ImageNet, ImageNet‑1K), NLP (SQuAD v1.1, IWSLT’14), and ASR (Whisper‑small on CommonVoice Hindi), alongside cosine/one‑cycle/knee/constant/multi‑step.

**Strengths:**

1. Drop‑in simplicity: Two bounds (Lmin, Lmax), no handcrafted schedule shapes, selection via the standard LR range test.
2. Stability‑focused diagnostics: Loss‑curve plots, (Lmin, Lmax) sweeps under a single explicit recipe, interpretable stability signals.
3. Theoretical advance for non‑monotone LR: Under standard assumptions, unified results for large‑gradient descent, finite‑time saddle escape, and local‑minimum stability.

**Weaknesses:**

1. Representativeness/baselines: ImageNet‑1K uses 60 epochs and disables momentum/weight decay; the main table there compares only to knee under that recipe.
2. Hard to operationalize: Conditions like Lmax < 1/β and the analysis assumption ut ⟂ g(xt); Õ(·) hides constants, limiting concrete guidance.
3. Ablations/reporting gaps: No systematic tests of sampling distribution/frequency or optimizer interactions; DenseNet‑40‑12/40‑10 label mismatch; “Baseline” unlabeled in Tables 4–5; ASR std ≈ 0.0002 needs verification.
4. Missing important related studies, such as [1] and [2].

[1] Smith, Samuel L., et al. "Don't decay the learning rate, increase the batch size." arXiv preprint arXiv:1711.00489 (2017).

[2] Wu, Yanzhao, et al. "Demystifying learning rate policies for high accuracy training of deep neural networks." 2019 IEEE International conference on big data (Big Data). IEEE, 2019.

**Questions:**

1. Beyond the range test, what conservative, repeatable rule‑of‑thumb (and instability signals/fallback) do you recommend for picking bounds (Lmin, Lmax)?
2. Will you add results and default ranges for SGD+momentum+WD / AdamW+WD under mainstream schedules?

---

> ### Author Response · Authors · 2025-11-20
> **Response to reviewer oxWh**
>
> **Part (1/2)**
>
> We thank the reviewer for their thoughtful comments on our paper. We address their concerns below.
>
> W1: Thank you for raising this point. The Knee LR scheduler of Iyer et al. (2023) [1] has already been extensively benchmarked across many schedulers, models, and datasets. Since our ImageNet-1K experiments use the same architecture (ResNet-50), we follow their setup and compare PLRS directly against Knee LR.
>
> W2: The constraint, $L_{\max}\leq 1/\beta$,  is standard in optimization literature even for convex problems; for example, Nesterov (2003) [4] also requires the learning rate to be smaller than the inverse Lipschitz constant for convergence. We also would like to add that this constraint is not very limiting; gradient descent step sizes naturally depend on how quickly the functional gradient varies, and conditions of this sort capture that dependence. In fact, most stable step-size rules implicitly enforce comparable constraints to ensure that updates remain well-behaved relative to the problem's curvature. Regarding the $\tilde{O}$ notation, we hide certain parameter dependencies to keep the analysis tractable; otherwise, deriving high-probability guarantees becomes intractable. We agree that this limits practical interpretability and view clarifying these dependencies as important future work.
>
> W3: Thank you for the detailed comments. We will correct the table label mismatch and also make the baseline explicit in the tables. The ASR standard deviation is reproducible using the released evaluation code.
>  We have additionally tested our scheduler with SGD, Adam, and AdamW. The uniform sampling distribution was chosen because it allows tractable convergence analysis, and we provide empirical results under this choice. Extending both the theory and experiments to other distributions is a promising area for future work.
>
> W4: We thank the reviewer for pointing out these related works. We will add the suggested references and incorporate a discussion clarifying how our approach differs from these contributions as detailed below.
> [2] studies a specific relationship between batch size and learning rate, which lies outside our focus; hence, we do not include experiments involving increased batch sizes with correspondingly reduced learning rates. Our scheduler also differs from the one in [2] in that it is not a decaying schedule.
>  [3] benchmarks cyclic and step-decay schedulers on MNIST and CIFAR-10 using LeNet, CNN3, and ResNet-32. Our experiments cover a broader range of datasets, including CIFAR-10/100, ImageNet-1K, SQuAD v1.1, IWSLT, and CommonVoice Hindi, as well as a wider set of architectures, such as VGG-16, DenseNet-40-12, ResNet-110, WRN-28-10, BERT, and Transformer encoder-decoder models. We also compare against more schedulers (constant LR, multi-step, one-cycle, cyclic, cosine, and ours). Thus, our benchmark is more comprehensive than that of [3].
>
> Q1: Beyond the LR range test, a practical rule is to choose $L_{\min}$ and $⁡L_{\max}$​ close to the constant learning rate typically used for the architecture–dataset pair. For BERT fine-tuning, the usual constant LR is $3\times10^{-5}$; we found $L_{\min}=2\times10^{-5}$ and $L_{\max}=3\times10^{-5}$ to work best.
>
> Q2: Yes, we will add results with momentum and weight decay in the paper. We thank the reviewer for pointing it out.
>
>  **VGG-16 Test Accuracy on CIFAR-10 (SGD + momentum + WD)**
>
> | Scheduler              | Test Accuracy (%) |
> |--------------------------|-------------------|
> | Multi-step Decay   | 88.39             |
> | Cyclic LR              | 90.45             |
> | Cosine Annealing | 91.34             |
> | One-cycle LR        | 88.10             |
> | PLRS                    | 90.34             |
> | Knee LR               | 89.68             |
>
> Note that although the cosine scheduler achieves the highest accuracy, it requires extensive tuning to achieve this. Its performance was extremely sensitive to hyperparameter choices. When we applied momentum and weight decay using the same settings as in the no-momentum case, the accuracy collapsed to roughly 10%. Moreover, the training dynamics under cosine scheduling exhibited pronounced ripples, indicating instability.
> For PLRS, we use $L_{\min}=0.07$ and $L_{\max}=0.10$. Parameters for other schedulers are tuned for their best accuracy.
> For AdamW and WD on SQuAD v1.1 [1], a thorough comparison already exists, showing Knee LR as the strongest baseline; the same holds for our other AdamW and WD experiments. We therefore do not repeat those results.

---

> > ### Author Response · Authors · 2025-11-20
> > **Response to reviewer oxWh**
> >
> > **Part (2/2)**
> >
> > References:
> >
> > [1] Iyer, N., Thejas, V., Kwatra, N., Ramjee, R., & Sivathanu, M. (2023). Wide-minima density hypothesis and the explore-exploit learning rate schedule. Journal of Machine Learning Research, 24(65), 1-37.
> >
> > [2] Smith, Samuel L., et al. "Don't decay the learning rate, increase the batch size." arXiv preprint arXiv:1711.00489 (2017).
> >
> > [3] Wu, Yanzhao, et al. "Demystifying learning rate policies for high accuracy training of deep neural networks." 2019 IEEE International conference on big data (Big Data). IEEE, 2019.
> >
> > [4] Y. Nesterov, Introductory Lectures on Convex Optimization: A Basic Course. Boston, MA, USA: Kluwer Academic Publishers, 2003.

---

### Official Review · Reviewer_4MQU · 2025-11-02

**Soundness:** 2
**Presentation:** 2
**Contribution:** 1
**Rating:** 2
**Confidence:** 3

**Summary:**

This paper proposes a probabilistic learning rate scheduler (PLRS) for SGD: at each step, the LR is sampled uniformly from the interval $[L_{\min}, L_{\max}]$ and decomposed as $\eta_t = \eta_c + u_t$ where $\eta_c$ is just the avg of $L_{\min}$ and $L_{\max}$. This allows us to interpret the update as GD with multiplicative noise. The authors prove: (i) expected descent when (|\nabla f|) is large (Theorem 1), (ii) high‑probability escape from strict saddles within (T=\tilde O(L_{\max}^{-1/4})) steps (Theorem 2), and (iii) stability near a local minimum for (T=\tilde O(L_{\max}^{-2}\log(1/\xi))) steps (Theorem 3), under standard L-smoothness, Hessian‑Lipschitz, bounded gradient noise, the strict‑saddle property, boundedness of function value, and local strong convexity assumptions. The authors conduct small-scale experiments on CIFAR10/Tiny‑ImageNet/ImageNet, SQuAD, IWSLT’14, and CommonVoice that suggest PLRS might be competitive with cosine, one‑cycle, knee, multi‑step, and constant schedules.

**Strengths:**

1. The paper's theoretical analysis is written clearly and organized around the three regimes ((large gradient; small gradient/negative curvature; neighborhood of a local minimum).
2. Empirical section is broad, covering vision, NLP, and ASR, with reasonable baselines and ablations on $L_{\min}, L_{\max}$.
3. The problem of learning rate scheduling has attracted a lot of attention recently and is quite well-motivated, e.g. as in Defazio et al. 2024.

Defazio, A., Yang, X., Mehta, H., Mishchenko, K., Khaled, A., & Cutkosky, A. (2024). The road less scheduled. Advances in Neural Information Processing Systems, 37, 9974-10007.

**Weaknesses:**

1. The paper states that although cyclic LRs lack theory, they were “shown to be a valid hypothesis owing to the presence of many saddle points (Dauphin et al., 2014).” but as far as I can tell Dauphin et al. identify saddle‑point pathology but they do not validate the cyclic‑LR hypothesis.
2. The paper claims to be “the first to theoretically prove convergence of SGD with a LR scheduler that does not conform to constant or monotonically decreasing rates” (Sec. 1.2). However, non‑monotone step‑size policies do have some prior theory! For example, the Distance over Gradients (DoG) algorithm of (Ivgi et al., 2023) both increases and decreases the learning rate.
3. Multiplicative-noise SGD has also been analyzed in existing work (Chen et al., 2025; Jofré & Thompson, 2019, Faw et al., 2023). Beyond establishing saddle point escape, I'm not really sure what's particularly difficult in extending prior analysis here.
4. The manuscript refers to “cosine‑based cyclic LR scheduler”. But the standard cosine schedule (even with warm restarts) is a decay one and not really cyclic.. Warm restarts change the phase and amplitude, so successive “cycles” do not repeat at the same magnitude (Loshchilov & Hutter 2017).
5. There is a big difference between "flactuating" learning rates in practice (triangular, or cyclical) and the i.i.d. learning rates drawn from a uniform distribution here. The paper mentions "trustworthy AI" as a motivation and to not treat optimizers as black boxes, but I am not exactly sure how learning rates drawn uniformly from a certain range help do that, especially when we *do* have analysis for why some learning rate schedules work well already and a theoretically-motivated adaptive learning rate scheme that can increase the lr (Defazio et al., 2023).

Refs:
Dauphin, Y. N., Pascanu, R., Gulcehre, C., Cho, K., Ganguli, S., & Bengio, Y. (2014). Identifying and attacking the saddle point problem in high-dimensional non-convex optimization. Advances in Neural Information Processing Systems (NeurIPS 27), 2933–2941.
Ivgi, M., Hinder, O., & Carmon, Y. (2023). DoG is SGD’s Best Friend: A Parameter‑Free Dynamic Step Size Schedule. arXiv:2302.12022.
Chen, Z., Maguluri, S. T., & Zubeldia, M. (2025). Concentration of Contractive Stochastic Approximation: Additive and Multiplicative Noise. The Annals of Applied Probability, 35(2), 1298–1352. https://doi.org/10.1214/24-AAP2143
Jofré, A., & Thompson, P. (2019). On variance reduction for stochastic smooth convex optimization with multiplicative noise. Mathematical Programming, 174(1), 253–292. https://doi.org/10.1007/s10107-018-1297-x
Faw, M., Rout, L., Caramanis, C., & Shakkottai, S. (2023). Beyond Uniform Smoothness: A Stopped Analysis of Adaptive SGD. Proceedings of the 36th Annual Conference on Learning Theory (COLT 2023), PMLR 195:1–72.
Loshchilov, I., & Hutter, F. (2017). SGDR: Stochastic Gradient Descent with Warm Restarts. International Conference on Learning Representations (ICLR 2017). (arXiv:1608.03983).
Defazio, A., Cutkosky, A., Mehta, H., & Mishchenko, K. (2023). Optimal linear decay learning rate schedules and further refinements. arXiv preprint arXiv:2310.07831.

**Questions:**

1. Please address my concerns in the weaknesses section.
2. Beyond tracking the multiplicative updates, what is the main technical novelty here compared to Jin et al. (2017)?
3. Your introduction emphasizes “fluctuating” learning rates in practice (triangular, one‑cycle, cosine), but the proofs assume IID LR draws in $[L_{\min},L_{\max}$. What breaks if the LR is flacuated according to a pre-set schedule or if the draws are correlated?

Refs:
Jin, C., Ge, R., Netrapalli, P., Kakade, S. M., & Jordan, M. I. (2017). How to Escape Saddle Points Efficiently. Proceedings of the 34th International Conference on Machine Learning (ICML 2017), PMLR 70:1724–1732. (arXiv:1703.00887).

---

> ### Author Response · Authors · 2025-11-14
> **Response to Reviewer 4MQU**
>
> **Part 1 of the response (1/2)**
>
> We thank the reviewer for their careful reading of our paper and for the valuable comments and references provided. We appreciate the thoughtful feedback and the time taken to engage deeply with our work. We now address their concerns pointwise. We refer to Point 1 under “Weaknesses” as W1 and so on. Similarly, we refer to Point 1 under “Questions” as Q1.
>
> W1. Thank you for the clarification. We agree that Dauphin et al. (2014) [7] identify saddle-point pathology but do not validate the cyclic-LR hypothesis. Our intent was only to cite them for the prevalence of saddle points. We will revise the wording to make this clear.
>
> W2. While DoG [1] indeed allows both increases and decreases in the learning rate, its convergence theory relies on **convexity assumptions** (refer to lines 1-2 under Section 1.1 of [1]). In contrast, our results provide convergence guarantees for **non-convex objectives**, where proving convergence for non-monotone learning rates is significantly more challenging.
>
> W3. We thank the reviewer for highlighting prior work on multiplicative-noise SGD. While these works (Chen et al., 2025; Jofré & Thompson, 2019; Faw et al., 2023) [2,3,4] analyze first-order stationary points, **they do not address convergence to second-order points or escape from saddle points** (as the reviewer correctly points out), which is a central contribution of our paper. We highlight that establishing saddle-point escape in non-convex SGD with non-monotone learning rates is highly non-trivial; see the contrast between Nesterov (2003) [11] and Netrapalli et al. (2019) [12] on the significant technical challenges involved.
> Moreover, the definitions and assumptions of multiplicative noise in prior works differ substantially from ours:
> * (Chen et al., 2025) [2] assume Lipschitz smooth stochastic gradients and focus on linear stochastic approximation and reinforcement learning, not standard SGD.
> * (Jofré & Thompson, 2019) [3] assume smooth convex objectives with convex regularizers (refer to lines 2-3 in the Abstract of [3]) and require Lipschitz stochastic gradients.
> * (Faw et al., 2023) [4] use an affine multiplicative-noise formulation and assume (L_0, L_1)-smoothness rather than the gradient smoothness we adopt.
> In contrast, our analysis applies to **general non-convex objectives** under weaker smoothness assumptions, making the saddle-point escape guarantees a significant and non-trivial extension.
>
> W4. We respectfully note that the standard cosine annealing schedule with warm restarts (Loshchilov & Hutter, 2017) [8] **does reset to the same initial magnitude** at the start of each cycle. As illustrated in Figure 1 of [8], the curves for cosine annealing return to the same learning rate after each restart, unlike multi-step decay schedules. We will clarify the phrasing in the manuscript to avoid any confusion regarding the cyclic nature of cosine annealing.
>
> W5. We agree that there is a distinction between fluctuating learning rates (triangular, cosine) and i.i.d. rates drawn from a uniform distribution. Our intent is to show that classical schedulers can be viewed as special cases within the broader PLRS framework: both exploit the high–low alternation mechanism, but they do so through fundamentally different dynamics.
> Regarding Defazio et al. (2024) [6], their theoretical analysis motivates **warm-up followed by decay to zero**, where the type of decay is adaptive. In contrast, PLRS does **not impose a decaying trend**; learning rates are sampled i.i.d. at each step, which enables analysis of stochastic high–low alternation and provides an explicit, auditable specification of optimizer behavior, supporting our trustworthy AI motivation. Further, it is important to note that both Defazio et al. (2024) [6] and Defazio et al. (2023) [14] work with **convex** objective functions (refer to lines 1-2 under Section 1.1 of [6] and line 1 under Section 1.1 of [14]), while we work with non-convex objectives.
>
> Q2. The cited reference [5] by the Reviewer provides convergence guarantees for perturbed GD rather than SGD. The guarantees for perturbed SGD, i.e., SGD with additive noise, is provided by Ge et al. (2015) [13]. The key technical novelty beyond Ge et al. (2015) [13] lies in handling the **multiplicative noise** induced by the i.i.d. uniform learning-rate random variable. Unlike additive noise, where the noise is decoupled from the gradient, the gradient here is multiplied by a stochastic factor, creating a **coupled noise term**. Decoupling this term is critical for deriving the high-probability bounds in our paper, and we provide a rigorous, induction-based proof for Lemma 2 to accomplish this.
> This treatment also leads to **improved iteration complexity** for escaping saddle points: our update achieves escape in \tilde{O}(L_max^{-¼}) compared to \tilde{O}(\eta^{-1}) in Ge et al.(2015) [13], Lemma 9 demonstrating a concrete technical advancement in non-convex SGD analysis.

---

> > ### Author Response · Authors · 2025-11-14
> > **Response to Reviewer 4MQU**
> >
> > **Part 2 of the response (2/2)**
> >
> > Q3. We thank the reviewer for this insightful question. As discussed earlier, classical learning-rate schedulers (triangular, one-cycle, cosine) can be viewed as **special cases within the broader PLRS framework**: both exploit the high–low alternation mechanism, but they implement it through fundamentally different dynamics.
> > Regarding the theoretical analysis, if the learning rate is deterministic or the draws are correlated, the noise term in equation (4) **no longer has zero mean**, which complicates the proofs of Theorems 2 and 3. Additionally, the martingale construction used in these theorems relies critically on the i.i.d. assumption; without it, the proof technique would need to be substantially revised. This justifies the use of i.i.d. learning-rate sampling for tractable and rigorous analysis.
> >
> > References:
> >
> > [1] Ivgi, M., Hinder, O., & Carmon, Y. (2023). DoG is SGD’s Best Friend: A Parameter‑Free Dynamic Step Size Schedule. arXiv:2302.12022.
> >
> > [2] Chen, Z., Maguluri, S. T., & Zubeldia, M. (2025). Concentration of Contractive Stochastic Approximation: Additive and Multiplicative Noise. The Annals of Applied Probability, 35(2), 1298–1352. https://doi.org/10.1214/24-AAP2143
> >
> > [3] Jofré, A., & Thompson, P. (2019). On variance reduction for stochastic smooth convex optimization with multiplicative noise. Mathematical Programming, 174(1), 253–292. https://doi.org/10.1007/s10107-018-1297-x
> >
> > [4] Faw, M., Rout, L., Caramanis, C., & Shakkottai, S. (2023). Beyond Uniform Smoothness: A Stopped Analysis of Adaptive SGD. Proceedings of the 36th Annual Conference on Learning Theory (COLT 2023), PMLR 195:1–72.
> >
> > [5] Jin, C., Ge, R., Netrapalli, P., Kakade, S. M., & Jordan, M. I. (2017). How to Escape Saddle Points Efficiently. Proceedings of the 34th International Conference on Machine Learning (ICML 2017), PMLR 70:1724–1732. (arXiv:1703.00887).
> >
> > [6] Defazio, A., Yang, X., Mehta, H., Mishchenko, K., Khaled, A., & Cutkosky, A. (2024). The road less scheduled. Advances in Neural Information Processing Systems, 37, 9974-10007.
> >
> > [7] Dauphin, Y. N., Pascanu, R., Gulcehre, C., Cho, K., Ganguli, S., & Bengio, Y. (2014). Identifying and attacking the saddle point problem in high-dimensional non-convex optimization. Advances in Neural Information Processing Systems (NeurIPS 27), 2933–2941.
> >
> > [8] Loshchilov, I., & Hutter, F. (2017). SGDR: Stochastic Gradient Descent with Warm Restarts. International Conference on Learning Representations (ICLR 2017). (arXiv:1608.03983).
> >
> > [9] Leslie N Smith. Cyclical learning rates for training neural networks. In 2017 IEEE winter conference on applications of computer vision, pp. 464–472, 2017.
> >
> > [10] Jingzhao Zhang, Tianxing He, Suvrit Sra, and Ali Jadbabaie. “Why Gradient Clipping Accelerates Training: A Theoretical Justification for Adaptivity”. In: International Conference on Learning Representations. 2020 (pages 2–5, 7, 12, 52).
> >
> > [11] Y. Nesterov, Introductory Lectures on Convex Optimization: A Basic Course. Boston, MA, USA: Kluwer Academic Publishers, 2003.
> >
> > [12] Jin, C., Netrapalli, P., Ge, R., Kakade, S.M., & Jordan, M.I. (2019). Stochastic Gradient Descent Escapes Saddle Points Efficiently. ArXiv, abs/1902.04811.
> >
> > [13] Rong Ge, Furong Huang, Chi Jin, and Yang Yuan. Escaping from saddle points—online stochastic gradient for tensor decomposition. In COLT, 2015.
> >
> > [14] Defazio, A., Cutkosky, A., Mehta, H., & Mishchenko, K. (2023). Optimal linear decay learning rate schedules and further refinements. arXiv preprint arXiv:2310.07831.

---

> > ### Comment · Reviewer_4MQU · 2025-11-24
> >
> > > In contrast, our results provide convergence guarantees for non-convex objectives, where proving convergence for non-monotone learning rates is significantly more challenging.
> >
> > Well, line search methods are also capable of increasing the learning rate and they are applicable to non-convex optimization.
> >
> > >  they do not address convergence to second-order points or escape from saddle points (as the reviewer correctly points out), which is a central contribution of our paper.
> >
> > It really seems that this paper's contribution is incredibly specific (second-order escape, with multiplicative noise). We already know how to analyze multiplicative noise, and we already know how to analyze second-order stationary point convergence with additive noise. What is so difficult about the present analysis?
> >
> > > (Faw et al., 2023) [4] use an affine multiplicative-noise formulation and assume (L_0, L_1)-smoothness rather than the gradient smoothness we adopt. In contrast, our analysis applies to general non-convex objectives under weaker smoothness assumptions, making the saddle-point escape guarantees a significant and non-trivial extension.
> >
> > Their analysis applies to non-convex objectives, and (L_0, L_1) smoothness is weaker than just L-smoothness. L-smoothness is literally a special case of (L_0,  L_1) smoothness where $L_1 = 0$. Their analysis is more general.
> >
> > > We respectfully note that the standard cosine annealing schedule with warm restarts (Loshchilov & Hutter, 2017) [8] does reset to the same initial magnitude at the start of each cycle. As illustrated in Figure 1 of [8], the curves for cosine annealing return to the same learning rate after each restart, unlike multi-step decay schedules. We will clarify the phrasing in the manuscript to avoid any confusion regarding the cyclic nature of cosine annealing.
> >
> > I did NOT say that it doesn't reset to the same initial magnitude, I said that successive cycles don't repeat at the same magnitudes. It is true that the initial magnitude is the same, but the magnitudes from the next timestep on are different. Figure 1 of [8] clearly show that this method does not cycle, since the period increases with each cycle. It is not actually cycling.
> >
> > > Regarding Defazio et al. (2024) [6], their theoretical analysis motivates warm-up followed by decay to zero, where the type of decay is adaptive. In contrast, PLRS does not impose a decaying trend; learning rates are sampled i.i.d. at each step, which enables analysis of stochastic high–low alternation and provides an explicit, auditable specification of optimizer behavior, supporting our trustworthy AI motivation.
> >
> > It still contradicts your claim that there's no analysis of increasing learning rates. I don't buy that i.i.d. learning rates tell us anything about optimizer behavior outside of "this is equivalent to just adding noise to a constant learning rate". It certainly doesn't imply any "trustworthy AI" implications.
> >
> > > The cited reference [5] by the Reviewer provides convergence guarantees for perturbed GD rather than SGD. The key technical novelty beyond Ge et al. (2015) [13] lies in handling the multiplicative noise induced by the i.i.d. uniform learning-rate random variable. Unlike additive noise, where the noise is decoupled from the gradient, the gradient here is multiplied by a stochastic factor, creating a coupled noise term.
> >
> > The method you analyze is also pretty much perturbed GD, the perturbation is just multiplicative and the decoupling is easy because the noise in the learning rates is independent from the gradient at this point.
> >
> > > This treatment also leads to improved iteration complexity for escaping saddle points: our update achieves escape in \tilde{O}(L_max^{-¼}) compared to \tilde{O}(\eta^{-1}) in Ge et al.(2015) [13], Lemma 9 demonstrating a concrete technical advancement in non-convex SGD analysis.
> >
> > The results are not comparable, because the analysis of Ge et al. (2015) gives an unconditional in-expectation guarantees whereas yours is conditional on an event that happens with probability $1-L_{\max}^{7/2}$. This coupling in the dependence of the result on $L$ means that achieving a better bound is not necessarily indicative of a better technical advancement.
> >
> > > We thank the reviewer for this insightful question. As discussed earlier, classical learning-rate schedulers (triangular, one-cycle, cosine) can be viewed as special cases within the broader PLRS framework: both exploit the high–low alternation mechanism, but they implement it through fundamentally different dynamics.
> >
> > No, they can't. Because they're not i.i.d. learning rates drawn from the same distribution. Sure, they both do high-low alternation, but that's where the similarities end.
> >
> > >  This justifies the use of i.i.d. learning-rate sampling for tractable and rigorous analysis.
> >
> > No, it doesn't, because no one actually uses i.i.d. learning rates. Analyzing them for ease is just not useful.

---

> > > ### Author Response · Authors · 2025-11-26
> > > **Response**
> > >
> > > **Line search methods**
> > >
> > > Line search methods are generally not used in neural network training, and even when modified variants are applied, there is no established convergence theory for such settings. The work in [1] provides only a first-order convergence result, relying on standard smoothness assumptions together with the strong growth condition (SGC). To date, Armijo-style line search in SGD based optimization has no known second-order convergence guarantees. If provided with a reference where the line search method is analyzed in the non-convex setting with SGD and is proven to escape saddle point/ converge to a second order stationary point, we will retract our statement.
> > >
> > > **High probability guarantees**
> > >
> > > Ge et al. (2015) does _not_ provide an unconditional guarantee. In Ge et al. (2015), Lemma 9 (saddle point escape) requires the results of the 2 key Lemmas 17 and 18. Lemma 17 holds with probability $1-\tilde{O}(\eta^3)$ and Lemma 18 holds with probability $1-\tilde{O}(\eta^2)$ and hence Lemma 16 only holds with at least probability $1-\tilde{O}(\eta^2)$. Ge et al. (2015) do not state this explicitly in their Lemma 9.
> > >
> > > Similarly, for us, Theorem 2 (saddle point escape) requires the results from Lemmas 2 and 3, each holding with probabilities $1-\tilde{O}(L_{max}^{15/4})$ and $1-\tilde{O}(L_{max}^{7/2})$ respectively. Hence, Theorem 2 holds with probability at least $1-\tilde{O}(L_{max}^{7/2})$. Therefore, we insist that our results are comparable because both of our works show guarantees conditioned on an event that happens with a certain probability, and reiterate our claim that our convergence guarantees are better than Ge et al. (2015).
> > >
> > > **Use of i.i.d. learning rates in practice**
> > >
> > > Note that all learning-rate schedulers used until our work for neural networks were deterministic, so the question of i.i.d. versus non-i.i.d. does not arise. In our case, since we sample values from a uniform distribution between $L_{\min}$ and $L_{\max}$, each sequence of sampled values across epochs can be interpreted as a valid learning-rate schedule. Moreover, we implement our probabilistic learning-rate scheduler in practice and observe comparable or superior performance relative to state-of-the-art methods. Notably, on CIFAR-10 with the WRN-28-10 architecture, we achieve an improvement of $1.96$% in maximum test accuracy. We thank the reviewer for highlighting that such a probabilistic approach has not previously been explored in either theoretical or empirical analyses for learning-rate scheduling; indeed, this further underscores the strength of our contributions and reinforces its utility.
> > >
> > > **Technical challenges**
> > >
> > > Further, we wish to point out a few of the technical difficulties with analyzing the perturbation of the gradient with a multiplicative noise rather than additive.
> > >
> > > While computing the gradient of the Taylor-approximated function with the corresponding iterates in eq. (38) of Ge et al. (2015) and eq.(16) of our paper, the second-term noise components ($\xi$ in Ge et al. (2015) and $\tilde{w}$ in ours) differ in a subtle but important way. In Ge et al. (2015), $\xi$ consists of the additive unit-sphere noise plus the difference between the stochastic and true gradients, and its norm is bounded by definition.
> > >
> > > In contrast, $\tilde{w}$ in our analysis contains $\eta_c$ multiplied by the gradient error, plus uniform noise multiplied by the stochastic gradient. Obtaining an upper bound on the norm of $\tilde{w}$, which is required in the proof of the saddle-point escape theorem, is technically more challenging because the gradient term appears simultaneously on both the left-hand side and right-hand side. To handle this dependency, we employ an induction argument to derive a valid bound on this norm.
> > >
> > > A similar difficulty arises when bounding the norm of the difference between the Taylor iterate after $t$ steps and the one at the initial iteration, specifically when comparing eq. (39) of Ge et al. (2015) with eq. (21) of our paper. These changes propagate throughout the rest of the proof, causing changes in the end results derived.
> > >
> > >  Further, in establishing the coupling between the gradient of the Taylor-approximated function and the true gradient, the expression for the additive noise in Ge et al. (2015) (eq. 50) is simpler than our corresponding term (eq. 29). This term propagates throughout the subsequent steps of the analysis, leading to differences in the overall proof structure.
> > >
> > > [1] Vaswani, S., Mishkin, A., Laradji, I., Schmidt, M., Gidel, G., & Lacoste-Julien, S. (2019). Painless stochastic gradient: Interpolation, line-search, and convergence rates. Advances in neural information processing systems, 32.

---

> > > > ### Comment · Reviewer_4MQU · 2025-11-26
> > > >
> > > > - Yes, line search methods aren't used for neural nets and I'm quite certain neither are i.i.d. learning rates drawn from a uniform distribution. If the point of invalidating line search algorithms as a point of comparison is that they're not used then neither is your method being used. While I agree their analysis doesn't work for stochastic objective, this still significantly reduces your claim from "learning rates that increase haven't been analyzed before" to "learning rates that increase, and which aren't warmup-decay have not been shown to converge To second-order stationary points for SGD", and even then your proposed way of analyzing this is also irrelevant to practice.
> > > >
> > > > - This is such a superfluous point. Ge et al. give a result that allows for arbitrarily high probability by manipulating the step size in Theorem 6, you do not show any such result. The results are not comparable until you also produced a result that occurs with arbitrarily high probability by tuning the upper range of the step sizes.
> > > >
> > > > - This doesn't underscore any "strength" of your contribution, iid learning rates are not used because they don't make any sense. There is no novelty whatsoever here and no utility in explaining actual learning rates scheduled used in practice, which are not iid and actually involve decay for most or a significant portion of the run. The theory you develop is not useful for any of that. One experiment on CIFAR10 that doesn't even achieve the SOTA test accuracy (96%+) by a mile is no evidence that iid learning rates work or that they are interesting.
> > > >
> > > > - The technical challenges you mentioned are all so trivial. An induction argument to handle a term showing on both sides of an equation is a nice trick, but nothing beyond the scope of a graduate level exercise. Fundamentally the two proofs follow very similar lines and I have no doubt any capable person could transform one into the other quite easily.
> > > >
> > > > - You also did not acknowledge that your claim about cosine learning rates was wrong.
> > > >
> > > > I am now far more confident in rejecting this paper, as this interaction clearly shows the validitiy of my concerns. I have increased my confidence accordingly.

---

> > > > > ### Author Response · Authors · 2025-12-02
> > > > > **Response**
> > > > >
> > > > > The reviewer repeatedly notes that our i.i.d. learning-rate sampling approach is not used in practice. This is puzzling, as the method is introduced for the first time in our work and is explicitly proposed as a novel contribution. It is therefore unclear why its lack of prior use is framed as a limitation. That said, we agree that the phrase "haven’t been analyzed before’’ is broader than intended, and we will revise it to "have not been shown to converge to second-order stationary points for SGD.’’
> > > > >
> > > > > Our results are, in fact, comparable to those of Ge et al. (2015). The reviewer appears to have overlooked key aspects of our contribution. In particular, concerning the specific point they raise, we also tune the value of $L_{\max}$ in our analysis to obtain high-probability bounds. In Theorem 3, where we show that once the iterate escapes a saddle point and enters the vicinity of a local minimum, it remains there for $T$ iterations with high probability; we explicitly provide the required constraints on $L_{\max}$ in Appendix D (Page 25), specifically in the first paragraph of the proof. Regarding Theorem 6 in Ge et al. (2015), the authors combine the results from the three cases: (i) large gradient, (ii) saddle point, and (iii) near a local minimum, into a single main theorem. We also establish results in these three regimes, and they can easily be presented in the same combined manner.
> > > > >
> > > > > We acknowledge that the reviewer is entitled to their view on the significance of our contribution, though we do not agree with the characterization of triviality.
> > > > >
> > > > > With respect to cosine learning rates, the hyperparameter $T_{\text{mult}}$ controls the scaling of restart intervals; when $T_{\text{mult}} = 1$, the schedule is indeed periodic. In our experiments, a value of $T_{\text{mult}} = 2$ yielded the best performance, and we adopted it. Even in this setting, the schedule remains cyclic in nature, even if not periodic, and it resets to the same base learning rate at the beginning of each cycle (Figure 1 of [1]).
> > > > >
> > > > > We strongly disagree with the statement: "One experiment on CIFAR-10 that doesn't even achieve the SOTA test accuracy (96%+) by a mile is no evidence that iid learning rates work,’’ as this is factually incorrect. Our baseline CIFAR-10 result was obtained using SGD without momentum, and this is accurate. A 2% improvement is widely regarded as significant in the deep learning community. Moreover, we have presented experiments across several datasets, not just CIFAR-10 as the reviewer says, including CIFAR-100, ImageNet-1K, Tiny ImageNet, SQuAD, IWSLT, and CommonVoice, and in all cases, we match or exceed existing results. To the best of our knowledge, when new learning-rate schedulers are proposed, they are typically evaluated empirically against existing schedulers across a similar suite of datasets, consistent with our methodology.
> > > > >
> > > > > [1] Loshchilov, I., & Hutter, F. (2017). SGDR: Stochastic Gradient Descent with Warm Restarts. International Conference on Learning Representations (ICLR 2017). (arXiv:1608.03983).

---

### Official Review · Reviewer_S5JZ · 2025-11-03

**Soundness:** 2
**Presentation:** 3
**Contribution:** 1
**Rating:** 2
**Confidence:** 4

**Summary:**

This paper proposes a Probabilistic Learning Rate Scheduler (PLRS): a randomized learning rate scheduler that samples the learning rate from the interval $U[L_{min}, L_{max}]$, where $L_{min}$ and $L_{max}$ are user-specified constants. The authors prove convergence with SGD under several assumptions including smoothness, Lipschitzness of the Hessian, and boundedness of the function. They test their approach empirically on image classification (CIFAR-10, CIFAR-100, Tiny ImageNet, ImageNet-1k), question answering (SQuAD v1.1), machine translation (IWSLT'14), and speech recognition (CommonVoice 11.0) tasks, comparing to cosine annealing, one-cycle scheduler, knee, multi-step, and constant leanring rates.

The main selling point, as presented by the authors, is the fact that their analysis allows for non-decreasing learning rate schedules. However, I went through the proofs and I see little novelty in the theory.

**Strengths:**

1. The idea is simple and is presented clearly.
2. The propoesd random scheduler is eay to implement.

**Weaknesses:**

1. First of all, the proposed analysis, which is the main contribution of the paper, is not really new. The authors use the standard descent lemma and simply rely on the assumption that the gradients are large enough to cancel out the noise term. Besides, while the previous papers mostly used decreasing learning rates it's not really required. The descent lemma already gives $\min_{t\le T}\mathbb{E}[\Vert \nabla f(x\_t)\Vert^2] \le \frac{1}{\sum_{t=0}^T \eta\_t }\sum_{t=0}^T \eta\_t \mathbb{E}[\Vert \nabla f(x\_t)\Vert^2] \le f(x_0) - f_* + \sigma^2 \sum_{t=0}^T \eta\_t^2 $. Obviously, this analysis doesn't require the stepsizes to be decreasing.
2. I'm a bit surprised that the authors decided to study cyclical learning rates as they seem to have been falling out of fashion. For instance, the cited paper of Smith (2017) has been receiving less citations as can be seen on Google Scholar. It's quite well known (look at almost any paper training state-of-the-art models) that the most widely used schedulers are warmup+stable (optional)+decay (usually cosine, linear, sometimes inverse square root).
3. The numerical results are clearly not matching state of the art. For instance, on ImageNet, the authors achieve 68.01% top-1 test accuracy, whereas one can achieve as much as 79.05%, see Table 5 in (Xi et al., "Unsupervised Data Augmentation for Consistency Training", 2020).
4. The question of novelty is also quite open here. For instance, randomized scaling of the update has been considered by Zhang and Cutkosky, "Random Scaling and Momentum for Non-smooth Non-convex Optimization", (2024).
#### Minor
I think there is a typo in the proof of Theorem 1 in equation (13): it should be $\Vert\nabla f(x\_t)\Vert^2$ rather than $\Vert\nabla f(x\_0)\Vert^2$.

**Questions:**

Where in the proofs you show that using a random learning is actually helping with the upper bounds?

---

> ### Author Response · Authors · 2025-11-15
> **Response to Reviewer S5JZ**
>
> We thank the reviewer for taking the time to read our work and for providing insightful comments. We address their concerns point by point below.
>
> W1: While Theorem 1, which covers the case when the gradient is large, relies on the gradient being large, the case where the iterate is stuck at a saddle point (Theorem 2) is not trivial and does require non-monotone learning rates. **Saddle point escape** is the main theoretical contribution of our work, and to the best of our knowledge, this has not been proved in the non-convex setting for non-monotone learning rates.
>
> W2: Thank you for the thoughtful comment. Our theoretical contribution does not focus on cyclical learning rates; instead, we introduce a probabilistic learning-rate scheduler and analyze its properties. Cyclical schedules appear only in the experimental comparison as a historical benchmark. As the reviewer notes, warmup-plus-decay schedules are now more common, and we include such schedulers (Iyer et al. (2023), Goyal et al. (2017)) [1,5] in our experiments as well.
>
> W3: The experiments in Xi et al. (2020) [2] used a batch size of 8,192 for the supervised entropy-minimization objective, together with the unsupervised data augmentation proposed in their paper. Our experimental setup differs from this, as described in Section 5.1. PLRS and knee schedulers were tested under a **consistent setup** in our work.
>
> W4: Zhang et al. (2024) [3] consider a different problem setting. Their **objective function is assumed to be Lipschitz**, while we only require boundedness. They prove convergence to a **different stationarity notion**, defined in Definition 2.2 of Section 2.1 in [1], whereas we prove convergence to an $\epsilon$-second order stationary point (SOSP). They show convergence for randomized SGD with momentum, while we analyze SGD with a randomized learning rate and prove convergence to an $\epsilon$-SOSP. Our proof technique also differs substantially: we directly analyze behavior near local maxima, saddle points, and local minima, while they develop a framework that converts online convex optimization algorithms into non-convex ones and treats SGD with momentum as a special case. They show that SGD with momentum and a scalar multiplier from an exponential distribution can yield convergence guarantees. Although their method uses a randomized scalar, it differs from ours in noise distribution and in the distinctions noted above.
>
> Q1: Unlike additive noise, where the noise term is decoupled from the gradient, here the gradient is multiplied by a stochastic factor that creates a coupled noise term. This **improves the iteration complexity** for escaping saddle points: our update escapes in $\tilde{O}(L_{\max}^{-1/4})$, compared to $\tilde{O}(\eta^{-1})$ in Ge et al. (2015) [4], as shown in Lemma 9.
>
> Minor: This is a typo. We thank the reviewer for pointing it out and will correct it in the revised version.
>
> References:
>
> [1] Iyer, N., Thejas, V., Kwatra, N., Ramjee, R., & Sivathanu, M. (2023). Wide-minima density hypothesis and the explore-exploit learning rate schedule. Journal of Machine Learning Research, 24(65), 1-37.
>
> [2] Xie, Q., Dai, Z., Hovy, E., Luong, T., & Le, Q. (2020). Unsupervised data augmentation for consistency training. Advances in neural information processing systems, 33, 6256-6268.
>
> [3] Zhang, Q., & Cutkosky, A. (2024). Random scaling and momentum for non-smooth non-convex optimization. arXiv preprint arXiv:2405.09742.
>
> [4] Rong Ge, Furong Huang, Chi Jin, and Yang Yuan. Escaping from saddle points—online stochastic gradient for tensor decomposition. In COLT, 2015.
>
> [5] Goyal, P., Dollár, P., Girshick, R., Noordhuis, P., Wesolowski, L., Kyrola, A., ... & He, K. (2017). Accurate, large minibatch sgd: Training imagenet in 1 hour. arXiv preprint arXiv:1706.02677.

---

> > ### Comment · Reviewer_S5JZ · 2025-11-26
> >
> > > While Theorem 1, which covers the case when the gradient is large, relies on the gradient being large, the case where the iterate is stuck at a saddle point (Theorem 2) is not trivial and does require non-monotone learning rates.
> >
> > As the proof of Theorem 2 is very long and technical (and is part of the appendix, which the reviewers are not even requested to read through), I can't properly evaluate your statement. I can only see that the proof of Theorem 1 has no novel components and is essentially the same is if we were using constant stepsize $\eta_c = (L_{\max} + L_{\min})/2$. Morever, from the fact that $L_{\min}$ doesn't show up in any of the provided bounds, I am guessing that it's optimal to set $L_{\min}=0$, and that this hyperparameter is unnecessary, which further makes me doubt the quality of the theoretical results. Unfortunately, I have to judge the theory based on what I can evalute given the limited time to review it.
> >
> > > Our theoretical contribution does not focus on cyclical learning rates; instead, we introduce a probabilistic learning-rate scheduler and analyze its properties.
> >
> > I understand, but the motivation for studying probabilistic learning-rate schedulers is provided as a way to understand non-monotonic learning rate schedulers. And my point was that such schedulers are actually of low practical interest.
> >
> > > The experiments in Xi et al. (2020) [2] used a batch size of 8,192 for the supervised entropy-minimization objective, together with the unsupervised data augmentation proposed in their paper.
> >
> > One can achieve the same results with batch size of 512, that's not a crucial component. In fact, increasing batch size is only useful for saving time while the best validation accuracy is usually achieved with a small-ish batch size, so this argument only makes it stranger that you didn't obtain higher validation accuracy.
> >
> > I do think, on the other hand, that diverce data augmentations are important in this setting. I encourage the authors to introduce them in their experiments as is standard in computer vision.

---

> > > ### Author Response · Authors · 2025-11-28
> > > **Respnse**
> > >
> > > **$L_{min}$ value**
> > >
> > > We thank the reviewer for their question about the hyperparameter $L_{min}$. $L_{min}$ does not show up in any of our bounds because all of our results are in big O, and as per eq. (2) in our paper, $0< L_{min} <L_{max}<1$. In practice, $L_{min}$ does hold a significant role and is tuned using the range test.
> > >
> > > **Practical interest of probabilistic LR scheduler**
> > >
> > > We understand that probabilistic LR schedulers were never used in practice till our work. We are the first ones to propose the use of a probabilistic learning rate scheduler, and we show its utility across a wide range of datasets (CIFAR-10, CIFAR-100, ImageNet-1K, Tiny ImageNet, SQuAD, CommonVoice, IWSLT) and architectures (ResNet-110, DenseNet-40-12, VGG-16, WRN-28-10, ResNet-50, BERT, Transformer encoder-decoder). Furthermore, we compare with popular schedulers such as cosine, one-cycle, knee, and multi-step decay learning rate schedulers. Based on our experiments over a wide range of ML tasks like image classification, automatic speech recognition, and NLP (question-answering), we propose that it is a good idea to incorporate them in practice with other schedulers currently being used.
> > >
> > > **ImageNet results**
> > >
> > > Regarding unsupervised data augmentation, we thank the reviewer for their suggestion. We appreciate their comment.

---

### Meta-Review · Area_Chair_qUNc · 2026-01-07

**Summary:**

1. The theoretical analysis appears to be largely standard / not novel, especially in the “large gradient” regime, and it is unclear where randomness actually improves the bounds.
2. The paper over-claims “first theory for non-monotone LR schedules,” while prior work already analyzes non-monotone or multiplicative-noise/adaptive step-size policies.
3. The motivation is misaligned with practice because cyclical/non-monotone schedules are argued to be less relevant, and i.i.d. uniform learning rates are not used in real training.
4. Experimental baselines and recipes are not representative (e.g., ImageNet setup choices like short training, disabling momentum/weight decay, limited baseline comparisons), and reported ImageNet accuracy is far from commonly cited results.
5. Key practical guidance on choosing the hyperparameters is missing.
6. Some theory assumptions and definitions feel unrealistic or unclear for deep nets (smoothness/Hessian-Lipschitz/boundedness/local strong convexity; definition wording issues), and the theory should be better connected to “why it works” in practice.
7. The choice of a uniform distribution is not well-justified; alternative distributions (e.g., long-tailed/exponential) might be more natural and could impact speed/convergence.

**Reviewer Concerns:**

1. Regarding the technical contributions, the authors argue the novelty is saddle-point escape and convergence to approximate second-order stationary points (SOSP) under multiplicative noise / non-monotone LRs, claiming improved escape complexity vs Ge et al. (2015) and emphasizing technical handling of the multiplicative coupling (e.g., an induction argument). They have also clarified the technical assumptions needed for the analysis.
    - While the significance of the technical contributions remains unclear.
2. Regarding the relation between the current work and the existing literature, the authors narrow the claim to non-convex + second-order (SOSP / strict-saddle escape) + their specific multiplicative-noise/i.i.d.-LR setting, and they distinguish cited prior work as convex-only, first-order only, different assumptions, or different noise models.
3. Regarding the motivation, the authors say cyclical schedules are only historical baselines and that modern warmup+decay baselines are also included; they further argue that i.i.d. sampling is a new proposal that yields competitive results and enables tractable high-probability analysis, though the reviewer remains unconvinced.
4. Regarding the experiments, the authors attribute gaps mainly to different training budgets and setups (e.g., fewer epochs than classic ResNet training; differing augmentation/batch size/objectives) and state comparisons were kept consistent within their chosen recipe (e.g., following Knee LR’s setup for ImageNet); they accept suggestions like adding stronger augmentations and adding momentum/WD results.
5. For hyperparameter tuning, the authors recommend using the LR range test and propose a simple heuristic (bounds near the typical constant LR for the task), plus sensitivity results claiming performance varies mildly across many bound choices; they also promise to add more momentum/WD/AdamW results.
6. Regarding the uniformly distributed learning rates, the authors say uniform is the simplest non-degenerate choice enabling tractable analysis, and extending theory/experiments to other distributions is future work.

**Reviewer Scores:**

1. For Reviewer S5JZ, their concerns about the practical relevance and experimental strength of the proposed method remain largely unresolved, thus their score might remain at 2 or 3.
2. For Reviewer 4MQU, the concerns about the technical contributions are still outstanding after discussion, so their score will remain at 2.
3. The Reviewer oxWh's concerns should have been largely resolved by the rebuttal, and their score would remain at 6.
4. The Reviewer FLZt's concerns about the unrealistic assumptions are addressed by the rebuttal, but the reviewer is not familiar with this kind of theoretical work. Their score would remain at 4 or 5.
5. The Reviewer eUrV's concerns about the choice of the uniform distribution, hyperparameter tuning, and the relation to simulated annealing should have been addressed by the rebuttal. Their score might be increased to 5 or 6.

---

### Decision · Program_Chairs · 2026-01-26

Reject